# Blocked Collaborative Bandits: Online Collaborative Filtering with Per-Item Budget Constraints

**Soumyabrata Pal**
Google Research
Bangalore, India
soumyabratapal13@gmail.com

**Arun Sai Suggala**
Google Research
Bangalore, India
arunss@google.com

**Karthikeyan Shanmugam**
Google Research
Bangalore, India
karthikeyanvs@google.com

**Prateek Jain**
Google Research
Bangalore, India
prajain@google.com

## Abstract

We consider the problem of *blocked* collaborative bandits where there are multiple users, each with an associated multi-armed bandit problem. These users are grouped into *latent* clusters such that the mean reward vectors of users within the same cluster are identical. Our goal is to design algorithms that maximize the cumulative reward accrued by all the users over time, under the *constraint* that no arm of a user is pulled more than $B$ times. This problem has been originally considered by [4], and designing regret-optimal algorithms for it has since remained an open problem. In this work, we propose an algorithm called B-LATTICE (Blocked Latent bAndiTs via maTrIx ComplEtion) that collaborates across users, while simultaneously satisfying the budget constraints, to maximize their cumulative rewards. Theoretically, under certain reasonable assumptions on the latent structure, with $M$ users, $N$ arms, $T$ rounds per user, and $C = O(1)$ latent clusters, B-LATTICE achieves a per-user regret of $\widetilde{O}(\sqrt{T(1 + NM^{-1})})$ under a budget constraint of $B = \Theta(\log T)$. These are the first sub-linear regret bounds for this problem, and match the minimax regret bounds when $B = T$. Empirically, we demonstrate that our algorithm has superior performance over baselines even when $B = 1$. B-LATTICE runs in phases where in each phase it clusters users into groups and collaborates across users within a group to quickly learn their reward models.

## 1 Introduction

Modern recommendation systems cater to millions of users and items [1] on a daily basis, typically in an online fashion. A critical feature of such systems is to quickly learn tastes of individual users from sequential actions and feedback, and suggest personalized products for each user. Furthermore, in practice, an item that has already been consumed by a user is recommended very few times to the same user (or *not* recommended at all). This is because, in applications such as movie/book recommendations, a typical user will find little interest in consuming the same item multiple times.

This problem – that we rename as *Blocked Collaborative Bandits* – was abstracted out by [4] with a modeling assumption that users have a latent clustering structure. That is, each user can belong to an unknown cluster and the expected reward for an item/movie is same for all users in a cluster. Formally, consider $M$ users, $N$ items and $T$ rounds with $M, N \gg T$ ($M, N \approx 10^6$ in recommendation systems such as YouTube) and in each round, every user is recommended some item (potentially different for each user). On consuming the item, a noisy reward is assigned. As mentioned above,

37th Conference on Neural Information Processing Systems (NeurIPS 2023).

| Paper | Setting | Metric | Guarantees (Worst-Case) |
|---|---|---|---|
| Bresler et al., 2014 [4] | $B = 1$, user clusters | pseudo regret (maximize likeable items) | $O(T)$ |
| Bresler et al., 2018 [5] | $B = 1$, user, item clusters; noiseless rewards | regret | $\widetilde{O}(1)$ |
| Ariu et al., 2020 [2] | $B = 1$, user, item clusters | regret | $\widetilde{O}(T^{2/3}(1 + NM^{-1})^{1/3})$ sub-optimal in $N, T$ |
| Pal et al., 2023 [25] | $B = T$, user clusters | regret | $\widetilde{O}(\sqrt{T(1 + NM^{-1})})$ minimax optimal in $N, M, T$ |
| **This Work** | $B = \Theta(\log T)$, user clusters | regret | $\widetilde{O}(\sqrt{T(1 + NM^{-1})})$ |

Table 1: Comparison of various approaches for blocked collaborative bandits. All the regret bounds stated here are worst-case bounds and assume $C = O(1)$. Moreover, the regret is averaged across users. The worst-case pseudo-regret in [5] is linear when the items have rewards close to $0.5$. In [2], the authors' proposed a greedy algorithm whose worst-case has $T^{2/3}$ dependence.

it is assumed that the users can be clustered in $C$ clusters, where $C \ll M, N$, and users in the same cluster have identical reward distributions over items. Furthermore, any item can be recommended to a particular user at most $B$ times, after which the item is *blocked* for the user.

[4] considered this problem for $B = 1$ in the setting where a user in cluster $c$ on being recommended item $j$ provides a like $(+1)$ with probability $p_{cj}$ and a dislike $(-1)$ with probability $1 - p_{cj}$. The authors studied a notion of pseudo-regret corresponding to minimizing the number of *un-likeable* items for a user, *i.e.,* items for which the probability of user giving a like $(+1)$ is less than $1/2$. To this end, the authors proposed the `Collaborative-Greedy` algorithm, an $\epsilon$-greedy style algorithm that performs random exploration in each round with certain probability. During exploitation it provides recommendations based on a neighborhood of similar users. However, maximizing the number of *likeable* items is limiting as it does not prioritize items with large rewards and completely disregards the item ordering. Despite these theoretical limitations, variants of the collaborative algorithm designed in [4] have found applications in predicting Bitcoin price, [29], information retrieval [12] among others.

Recent works have studied this problem under a more practical notion of regret which involves maximizing the cumulative rewards accrued over time [2, 5]. However, these works assume a cluster structure among both users and items. This assumption entails that there are many copies of the highest rewarding item for any user; this voids the blocking constraint and makes the problem significantly easier. Moreover, the algorithms designed in [2] are greedy (they perform exploration first to cluster users, items, and then perform exploitation) and achieve significantly sub-optimal regret. These bounds, translated to the original blocked bandit problem where there are no item clusters, have sub-optimal dependence on the number of items $N$. The algorithms designed in [5] assumed a *noiseless* binary feedback model which allowed for constant regret. However, these algorithms are not easily extendable to the more general noisy reward setting we consider in this work.

Another line of work has studied this problem when $B = T$ (i.e., no budget constraints on arms) [24, 11, 25]. While some of these works have designed regret optimal algorithms [25], the $B = T$ budget constraint is too lenient in many modern recommendation systems. To summarize, existing works have either attempted to solve the much harder setting of $B = 1$ by imposing additional restrictions on the problem or the simpler $B = T$ setting which is not very relevant in practice.

**This Work.** In this work, we make progress on this problem by considering the intermediate setting of $B = \Theta(\log T)$. We do not impose any cluster structure among items (as in [4]), and consider a general notion of regret which involves maximizing the cumulative rewards under the budget constraints. We propose `B-LATTICE`(Alg. 1), a phased algorithm which carefully balances exploration and exploitation. Our algorithm also performs on-the-fly clustering of users and collaborates across users within a cluster to quickly learn their preferences, while simultaneously satisfying the budget constraints. The key contribution of our work is that under certain standard incoherence assumptions (also used recently in [25]) and a budget constraint of $B = \Theta(\log T)$, we show that `B-LATTICE` achieves a per-user regret of $\widetilde{O}(\sqrt{T(1 \bigvee NM^{-1})})$ [1]. Empirically, our algorithm can even handle a budget constraint of $B = 1$ and perform better than baselines (Alg. 6). However, bounding its regret

---

[1] $\widetilde{O}(\cdot), \widetilde{\Omega}(\cdot)$ hides logarithmic factors in $M, N, T$

in this setting is much more challenging which we aim to address in a future work. That being said, under an additional cluster structure on the items, we show that our theoretical guarantees hold even for $B = 1$ (Appendix F).

We also provide a non-trivial regret lower bound of $\widetilde{\Omega}(\sqrt{NM^{-1}} \bigvee TM^{-1/2})$. Our proof involves a novel reduction to multi-hypothesis testing and relies on Fano's inequality for approximate recovery [28] (see Appendix E). Our techniques could be of independent interest even outside the context of this work. Our lower bound is tight in two regimes - when the number of rounds $T$ is very small or very large. However, we conjecture that our upper bounds are actually tight since they recover the same rates as in the $B = T$ setting [25]; tightening the lower bound is left as a future work. Finally, we verify our theoretical results by simulations on synthetic data (see Appendix C). Here, we compare a more practical version of `B-LATTICE`(Algorithm 6) with `Collaborative-Greedy` [4]. We demonstrate that `B-LATTICE` not only has good regret but also other practically desirable properties such as a small cold-start period, and repeated high quality recommendations.

**Techniques.** Our algorithm is built on recent works that have studied this problem without the budget constraint [17, 25]. [17] developed a regret optimal algorithm called LATTICE that runs in phases. At any phase, the algorithm maintains a grouping of users which it refines over time (these groups are nothing but our current estimate of user clusters). Within each group, the algorithm relies on collaboration to quickly estimate the reward matrix. To be precise, the algorithm performs random exploration followed by low-rank matrix completion [8] to get a good estimate of the user-item reward matrix. Next, the algorithm uses the estimated reward matrix to eliminate sub-optimal arms for each user. Finally, it refines the user clusters by placing users with similar reward structure in the same group.

Extending the above algorithmic recipe to our setting poses a couple of challenges. First, observe that the *oracle* optimal strategy in the budget constrained setting is to recommend each of the top $T/B$ items $B$ times; we refer to these top times as *golden* items in the sequel. To compete against such a policy, our algorithm needs to quickly identify the golden items for each user. The LATTICE algorithm described above doesn't perform this, as it aims to only identify the top item for every user. So, one of the key novelties in our algorithm is to design a test to quickly identify the golden items and recommend them to the users. The second challenge arises from the usage of low-rank matrix completion oracles in LATTICE. To be precise, LATTICE requires more accurate estimates of the user-item reward matrix in the later phases of the algorithm. To this end, it repeatedly recommends an item to a user to reduce the noise in its estimates. However, this is infeasible in our setting due to the budget constraints. So, the second novelty in our algorithm is to avoid repeated recommendations and design an alternate exploration strategy that adheres to the budget constraints.

**Other Related Work:**

*Item-Item Collaborative Filtering (CF).* A complementary theoretical line of work proposes to exploit a clustering structure across the items instead of users [7, 23, 27, 22]. Under a similar blocking constraint, the authors have provided a sub-linear regret guarantee based on a certain measure of complexity called the doubling dimension. Since then, there have been several works in the literature that have attempted to exploit a cluster structure on both users and items [6, 2].

*Variants of User-User CF.* [15] has looked into the problem of non-stationary user-user collaborative filtering where the preferences of users change over time. [13] studies a variant where the user only provides positive ratings i.e when the user has liked an item. [9] and [3] studies probabilistic models for user user CF in an online and offline model respectively.

*Cluster Structure across users.* In several problems related to multi-armed bandits, cluster structure across users have been explored. In particular, in [11, 20, 10, 21, 26], the authors have imposed a cluster-structure across users while each item has a $d$-dimensional context chosen uniformly at random at each round. The preferences of each user is a linear function of the context vector and the cluster id. Under these settings, the authors prove a strong regret bound. However, note that in our setting, the item contexts are hidden; hence, these techniques are not applicable to our setting.

## 2 Problem Setting and Background

**Notation.** We write $[m]$ to denote the set $\{1, 2, \ldots, m\}$. For a vector $\mathbf{v} \in \mathbb{R}^m$, $\mathbf{v}_i$ denotes the $i^{\text{th}}$ element; for any set $\mathcal{U} \subseteq [m]$, let $\mathbf{v}_{\mathcal{U}}$ denote the vector $\mathbf{v}$ restricted to the indices in $\mathcal{U}$. Similarly,

for $\mathbf{A} \in \mathbb{R}^{m \times n}$, $\mathbf{A}_{ij}$, $\mathbf{A}_i$ denotes the $(i,j)$-th element and the $i^{\text{th}}$ row of $\mathbf{A}$ respectively. For any set $\mathcal{U} \subseteq [m], \mathcal{V} \subseteq [n]$, $\mathbf{A}_{\mathcal{U},\mathcal{V}}$ denotes $\mathbf{A}$ restricted to the rows in $\mathcal{U}$ and columns in $\mathcal{V}$. Let $\|\mathbf{A}\|_\infty$ denote absolute value of the largest entry in matrix $\mathbf{A}$. $\mathcal{N}(0, \sigma^2)$ denotes the Gaussian distribution with 0 mean and variance $\sigma^2$. We write $\mathbb{E}X$ to denote the expectation of a random variable $X$. $\|\mathbf{U}\|_{2,\infty}$ corresponds to the maximum euclidean norm of a row of the matrix $\mathbf{U}$. More precisely, for a matrix $\mathbf{U} \in \mathbb{R}^{M \times r}$, the norm $\|\mathbf{U}\|_{2,\infty} = \max_{i \in [M]} \|\mathbf{U}_i\|_2$. Thus, if $\|\mathbf{U}\|_{2,\infty}$ is small as in Lemma 1, then all rows of $\mathbf{U}$ have a small $\ell_2$ norm.

**Problem Setting.** Consider an online recommendation system with M users, N items and T rounds. Let $\mathbf{P} \in \mathbb{R}^{M \times N}$ be the reward matrix which is unknown. Here, we assume the set of M users can be partitioned into C disjoint but unknown clusters $\mathcal{C}^{(1)}, \mathcal{C}^{(2)}, \ldots, \mathcal{C}^{(C)}$ such that any two users $u, v \in [M]$ belonging to the same cluster have identical reward vectors i.e. $\mathbf{P}_u = \mathbf{P}_v$. In each round $t \in [T]$, every user is recommended an item (can be different for each user) by the system. In turn, the system receives feedback in the form of reward from each user. Let, $\mathbf{R}^{(t)}_{u\rho_u(t)}$ be the observed reward for recommending item $\rho_u(t) \in [N]$ to user $u$ at round $t$ such that:

$$\mathbf{R}^{(t)}_{u\rho_u(t)} = \mathbf{P}_{u\rho_u(t)} + \mathbf{E}^{(t)}_{u\rho_u(t)} \tag{1}$$

where $\mathbf{E}^{(t)}_{u\rho_u(t)}$ denotes the unbiased additive noise. [2] We assume that elements of $\{\mathbf{E}^{(t)}_{u\rho_u(t)}\}_{\substack{u \in [M] \\ t \in [T]}}$ are i.i.d. zero mean sub-gaussian random variables with variance proxy $\sigma^2$ i.e. for all $u, t \in [M] \times [T]$, we have $\mathbb{E}\mathbf{E}^{(t)}_{u\rho_u(t)} = 0$ and for all $s \in \mathbb{R}$, we have $\mathbb{E}\exp(s\mathbf{E}^{(t)}_{u\rho_u(t)}) \leq \exp(\sigma^2 s^2/2)$. As in practical recommendation systems, we impose an additional constraint that the same item cannot be recommended more than B times to a user, for some small $B = \Theta(\log T)$. For simplicity of presentation, we assume T is a multiple of B. However, we would like to note that our results hold for general T. Without loss of generality, we assume $N \geq T/B$, since otherwise the budget constraints cannot be satisfied.

Our goal is to design a method that maximizes cumulative reward. Let $\pi_u : [N] \to [N]$ denote the function that sorts the rewards of user $u \in [M]$ in descending order, i.e., for any $i < j$, $\mathbf{P}_{u\pi_u(i)} \geq \mathbf{P}_{u\pi_u(j)}$. The *oracle* optimal strategy for this problem is to recommend $\{\pi_u(s)\}_{s=1\ldots T/B}$, the top T/B items with the highest reward, B times each. This leads us to the following notion of regret

$$\text{Reg}(T) \triangleq \sum_{s \in [T/B], u \in [M]} \frac{B\mathbf{P}_{u\pi_u(s)}}{M} - \mathbb{E} \sum_{t \in [T], u \in [M]} \frac{\mathbf{P}_{u\rho_u(t)}}{M}, \tag{2}$$

where the expectation is over the randomness in the policy and the rewards, and $\rho_u(t)$ is any policy that satisfies the budget constraints.

**Importance of collaboration.** Suppose we treat each user independently and try to minimize their regret. In this case, when B is as small as $O(\log T)$, we will incur a regret that is almost linear in T. This is because we will not have enough data for any item to know if it has a high or a low reward. This shows the importance of collaboration across users to achieve optimal regret. The latent cluster structure in our problem allows for collaboration and sharing information about items across users.

For ease of exposition of our ideas, we introduce a couple of definitions.

**Definition 1.** *A subset of users $\mathcal{S} \subseteq [M]$ is called "nice" if $\mathcal{S} \equiv \bigcup_{j \in \mathcal{A}} \mathcal{C}^{(j)}$ for some $\mathcal{A} \subseteq [C]$. In other words, $\mathcal{S}$ can be represented as the union of some subset of clusters.*

**Definition 2.** *For any user $u \in [M]$, we call the set of items $\{\pi_u(t)\}_{t=1}^{T/B}$ (i.e., the set of best T/B items) to be the golden items for user $u$.*

---

[2]This corresponds to the following analogue of multi-agent multi-armed bandits (MAB) - we have M users each involved in a separate MAB problem with the same set of N arms (corresponding to the N items) and T rounds. The mean reward of each arm is different for each user captured by the $M \times N$ reward matrix $\mathbf{P}$. In each round $t$, every agent $u \in [M]$ simultaneously pulls an unblocked arm $\rho_u(t)$ of their choice based on the feedback history of all users (including $u$) from previous rounds. Subsequently the noisy feedback for all M users and their corresponding arm pulls at round $t$ is revealed to everyone.

## 2.1 Low-Rank Matrix Completion

**Additional Notation.** For an estimate $\widehat{\mathbf{P}}$ of reward matrix $\mathbf{P}$, we will use $\widetilde{\pi}_u : [\mathsf{N}] \to [\mathsf{N}]$ to denote the items ordered according to their estimated reward for the user $u$ i.e., $\widehat{\mathbf{P}}_{u\widetilde{\pi}_u(i)} \geq \widehat{\mathbf{P}}_{u\widetilde{\pi}_u(j)}$ when $i < j$. For any user $u \in [\mathsf{M}]$ and any subset of items $\mathcal{A} \subseteq [\mathsf{N}]$, we will use $\pi_u \mid \mathcal{A} : [|\mathcal{A}|] \to [\mathsf{N}]$ to denote the permutation $\pi_u$ restricted to items in $\mathcal{A} \subseteq [\mathsf{N}]$ i.e. for any user $u \in [\mathsf{M}]$ and any $i, j \in [|\mathcal{A}|]$ satisfying $i < j$, we will have $\mathbf{P}_{u\pi_u(i)|\mathcal{A}} \geq \mathbf{P}_{u\pi_u(j)|\mathcal{A}}$. Here, $\pi_u(i) \mid \mathcal{A}$ corresponds to the $i^{\text{th}}$ item among items in $\mathcal{A}$ sorted in descending order of expected reward for user $u$.

An important workhorse of our algorithm is low-rank matrix completion, which is a well studied problem in the literature [16, 18, 8, 17]. Given a small set of entries that are randomly sampled from a matrix, these algorithms infer the missing values of the matrix. More precisely, consider a low-rank matrix $\mathbf{Q} \in \mathbb{R}^{\mathsf{M} \times \mathsf{N}}$. Let $\Omega \subseteq [\mathsf{M}] \times [\mathsf{N}]$ be a random set of indices obtained by picking each index in $[\mathsf{M}] \times [\mathsf{N}]$ independently with probability $p > 0$. Let $\{\mathbf{Z}_{ij}\}_{(i,j) \in \Omega}$ be the corresponding noisy entries (sub-gaussian random variables with variance proxy $\sigma^2 > 0$) that we observe which satisfy $\mathbb{E}\mathbf{Z}_{ij} = \mathbf{Q}_{ij} \forall (i,j) \in \Omega$. In this work we rely on nuclear norm minimization to estimate $\mathbf{Q}$ (see Algorithm 4). Recent works have provided tight element-wise recovery guarantees for this algorithm [8, 17]. We state these guarantees in the following Lemma.

**Lemma 1** (Lemma 2 in [17]). *Consider matrix $\mathbf{Q} \in \mathbb{R}^{\mathsf{M} \times \mathsf{N}}$ with rank $r$ and SVD decomposition $\bar{\mathbf{U}}\mathbf{\Sigma}\bar{\mathbf{V}}^{\mathsf{T}}$ satisfying $\|\bar{\mathbf{U}}\|_{2,\infty} \leq \sqrt{\mu r/\mathsf{M}}, \|\bar{\mathbf{V}}\|_{2,\infty} \leq \sqrt{\mu r/\mathsf{N}}$ and condition number $\kappa = O(1)$. Let $d_1 = \max(\mathsf{M}, \mathsf{N})$, $d_2 = \min(\mathsf{M}, \mathsf{N})$, noise variance $\sigma^2 > 0$, and let sampling probability $p$ be such that $p = \widetilde{\Omega}\left(\frac{\mu^2}{d_2}\max\left(1, \frac{\sigma^2\mu}{\|\mathbf{P}\|_\infty^2}\right)\right)$ (for some constant $c > 0$). Suppose we sample a subset of indices $\Omega \subseteq [\mathsf{M}] \times [\mathsf{N}]$ such that each tuple of indices $(i,j) \in [\mathsf{M}] \times [\mathsf{N}]$ is included in $\Omega$ independently with probability $p$. Let $\{\mathbf{Z}_{ij}\}_{(i,j) \in \Omega}$ be the noisy observations corresponding to all indices in $\Omega$. Then Algorithm 4, when run with inputs $(\mathsf{M}, \mathsf{N}, \sigma^2, r, \Omega, \{\mathbf{Z}_{ij}\}_{(i,j) \in \Omega})$, returns an estimate $\widetilde{\mathbf{Q}}$ of $\mathbf{Q}$ which satisfies the following error bounds with probability at least $1 - O(\delta + d_2^{-12})$*

$$\|\mathbf{Q} - \widetilde{\mathbf{Q}}\|_\infty = O\left(\frac{\sigma\sqrt{\mu^3 \log d_1}}{\sqrt{pd_2}}\right). \tag{3}$$

Lemma 1 provides guarantees on the estimate $\widetilde{\mathbf{Q}}$ of $\mathbf{Q}$ given the set of noisy observations corresponding to the entries in $\Omega$. Equation 3 says that the quality of the estimate becomes better with the increase in sampling probability $p$ that determines the number of entries in $\Omega$. Note the detailed Algorithm ESTIMATE (Alg. 4) designed in [17, 8] is provided in Appendix A.

**Remark 1** (Warm-up (Greedy Algorithm)). *Using Lemma 1, it is easy to design a greedy algorithm (Alg. 5 in Appendix B) for $\mathsf{B} = 1$; for a fixed number of rounds $m$, we recommend random unblocked items to every user. Since the reward matrix $\mathbf{P}$ is a low-rank matrix, we use Algorithm 4 to estimate the entire matrix. Subsequently, we recommend the best estimated unblocked items for each user for the remaining rounds. We can prove that the regret suffered by such an algorithm is $\widetilde{O}(\mathsf{T}^{2/3}(1 \bigvee (\mathsf{N}/\mathsf{M}))^{1/3})$. The dependence on number of rounds $\mathsf{T}$ is sub-optimal, but Alg. 5 does not require the knowledge of any gaps corresponding to the reward matrix $\mathbf{P}$. See Appendix B for a detailed proof. We would like to note that the result can be generalized easily to $\mathsf{B} = \Theta(\log \mathsf{T})$.*

## 3 Main Results

Let $\mathbf{X} \in \mathbb{R}^{\mathsf{C} \times \mathsf{N}}$ be the sub-matrix of the expected reward matrix $\mathbf{P}$ comprising $\mathsf{C}$ distinct rows of $\mathbf{P}$ corresponding to each of the $\mathsf{C}$ true clusters. Also, let $\tau := \max_{i,j \in [r]} |\mathcal{C}^{(i)}|/|\mathcal{C}^{(j)}|$ denote the ratio of the maximum and minimum cluster size. To obtain regret bounds, we make the following assumptions on the matrix $\mathbf{X}$:

**Assumption 1** (Assumptions on $\mathbf{X}$). *Let $\mathbf{X} = \mathbf{U}\mathbf{\Sigma}\mathbf{V}^{\mathsf{T}}$ be the SVD of $\mathbf{X}$. Also, let $\mathbf{X}$ satisfy the following: 1) Condition number: $\mathbf{X}$ is full-rank and has non zero singular values $\lambda_1 > \cdots > \lambda_\mathsf{C}$ with condition number $\lambda_1/\lambda_\mathsf{C} = O(1)$, 2) $\mu$-incoherence: $\|\mathbf{V}\|_{2,\infty} \leq \sqrt{\mu\mathsf{C}/\mathsf{N}}$, 3) Subset Strong Convexity: For some $\alpha$ satisfying $\alpha \log \mathsf{M} = \Omega(1)$, $\gamma = \widetilde{O}(1)$ for all subset of indices $\mathcal{S} \subseteq [\mathsf{M}], |\mathcal{S}| \geq \gamma\mathsf{C}$, the minimum non zero singular value of $\mathbf{V}_\mathcal{S}$ must be at least $\sqrt{\alpha |\mathcal{S}|/\mathsf{M}}$.*

**Remark 2** (Discussion on Assumption 1). *We need Assumption 1 only to bound the incoherence factors and condition numbers of sub-matrices whose rows correspond to a nice subset of users and the columns comprise of at least $\mathsf{T}^{1/3}$ golden items for each user in that nice subset. This implies that information is well spread-out across the entries instead of being concentrated in a few. Thus, we invoke Lemma 1 for those sub-matrices and utilize existing guarantees for low rank matrix completion. Our theorem statement can hold under significantly weaker assumptions but for clarity, we used the theoretically clean and simple to state Assumption 1. Note that the assumption is not required to run our algorithm or any offline matrix completion oracle - the only purpose of the assumption is to invoke existing theoretical guarantees of offline low rank matrix completion algorithms with $\ell_\infty$ error guarantees.*

**Assumption 2.** *We will assume that $\mu, \sigma, \tau, ||\mathbf{P}||_\infty, \mathsf{C}$ are positive constants and do not scale with the number of users $\mathsf{M}$, items $\mathsf{N}$ or the number of rounds $\mathsf{T}$.*

Assumption 2 is only for simplicity of exposition/analysis and is standard in the literature. As in [25], we can easily generalize our guarantees if any of the parameters in Assumption 2 scale with In that case, the regret has additional polynomial factors of $\mu, \sigma, \tau, ||\mathbf{P}||_\infty, \mathsf{C}$ - moreover these quantities (or a loose upper bound) are assumed to be known to the algorithm. Now, we present our main theorem:

**Theorem 1.** *Consider the blocked collaborative bandits problem with $\mathsf{M}$ users, $\mathsf{N}$ items, $\mathsf{C}$ clusters, $\mathsf{T}$ rounds with blocking constraint $\mathsf{B} = \Theta(\log \mathsf{T})$ [3] such that at every round $t \in [\mathsf{T}]$, we observe reward $\mathbf{R}^{(t)}$ as defined in eq. (1) with noise variance proxy $\sigma^2 > 0$. Let $\mathbf{P} \in \mathbb{R}^{\mathsf{M} \times \mathsf{N}}$ be the expected reward matrix and $\mathbf{X} \in \mathbb{R}^{\mathsf{C} \times \mathsf{M}}$ be the sub-matrix of $\mathbf{P}$ with distinct rows. Suppose Assumption 1 is satisfied by $\mathbf{X}$ and Assumption 2 is true. Then Alg. $\mathtt{B\text{-}LATTICE}$ initialized with phase index $\ell = 1, \mathcal{M}^{(1)} = [[\mathsf{M}]], \mathcal{N}^{(1)} = [[\mathsf{N}]], \mathsf{C}, \mathsf{T}, t_0 = 1, t_{\mathsf{exploit}} = 0, \widetilde{\mathbf{P}} = \mathbf{0}$ and $\epsilon_1 = c(\log \mathsf{M})^{-1}$ for some constant $c > 0$ guarantees the regret $\mathsf{Reg}(\mathsf{T})$ defined under the blocking constraint (eq. 2) to be:*

$$\mathsf{Reg}(\mathsf{T}) = \widetilde{O}\Big(\sqrt{\mathsf{T} \max\Big(1, \frac{\mathsf{N}}{\mathsf{M}}\Big)}\Big) \tag{4}$$

Note that the regret guarantee in Theorem 1 is small as long as $\mathsf{N} \approx \mathsf{M}$ (but $\mathsf{M}, \mathsf{N}$ can be extremely large). The dependence of the regret on number of rounds scales as $\sqrt{\mathsf{T}}$. This result also matches the $\sqrt{\mathsf{T}}$ dependence on the number of rounds without the blocking constraint [17, 25]. Furthermore, Theorem 1 also highlights the importance of collaboration across users - compared with standard multi-armed bandits, the effective number of arms is now the number of items per user.

**Remark 3.** *Note that in our model, we have assumed that users in the same latent cluster have identical true rewards across all items. Although we have made this assumption for simplicity as in other works in the literature [25, 2, 3, 5], the assumption of each user having exactly 1 of $\mathsf{C}$ different mean vectors can be relaxed significantly. In fact, in [25], a similar cluster relaxation was done in the following way for a known value of $\nu > 0$ - there are $\mathsf{C}$ clusters such that 1) users in the same cluster have same best item and mean reward vectors that differ entry-wise by at most $\nu$ 2) any 2 users in different clusters have mean reward vectors that differ entry-wise by at least $20\nu$. Our analysis can be extended to the relaxed cluster setting with an addition cost in regret in terms of $\nu$.*

We now move on to a rigorous lower bound in this setting.

**Theorem 2.** *Consider the blocked collaborative bandits problem with $\mathsf{M}$ users, $\mathsf{N}$ items, $\mathsf{C} = 1$ cluster, reward matrix $\mathbf{P} \in [0, 1]^{\mathsf{M} \times \mathsf{N}}$, $\mathsf{T}$ rounds with blocking constraint $\mathsf{B}$ such that at every round $t \in [\mathsf{T}]$, we observe reward $\mathbf{R}^{(t)}$ as defined in eq. (1) with noise random variables $\{\mathbf{E}_{u\rho_u(t)}^{(t)}\}_{\substack{u \in [\mathsf{M}] \\ t \in [\mathsf{T}]}}$ generated i.i.d according to a zero mean Gaussian with variance $1$. In that case, any algorithm must suffer a regret of at least*

$$\mathsf{Reg}(\mathsf{T}) = \Omega\Big(\max\Big(\sqrt{\frac{\mathsf{NB}}{\mathsf{M}}}, \frac{\mathsf{T}\sqrt{\log(\mathsf{N}/\mathsf{T})}}{\mathsf{B}\sqrt{\mathsf{M}}}\Big)\Big). \tag{5}$$

Note that the main difficulty in proving tight lower bounds in the blocked collaborative bandits setting is that, due to the blocking constraint of $\mathsf{B}$, a single item having different rewards can only cause a

---

[3]It suffices for us if $\mathsf{B} = c \log \mathsf{T}$ for any constant $c > c'$ where $c'$ is a constant independent of all other model parameters. This is because, we simply want $B$ to be at least the number of phases. Since our phase lengths increase exponentially, the number of phases is $O(\log \mathsf{T})$.

difference in regret of at most B in two separate instances constructed in standard measure change arguments [19]. However, we prove two regret lower bounds on this problem, out of which the latter is the technically more interesting one.

The first term in the lower bound in Thm. 2 follows from a simple reduction from standard multi-armed bandits. Intuitively, if we have $\mathsf{T}/\mathsf{B}$ known identical copies of each item, then the blocking constraint is void - but with $\mathsf{C} = 1$, this is (almost) equivalent to a standard multi-armed bandit problem with $\mathsf{MT}$ rounds, $\mathsf{NBT}^{-1}$ distinct arms (up to normalization with $\mathsf{M}$). The lower bound follows from invoking standard results in MAB literature. For $\mathsf{B} = \mathsf{T}$, we recover the matching regret lower bound in [25] without the blocking constraint. Furthermore, for $\mathsf{B} = \Theta(\log \mathsf{T})$, the first term is tight up to log factors when number of rounds is small for example when $\mathsf{T} = O(1)$. (Appendix E)

The second term in the lower bound is quite non-trivial. Note that the second term is tight up to log factors when the number of rounds $\mathsf{T} = \Theta(\mathsf{N})$ is very large (close to the number of items) and $\mathsf{B} = O(\log \mathsf{T})$ is small. We obtain this bound via reduction of the regret problem to a multiple hypothesis testing problem with exponentially many instances and applying Fano's inequality [28] (commonly used in proving statistical lower bounds in parameter estimation) for approximate recovery. Our approach might be of independent interest in other online problems as well. (Appendix E)

**Remark 4.** *We leave the problem of extending our guarantees for* $\mathsf{B} = 1$ *as future work. However we make progress by assuming a cluster structure over items as well. More precisely, suppose both items and users can be grouped into a constant number of disjoint clusters as such that (user,item) pairs in same cluster have same expected reward. Then, we can sidestep the statistical dependency issues in analysis of Alg. 1 for* $\mathsf{B} = 1$ *and provide similar guarantees as in Thm. 1 (see Appendix F).*

## 4 `B-LATTICE` **Algorithm**

`B-LATTICE` is a recursive algorithm and runs in $O(\log \mathsf{T})$ phases of exponentially increasing length. `B-LATTICE` assume the knowledge of the following quantities - users $\mathsf{M}$, items $\mathsf{N}$, clusters $\mathsf{C}$, rounds $\mathsf{T}$, blocking constraint $\mathsf{B}$, maximum true reward $||\mathbf{P}||_\infty$, incoherence factor $\mu$, cluster size ratio $\tau$, noise variance $\sigma^2 > 0$ and other hyper-parameters in Assumption 1. At any phase, the algorithm maintains a partitioning of users into crude clusters, and a set of active items for each such crude cluster. Let $\mathcal{M}^{(\ell)}$ be the partitioning of users in the $\ell^{th}$ phase, and $\mathcal{N}^{(\ell)}$ be the list containing the set of active items for each group of users in $\mathcal{M}^{(\ell)}$. In the first phase, we place all users into a single group and keep all items active for every user; that is, $\mathcal{M}^{(1)} = [[\mathsf{M}]], \mathcal{N}^{(1)} = [[\mathsf{N}]]$. There are three key components in each phase: (a) exploration, (b) exploitation, and (c) user clustering. In what follows, we explain these components in detail. For simplicity of exposition, suppose Assumption 2 is true namely the parameters $\mu, \sigma, \tau, ||\mathbf{P}||_\infty, \mathsf{C}$ are constants.

**Exploration (Alg. 2).** The goal of the `Explore` sub-routine is to gather enough data to obtain a better estimate of the reward matrix. To be precise, let $\mathcal{M}^{(\ell,i)}$ be the users in the $i^{th}$ group at phase $\ell$, and $\mathcal{N}^{(\ell,i)}$ be the set of active items for this group. Let $\mathbf{P}_{\mathcal{M}^{(\ell,i)}, \mathcal{N}^{(\ell,i)}}$ be the sub-matrix of $\mathbf{P}$ corresponding to rows $\mathcal{M}^{(\ell,i)}$ and columns $\mathcal{N}^{(\ell,i)}$ (recall, this sub-matrix has rank at most $\mathsf{C}$). Our goal is to get an estimate $\widetilde{\mathbf{P}}_{\mathcal{M}^{(\ell,i)}, \mathcal{N}^{(\ell,i)}}$ of this matrix that is entry-wise $O(2^{-\ell})$ close to the true matrix. To do this, for each user in $\mathcal{M}^{(\ell,i)}$, we randomly recommend items from $\mathcal{N}^{(\ell,i)}$ with probability $p = O\left(\frac{2^{2\ell} \log d_1}{d_2}\right)$ (Line 6). Here, $d_1 = \max(|\mathcal{M}^{(\ell,i)}|, |\mathcal{N}^{(\ell,i)}|)$, $d_2 = \min(|\mathcal{M}^{(\ell,i)}|, |\mathcal{N}^{(\ell,i)}|)$. We then rely on low-rank matrix completion algorithm (Algorithm 4) to estimate the low rank sub-matrix. By relying on Lemma 1, we show that our estimate is entry-wise $O(2^{-\ell})$ accurate. This also shows that our algorithm gets more accurate estimate of $\mathbf{P}_{\mathcal{M}^{(\ell,i)}, \mathcal{N}^{(\ell,i)}}$ as we go to the later phases.

**User Clustering (lines 7-8 of Alg. 1).** After the *exploration* phase, we refine the user partition to make it more fine-grained. An important property we always want to ensure in our algorithm is that $\mathcal{M}^{(\ell,i)}$, the $i^{th}$ group/crude cluster in phase $\ell$, is a *nice* subset for all $(\ell, i)$ (see Definition 1 for a definition of nice subset). To this end, we build a user-similarity graph whose nodes are users in $\mathcal{M}^{(\ell,i)}$, and draw an edge between two users if they have similar rewards for all the arms in $\mathcal{N}^{(\ell,i)}$ (Line 8 in Alg. 1). Next, we partition $\mathcal{M}^{(\ell,i)}$ based on the connected components of the graph. That is, we group all the users in a connected component into a single cluster. In our analysis, we show that each connected component of the graph is a *nice subset*.

---

**Algorithm 1** B-LATTICE (Blocked Latent bAndiTs via maTrIx ComplEtion )

---

**Require:** Phase index $\ell$, List of disjoint nice subsets of users $\mathcal{M}^{(\ell)}$, list of corresponding subsets of active items $\mathcal{N}^{(\ell)}$, clusters $\mathsf{C}$, rounds $\mathsf{T}$, round index $t_0$, exploit rounds $t_{\text{exploit}}$, estimate $\widetilde{\mathbf{P}}$ of $\mathbf{P}$, entry-wise error guarantee $\epsilon_\ell$ of $\widetilde{\mathbf{P}}$ restricted to users in $\mathcal{M}^{(\ell)}$ and all items in $\mathcal{N}^{(\ell)}$, count matrices $\mathbf{K}, \mathbf{L} \in \mathbb{N}^{\mathsf{M} \times \mathsf{N}}$. // 1) $\mathbf{K}, \mathbf{L}$ are global variables 2) $\mathcal{M}^{(\ell)}$ is a list of disjoint subset of users, each of which are proved to be nice w.h.p.

1: **for** $i^{\text{th}}$ nice subset of users $\mathcal{M}^{(\ell,i)} \in \mathcal{M}^{(\ell)}$ with active items $\mathcal{N}^{(\ell,i)}$ ($i^{\text{th}}$ set in list $\mathcal{N}^{(\ell)}$) **do**
2:    Set $t = t_0$. Set $\epsilon_{\ell+1} = \epsilon_\ell/2$, $\Delta_\ell = \epsilon_\ell/88\mathsf{C}$ if $\ell > 1$ and $\Delta_\ell = ||\mathbf{P}||_\infty$ otherwise. Set $\Delta_{\ell+1} = \epsilon_{\ell+1}/88\mathsf{C}$. // Beginning of a phase
3:    Run *exploit* component for users in $\mathcal{M}^{(\ell,i)}$ with active items $\mathcal{N}^{(\ell,i)}$. Obtain updated active set of items, round index and exploit rounds $\mathcal{N}^{(\ell,i)}, t, t_{\text{exploit}} \leftarrow$ Exploit$(\mathcal{M}^{(\ell,i)}, \mathcal{N}^{(\ell,i)}, t, t_{\text{exploit}}, \widetilde{\mathbf{P}}_{\mathcal{M}^{(\ell,i)}, \mathcal{N}^{(\ell,i)}}, \Delta_\ell)$. //Recommend common set of identified golden items
4:    Set $d_1 = \max(|\mathcal{M}^{(\ell,i)}|, |\mathcal{N}^{(\ell,i)}|)$, $d_2 = \min(|\mathcal{M}^{(\ell,i)}|, |\mathcal{N}^{(\ell,i)}|)$ and $p = c\left(\frac{\log d_1}{\Delta_{\ell+1}^2 d_2}\right)$ for some appropriate fixed constant $c > 0$.
5:    **if** $|\mathcal{N}^{(\ell,i)}| \geq \mathsf{T}^{1/3}$ and $p < 1$ **then**
6:       Run *explore* component for users in $\mathcal{M}^{(\ell,i)}$ with active items $\mathcal{N}^{(\ell,i)}$. Update estimate and round index $\widetilde{\mathbf{P}}, t \leftarrow$ Explore$(\mathcal{M}^{(\ell,i)}, \mathcal{N}^{(\ell,i)}, t, p)$ such that $\left\|\widetilde{\mathbf{P}}_{\mathcal{M}^{(\ell,i)}, \mathcal{N}^{(\ell,i)}} - \mathbf{P}_{\mathcal{M}^{(\ell,i)}, \mathcal{N}^{(\ell,i)}}\right\|_\infty \leq \Delta_{\ell+1}$ w.h.p.

       //Random Exploration and estimation of relevant reward sub-matrix
7:       For every user $u \in \mathcal{M}^{(\ell,i)}$, compute $\mathcal{T}_u^{(\ell)} \equiv \{j \in \mathcal{N}^{(\ell,i)} \mid \widetilde{\mathbf{P}}_{u\pi_u(\mathsf{TB}^{-1} - t_{\text{exploit}}\mathsf{B}^{-1})} - \widetilde{\mathbf{P}}_{uj} \leq 2\Delta_{\ell+1}\}$.
8:       Construct graph $\mathcal{G}^{(\ell,i)}$ whose nodes are users in $\mathcal{M}^{(\ell,i)}$ and an edge exists between two users $u, v \in \mathcal{M}^{(\ell,i)}$ if $\left|\widetilde{\mathbf{P}}_{ux}^{(\ell)} - \widetilde{\mathbf{P}}_{vx}^{(\ell)}\right| \leq 2\Delta_{\ell+1}$ for all items $x \in \mathcal{N}^{(\ell,i)}$. // Construct graph encoding user similarity
9:       Initialize lists $\mathcal{M}_i^{(\ell+1)} = []$ and $\mathcal{N}_i^{(\ell+1)} = []$.
10:      For each connected component $\mathcal{M}^{(\ell,i,j)}$ ($\cup_j \mathcal{M}^{(\ell,i,j)} \equiv \mathcal{M}^{(\ell,i)}$), compute $\mathcal{N}^{(\ell,i,j)} \equiv \cup_{u \in \mathcal{M}^{(\ell,i,j)}} \mathcal{T}_u^{(\ell)}$. Append $\mathcal{M}^{(\ell,i,j)}$ into $\mathcal{M}_i^{(\ell+1)}$ and $\mathcal{N}^{(\ell,i,j)}$ into $\mathcal{N}_i^{(\ell+1)}$. //Construct finer clusters and identify joint good items
11:      Invoke B-LATTICE$(\ell+1, \mathcal{M}_i^{(\ell+1)}, \mathcal{N}_i^{(\ell+1)}, \mathsf{C}, \mathsf{T}, \sigma^2, ||\mathbf{P}||_\infty, \mu, t, t_{\text{exploit}}, \widetilde{\mathbf{P}}, \epsilon_{\ell+1}, \mathbf{K}, \mathbf{L})$. // Recurse for new list of user groups and corresponding surviving items
12:   **else**
13:      For each user $u \in \mathcal{M}^{(\ell,i)}$, recommend unblocked items in $\mathcal{N}^{(\ell,i)}$ until end of rounds.
14:   **end if**
15: **end for**

---

**Exploitation (Alg. 3).** The main goal in the Exploit sub-routine of our algorithm is to identify common golden items for a group of users jointly. Consider a group of users $\mathcal{M}$ with active items $\mathcal{N}$ at the beginning of *exploit* sub-routine invoked in the $\ell^{\text{th}}$ phase of Algorithm 1. We perform the following test in Line 1 of the Exploit sub-routine: for every user $u \in \mathcal{M}$, we check if $\widetilde{\mathbf{P}}_{u\widetilde{\pi}_u(1)|\mathcal{N}} - \widetilde{\mathbf{P}}_{u\widetilde{\pi}_u(|\mathcal{N}|)|\mathcal{N}} \geq c'2^{-\ell}$ for some constant $c' > 0$ - that is if the estimated highest rewarding and lowest rewarding items of the user $u$ have a significant gap. For all the users that satisfy the above property, we take a union of the items close to the estimated highest rewarding item for each of them (Line 3). We can show that these identified items are actually golden items for all users in the nice subset $\mathcal{M}$. Hence, we immediately recommend these identified golden items to every user in the nice subset B times. In case a golden item is blocked for a user, we recommend an unblocked active item (Line 5). Subsequently we remove the golden items from the active set of items (Line 8) and prune it. We go on repeating the process with the pruned set of items until we can find no user that satisfy the above gap property between highest and lowest estimated rewarding items in the active set.

To summarize, in each phase of Algorithm 1, we perform the exploration, exploitation and user clustering steps described above. As the algorithm proceeds, we get more fine grained clustering of users, and more accurate estimates of the rewards of active items. Using this information, the algorithm tries to identify golden items and recommends the identified golden items to users.

**Implementation Details:** The actual implementation of the algorithm described above is a bit more involved due to the fact that every user has to be recommended an item at every time step (see

---

**Algorithm 2** Explore (Explore Component of a phase)

---

**Require:** Phase index $\ell$, nice subset of users $\mathcal{M}$, active items $\mathcal{N}$, round index $t_0$, sampling probability $p$. //Takes a particular set of users (nice w.h.p.), their corresponding set of active items and returns an estimate of corresponding reward sub-matrix. Unblocked items are (almost) randomly recommended to obtain data - recommendations are kept track of in global variables $\mathbf{K}, \mathbf{L}$. Data from previous phases are not used to maintain independence.

1: For each $(i, j) \in \mathcal{M} \times \mathcal{N}$, independently set $\delta_{ij} = 1$ with probability $p$ and $\delta_{ij} = 0$ with probability $1 - p$.

2: Denote $\Omega = \{(i, j) \in \mathcal{M} \times \mathcal{N} \mid \delta_{ij} = 1\}$ and $m = \max_{i \in \mathcal{M}} |j \in \mathcal{N} \mid (i, j) \in \Omega|$ to be the maximum number of index tuples in a particular row. Initialize observations corresponding to indices in $\Omega$ to be $\mathcal{A} = \phi$. Set $m_u = |j \in \mathcal{N} \mid (u, j) \in \Omega|$
// Create Bernoulli Mask $\Omega$ with the idea of recommending all items in $\Omega$

3: **for** each user $u \in \mathcal{M}$ **do**

4:    **for** rounds $t = t_0 + 1, t_0 + 2, \ldots, t_0 + m_u$ **do**

5:       Find an item $z$ in $\{j \in \mathcal{N} \mid (u, j) \in \Omega, \delta_{uj} = 1\}$. Set $\delta_{uj} = 0$.
// Find item in $\Omega$ for the user that has not been recommended yet in this function instantiation

6:       If $\mathbf{K}_{uz} + \mathbf{L}_{uz} < \mathsf{B}$ ($z$ is unblocked), set $\rho_u(t) = z$ and recommend $z$ to user $u$. Observe $\mathbf{R}^{(t)}_{u\rho_u(t)}$ and update $\mathcal{A} = \mathcal{A} \cup \{\mathbf{R}^{(t)}_{u\rho_u(t)}\}$, $\mathbf{K}_{uz} \leftarrow \mathbf{K}_{uz} + 1$. // Recommend unblocked item in $\Omega$

7:       If $\mathbf{K}_{uz} + \mathbf{L}_{uz} = \mathsf{B}$ ($z$ is blocked), recommend any unblocked item $\rho_u(t)$ in $\mathcal{N}$ s.t. $(u, \rho_u(t)) \notin \Omega$. Update $\mathbf{L}_{u\rho_u(t)} \leftarrow \mathbf{L}_{u\rho_u(t)} + 1$. Set $\mathcal{A} = \mathcal{A} \cup \{\mathbf{R}^{(t')}_{u\rho_u(t')}\}$ where $t' < t$ is the last round when $\rho_u(t') = z$ was recommended to user $u$ but the observation $\mathbf{R}^{(t')}_{uz}$ has not been used in an invocation of the function Estimate. Update $\mathbf{K}_{uz} \leftarrow \mathbf{K}_{uz} + 1$ and $\mathbf{L}_{uz} \leftarrow \mathbf{L}_{uz} - 1$.
//Found item is blocked but some observation can be re-used

8:    **end for**

9:    **for** rounds $t = t_0 + m_u + 1, \ldots, t_0 + m$ **do**

10:       Recommend any unblocked item $\rho_u(t)$ in $\mathcal{N}$ s.t. $(u, \rho_u(t)) \notin \Omega$. Update $\mathbf{L}_{u\rho_u(t)} \leftarrow \mathbf{L}_{u\rho_u(t)} + 1$.
// No available items in $\Omega$ to recommend. Recommend some other item

11:    **end for**

12: **end for**

13: Compute the estimate $\widetilde{\mathbf{P}} = \texttt{Estimate}(\mathcal{M}, \mathcal{N}, \sigma^2, \mathsf{C}, \Omega, \mathcal{A})$ and return $\widetilde{\mathbf{P}}, t_0 + m$.
// Low Rank Matrix Completion of Reward Sub-matrix $\mathbf{P}_{\mathcal{M}, \mathcal{N}}$ using observations in $\Omega$

---

problem setting in Section 2). To see this, consider the following scenario. Suppose, we identified item $j$ as a golden item for users in the nice subset $\mathcal{M}$ (crude cluster) in the Exploit sub-routine invoked in some phase of Alg. 1. Moreover, suppose there are two users $u_1, u_2$ in the cluster for whom the item has been recommended $n_1, n_2$ times respectively, for some $n_1 < n_2$. So, for the final $n_2 - n_1$ iterations during which the algorithm recommends item $j$ to $u_1$, it has to recommend some other item to $u_2$. In our algorithm, we randomly recommend an item from the remaining active set of items for $u_2$ during those $n_2 - n_1$ rounds. We store the rewards from these recommendations and use them in the exploration component of future phases. A similar phenomenon also occurs in the *Explore* sub-routine where we sometimes need to recommend items outside the sub-sampled set of entries in $\Omega$ (Line 7 and 10 in Alg. 2) To this end, we introduce matrices $\mathbf{K}, \mathbf{L} \in \mathbb{R}^{\mathsf{M} \times \mathsf{N}}$ which perform the necessary book-keeping for us.

$\mathbf{K}$ tracks number of times an item has been recommended to a user and the corresponding observation has been already used in computing an estimate of some reward sub-matrix (Line 2 of Alg. 2). Since these estimates are used to cluster users and eliminate items (Lines 7-10 in Alg. 1), these observations are not reused in subsequent phases to avoid statistical dependencies. $\mathbf{L}$ tracks the number of times an item has been recommended to a user and the corresponding observation has not been used in computing an estimate of reward sub-matrix so far. These observations can still be used once in Line 2 of Alg. 2. Observe that $\mathbf{K}_{ij} + \mathbf{L}_{ij}$ is the total times user $i$ has been recommended item $j$. In practice, eliminating observations is unnecessary and we can reuse observations whenever required. Hence Alg. 1 can work even when $\mathsf{B} = 1$ (see Alg. 6 for a simplified and practical version).

**Handling very few active items (Line 5 in Alg. 1).** Recall, in the exploration step of phase $\ell$, we randomly recommend active items with probability $p = O\left(\frac{2^{2\ell} \log d_1}{d_2}\right)$. If $p > 1$, then we simply recommend all the remaining unblocked items for each user until the end. In our analysis, we can

---

**Algorithm 3** Exploit (Exploit Component of a phase)

---

**Require:** Phase index $\ell$, nice subset of users $\mathcal{M}$, active items $\mathcal{N}$, round index $t_0$, exploit rounds $t_{\text{exploit}}$, estimate $\widetilde{\mathbf{P}}$ of $\mathbf{P}$ and error guarantee $\Delta_\ell$ such that $\left\|\widetilde{\mathbf{P}}_{\mathcal{M},\mathcal{N}} - \mathbf{P}_{\mathcal{M},\mathcal{N}}\right\|_\infty \leq 88\mathsf{C}\Delta_\ell$ with high probability. `//Takes a particular set of users (nice w.h.p.), their corresponding set of active items and an estimate of corresponding reward sub-matrix as input. Identifies common golden items for all aforementioned users in this module and recommends them jointly until exhausted - recommendations are kept track of in global variables` $\mathbf{K}, \mathbf{L}$ `and active items are pruned.`

1: **while** there exists $u \in \mathcal{M}$ such that $\widetilde{\mathbf{P}}_{u\widetilde{\pi}_u(1)|\mathcal{N}} - \widetilde{\mathbf{P}}_{u\widetilde{\pi}_u(|\mathcal{N}|)|\mathcal{N}} \geq 64\mathsf{C}\Delta_\ell\}$ **do**
2:     Compute $\mathcal{R}_u = \{j \in \mathcal{N} \mid \widetilde{\mathbf{P}}_{uj} \geq \widetilde{\mathbf{P}}_{u\widetilde{\pi}_u(1)|\mathcal{N}} - 2\Delta_{\ell+1}\}$ for every user $u \in \mathcal{M}$. Compute $\mathcal{S} = \cup_{u \in \mathcal{M}} \mathcal{R}_u$. `// Find common set of golden items for all users in` $\mathcal{M}$
3:     **for** rounds $t = t_0 + 1, t_0 + 2, \ldots, t_0 + |\mathcal{S}|\mathsf{B}$ **do**
4:         **for** each user $u \in \mathcal{M}$ **do**
5:             Denote by $x$ the $\lceil (t - t_0/\mathsf{B}) \rceil$ item in $\mathcal{S}$. If $\mathbf{K}_{ux} + \mathbf{L}_{ux} < \mathsf{B}$ ($x$ is unblocked), then recommend $x$ to user $u$ and update $\mathbf{L}_{ux} \leftarrow \mathbf{L}_{ux} + 1$. If $\mathbf{K}_{ux} + \mathbf{L}_{ux} = \mathsf{B}$ ($x$ is blocked), recommend any unblocked item $y$ in $\mathcal{N}$ (i.e $\mathbf{K}_{uy} + \mathbf{L}_{uy} < \mathsf{B}$) for the user $u$ and update $\mathbf{L}_{uy} \leftarrow \mathbf{L}_{uy} + 1$. `// Recommend all identified golden items`
6:         **end for**
7:     **end for**
8:     Update $\mathcal{N} \leftarrow \mathcal{N} \setminus \mathcal{S}$. `// Remove golden items and prune active items` $\mathcal{N}$
9:     Update $t_0 \leftarrow t_0 + |\mathcal{S}|\mathsf{B}$ and $t_{\text{exploit}} \leftarrow t_{\text{exploit}} + |\mathcal{S}|\mathsf{B}$.
10: **end while**
11: Return $\mathcal{N}, t_0, t_{\text{exploit}}$.

---

show that this happens only if the size of surviving items is too small, and when the number of remaining rounds is very small. Hence the regret is small if we follow this approach (Lemma 5). Similarly, when remaining rounds become smaller than $\mathsf{T}^{1/3}$, we follow the same approach.

**Running Time:** Computationally speaking, the main bottleneck of our algorithm is the matrix completion function *Estimate* invoked in Line 13 of the Explore Component (Algorithm 2). All the remaining steps have lower order run-times. Note that the *Estimate* function is invoked at most $O(\mathsf{C} \log \mathsf{T})$ times since there are can be at most $\mathsf{C}$ disjoint nice subsets of users at a time and the number of phases is $\log \mathsf{T}$. Moreover, note that the *Estimate* function (Algorithm 4) solves a convex objective in Line 3 - this has a time complexity of $O(\mathsf{M}^2\mathsf{N} + \mathsf{N}^2\mathsf{M})$ which is slightly limiting because of the quadratic dependence. However, a number of highly efficient techniques have been proposed for optimizing the aforementioned objective even when $\mathsf{M}, \mathsf{N}$ are in the order of millions (see [14]).

**Proof Sketch of Theorem 1** We condition on the high probability event that the low rank matrix completion estimation step is always successful (Lemma 17). Note that for a fixed user, the items chosen for recommendation in the *exploit* component of any phase are *golden items* and costs zero regret if they are unblocked and recommended. Even if it is blocked, we show a swapping argument to a similar effect. That is, with an appropriate permutation of the recommended items, we can ignore the regret incurred from golden-items altogether. Moreover, in the *explore* component of the $\ell^{\text{th}}$ phase, we can bound the sub-optimality gap of the surviving active items by some pre-determined $\epsilon_\ell$. We prove that this holds even with the (chosen) permutation of the recommended items (Lemma 18). We choose $\epsilon_\ell$ to be exponentially decreasing in $\ell$ and the number of rounds in the *explore* component of phase $\ell$ is roughly $1/\epsilon_\ell^2$. Putting these together, we obtain the regret guarantee in Theorem 1.

## 5 Conclusion and Future Work

We study the problem of Collaborative Bandits in the setting where each item can be recommended to a user a small number of times. Under some standard assumptions and a blocking constraint of $\Theta(\log \mathsf{T})$, we show a phased algorithm B-LATTICE with regret guarantees that match the tight results with no blocking constraint [25]. To the best of our knowledge, this is the first regret guarantee for such a general problem with no assumption of item clusters. We also provide novel regret lower bounds that match the upper bound in several regimes. Relaxing the assumptions, extending guarantees to a blocking constraint of $\mathsf{B} = 1$ and tightening the gap between the regret upper and lower bounds are very interesting directions for future work.

## Acknowledgements

We would like to thank Sandeep Juneja for helpful discussions on understanding the information theoretic limits of the problem.

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

**Limitations:** The main contributions of our works are theoretical. From a theoretical point of view, the limitations of our paper are discussed in Sections 3 and 5. In particular, we believe that tightening the gap between the upper and lower bounds in regret will require novel and non-trivial algorithmic ideas - we leave this as an important direction of future work.

**Broader Impact:** Due to the theoretical nature of the work, we do not foresee any adverse societal impact.

## A  Low rank matrix completion

Below, we describe the `Estimate` sub-routine to estimate a $M \times N$ low rank matrix $\mathbf{Q}$ of rank $r$ given a partially observed set of noisy entries $\{\mathbf{Z}_{ij}\}_{(i,j) \in \Omega}$ corresponding to a subset $\Omega \subseteq [M] \times [N]$. Here $\mathbb{E}\mathbf{Z}_{ij} = \mathbf{Q}_{ij}$ for all $(i,j) \in \Omega$ and furthermore, $\{\mathbf{Z}_{ij}\}_{(i,j) \in \Omega}$ are independent sub-gaussian random variables with variance proxy at most $\sigma^2$.

---

**Algorithm 4** `Estimate` (Low-rank matrix completion ) [17]

---

**Require:** Matrix dimensions $(M, N)$, noise variance $\sigma^2$, rank $r$, subset of indices that are observed $\Omega \subseteq [M] \times [N]$ and noisy observations $\{\mathbf{Z}_{ij}\}_{(i,j) \in \Omega}$.
1: Partition the rectangular matrix into square matrices. Without loss of generality, assume $M \leq N$. For each $i \in [N]$, randomly set $\zeta_i$ to be a value in the set $[\lceil N/M \rceil]$ uniformly at random. Partition indices in $[N]$ into $[N]^{(1)}, [N]^{(2)}, \ldots, [N]^{(k)}$ where $k = \lceil N/M \rceil$ and $[N]^{(q)} = \{i \in [N] \mid \zeta_i = q\}$ for each $q \in [k]$. Set $\Omega^{(q)} \leftarrow \Omega \cap ([M] \times [N]^{(q)})$ for all $q \in [k]$.       {*If* $M \geq N$, *we partition the indices in* $[M]$.}
2: **for** $q \in [k]$ **do**
3:     Solve the following convex program with $\lambda = C_\lambda \sigma \sqrt{|\Omega| / \max(M, N)}$, for some constant $C_\lambda > 0$

$$\min_{\widetilde{\mathbf{Q}}^{(q)} \in \mathbb{R}^{M \times |[N]^{(q)}|}} \sum_{(i,j) \in \Omega^{(q)}} \frac{(\widetilde{\mathbf{Q}}^{(q)}_{i\gamma_u(j)} - \mathbf{Z}_{ij})^2}{2} + \lambda \|\widetilde{\mathbf{Q}}^{(q)}\|_\star$$

    where $\|\widetilde{\mathbf{Q}}^{(q)}\|_\star$ denotes nuclear norm of matrix $\widetilde{\mathbf{Q}}^{(q)}$ and $\gamma_u(j)$ is index of $j$ in set $[N]^{(q)}$.
4: **end for**
5: Return $\widetilde{\mathbf{Q}} \in \mathbb{R}^{M \times N}$ such that $\widetilde{\mathbf{Q}}_{[M],[N]^{(q)}} = \widetilde{\mathbf{Q}}^{(q)}$ for all $q \in [k]$.

---

## B  Explore-Then-Commit (ETC)

We first present a greedy algorithm in the blocked setting with $B = 1$ (no repetition) that uses the Explore-Then-Commit (ETC) framework. Such an algorithm has two disjoint phases - *exploration and exploitation*. We will first jointly explore the set of items (without repeating same item for any user) for all users for a certain number of rounds and compute an estimate $\widetilde{\mathbf{P}}$ of the reward matrix $\mathbf{P}$. Subsequently, in the exploitation phase, for each user, we recommend the best estimated distinct items (that have not been recommended in the exploration phase to that user) inferred from the estimated reward matrix $\widetilde{\mathbf{P}}$. Note that if we explore too less, then our estimate will be poor and hence we will suffer large regret once we commit in the exploitation phase. On the other hand, if we explore too much, then the exploration cost will be high. Our goal is to balance both the exploration length and the exploitation cost under the blocked setting. Thus, we obtain the following result:

**Theorem 3.** *Consider the GBB setting with* $M$ *users,* $C = O(1)$ *clusters,* $N$ *items,* $T$ *recommendation rounds and blocking constraint* $B = 1$. *Set* $d_2 = \min(M, N)$. *Let* $\mathbf{R}^{(t)}_{u\rho_u(t)}$ *be the reward in each round, defined as in* (1). *Suppose* $d_2 = \Omega(\mu r \log(r d_2))$. *Let* $\mathbf{P} \in \mathbb{R}^{M \times N}$ *be the expected reward matrix that satisfies the conditions stated in Lemma 1 , and let* $\sigma^2$ *be the noise variance in rewards. Then, Algorithm 5, applied to the online rank-r matrix completion problem under the blocked setting guarantees the regret defined as in eq. 2 to be:*

$$\mathsf{Reg}(T) = \widetilde{O}\Big(\mu T^{2/3} \|\mathbf{P}\|_\infty^{1/3} \max\Big(1, \frac{N}{M}\Big)^{1/3} + \mu^2 \|\mathbf{P}\|_\infty \max\Big(1, \frac{N}{M}\Big) + \|\mathbf{P}\|_\infty T^{-2}\Big). \quad (6)$$

In order to understand the result, note that the second term in the regret bound stems from the fact that in our algorithmic framework, the low rank matrix completion module needs a certain number of

**Algorithm 5** ETC (Explore-Then-Commit Algorithm with Blocking constraint $\mathsf{B} = 1$)

**Require:** users $\mathsf{M}$, items $\mathsf{N}$, rounds $\mathsf{T}$, noise $\sigma^2$, rank $r$ of $\mathbf{P}$.

1: Set $p = \widetilde{O}\Big((\mathsf{N}\|\mathbf{P}\|_\infty)^{-2/3}\Big(\frac{\mathsf{T}\sigma r}{\sqrt{d_2}}\sqrt{\mu^3}\Big)^{2/3} \vee \frac{\mu^2}{d_2}\Big)$. Set $d_2 = \min(\mathsf{M}, \mathsf{N})$ and $\lambda = C\sigma\sqrt{d_2 p}$ for constant $C$.

2: For each tuple of indices $(i, j) \in [\mathsf{M}] \times [\mathsf{N}]$, independently set $\delta_{ij} = 1$ with probability $p$ and $\delta_{ij} = 0$ with probability $1 - p$.

3: Denote $\Omega = \{(i, j) \in [\mathsf{M}] \times [\mathsf{N}] \mid \delta_{ij} = 1\}$ and $m = \max_{i \in [\mathsf{M}]} |\{j \in [\mathsf{N}] \mid (i, j) \in \Omega\}|$ to be the maximum number of index tuples in a particular row. For all $(i, j) \in \Omega$, set $\mathsf{Mask}_{ij} = 0$.

4: **for** rounds $t = 1, 2, \dots, m$ **do**

5:      For each user $u \in \mathcal{U}$, recommend an item $\rho_u(t)$ in $\{j \in [\mathsf{N}] \mid (u, j) \in \Omega, \mathsf{Mask}_{uj} = 0\}$ and set $\mathsf{Mask}_{u\rho_u(t)} = 1$. If not possible then recommend any item $\rho_u(t)$ in $[\mathsf{N}]$ s.t. $(u, \rho_u(t)) \notin \Omega$ and has not been recommended yet to user $u$. Observe $\mathbf{R}^{(t)}_{u\rho_u(t)}$.

6: **end for**

7: Compute the estimate $\widetilde{\mathbf{P}} \in \mathbb{R}^{\mathsf{M} \times \mathsf{N}}$ as output of $\widetilde{\mathbf{P}} \leftarrow \texttt{Estimate}([\mathsf{M}], [\mathsf{N}], \sigma^2, r, \Omega, \{\mathbf{R}^{(t)}_{u\rho_u(t)}\}_{t \in [m]})$.

8: **for** each of remaining rounds **do**

9:      Set $j'_u = \mathsf{argmax}_{j \in [\mathsf{N}]} \widetilde{\mathbf{P}}_{u\widetilde{\pi}_u(j)}$ for each user $u$, s.t. $\widetilde{\pi}_u(j'_u)$ has not been recommended before to $u$.

10:      For each user $u \in [\mathsf{M}]$, recommend the item $\widetilde{\pi}_u(j'_u)$ to user $u$.

11: **end for**

---

observed indices and therefore a certain number of exploration rounds (note that $p \geq C\mu^2 d_2^{-1} \log^3 d_2$ for some constant $C > 0$ in Lemma 1). Similarly, the third term stems from the failure of the estimation module; again, the term $\mathsf{T}^{-2}$ can be replaced by $\mathsf{T}^{-c}$ for any constant $c > 0$. The first term in the regret bound captures the dependence on the number of rounds $\mathsf{T}$ - the scaling of $\mathsf{T}^{2/3}$ is sub-optimal and our subsequent goal is to improve this dependence to the rate of $\sqrt{\mathsf{T}}$.

*Proof of Theorem 3.* Suppose we explore for a period of $\mathsf{S}$ rounds such that the exploration period succeeds with a probability of $1 - \nu$. Conditioned on the event that the exploration period succeeds, we obtain an estimate $\widehat{\mathbf{P}}$ of the reward matrix $\mathbf{P}$ satisfying $\|\mathbf{P} - \widehat{\mathbf{P}}\|_\infty \leq \rho$. Recall $\pi_u : [\mathsf{N}] \to [\mathsf{N}]$ to be the permutation on $[\mathsf{N}]$ such that for any $i, j \in [\mathsf{N}]; i < j$, we have $\mathbf{P}_{u\pi_u(i)} \geq \mathbf{P}_{u\pi_u(j)}$. Similarly, denote $\widetilde{\pi}_u : [\mathsf{N}] \to [\mathsf{N}]$ such that for for any $i, j \in [\mathsf{N}]; i < j$, we have $\widetilde{\mathbf{P}}_{u\widetilde{\pi}_u(i)} \geq \widetilde{\mathbf{P}}_{u\widetilde{\pi}_u(j)}$. Now consider any index $i \in [\mathsf{T}]$ for which we will analyze $\mathbf{P}_{u\widetilde{\pi}_u(i)} - \mathbf{P}_{u\pi_u(i)}$ which is the error if we choose the $i^{\text{th}}$ item according to the estimated matrix $\widetilde{\mathbf{P}}$ (instead of $\mathbf{P}$). There are several cases that we need to consider. First, suppose $\widetilde{\pi}_u(i) = \pi_u(j)$ where $j \leq i$. In that case, we have $\mathbf{P}_{u\widetilde{\pi}_u(i)} - \mathbf{P}_{u\pi_u(i)} \geq 0$. Now, consider the other case where $j > i$ implying that the element in the $j^{\text{th}}$ position in the permutation $\pi_u$ has shifted to the left in $\widetilde{\pi}_u$. In order for this to happen, there must exist an element $i_1 \leq i \leq i_2$ for which $\widetilde{\pi}_u(i_2) = \pi_u(i_1)$ implying that an element $i_1$ in the permutation $\pi_u$ has shifted to the right in $\widetilde{\pi}_u$. Therefore,

$$\mathbf{P}_{u\widetilde{\pi}_u(i)} - \mathbf{P}_{u\pi_u(i)} = \mathbf{P}_{u\widetilde{\pi}_u(i)} - \widetilde{\mathbf{P}}_{u\widetilde{\pi}_u(i)} + \widetilde{\mathbf{P}}_{u\widetilde{\pi}_u(i)} - \widetilde{\mathbf{P}}_{u\widetilde{\pi}_u(i_2)}$$
$$+ \widetilde{\mathbf{P}}_{u\widetilde{\pi}_u(i_2)} - \mathbf{P}_{u\widetilde{\pi}_u(i_2)} + \mathbf{P}_{u\widetilde{\pi}_u(i_2)} - \mathbf{P}_{u\pi_u(i)} \geq -2\rho$$

where we used the fact that $\|\mathbf{P} - \widehat{\mathbf{P}}\|_\infty \leq \rho$, $\widetilde{\mathbf{P}}_{u\widetilde{\pi}_u(i)} - \widetilde{\mathbf{P}}_{u\widetilde{\pi}_u(i_2)} \geq 0$ (since $i \leq i_2$), $\mathbf{P}_{u\widetilde{\pi}_u(i_2)} - \mathbf{P}_{u\pi_u(i)} = \mathbf{P}_{u\pi_u(i_1)} - \mathbf{P}_{u\pi_u(i)} \geq 0$ (since $i_1 \leq i$). Therefore, at each step of the exploitation stage, for each user $u$, we recommend one of the top $\mathsf{T} - \mathsf{S}$ items (as inferred from $\widehat{\mathbf{P}}$) with the highest reward that have not been recommended until that round to the user $u$; in each such step, we will suffer a regret of at most $2\rho$ if we compare with the item at the same index in $\pi_u$. As before, conditioned on the event that the exploration fails (and in the exploration stage as well), the regret at each step can be

bounded from above by $2\|\mathbf{P}\|_\infty$. In that case, we have

$$\mathrm{Reg}_\Pi(\mathsf{T}) = \frac{1}{\mathsf{M}} \sum_{u\in[\mathsf{M}]}\sum_{t\in[\mathsf{T}]}\mathbf{P}_{u\pi_u(t)} - \sum_{t\in[\mathsf{T}]}\mathbf{P}_{u\rho_u(t)} \leq \max_{u\in[\mathsf{M}]}\Big(\sum_{t\in[\mathsf{T}]}\mathbf{P}_{u\pi_u(t)} - \sum_{t\in[\mathsf{T}]}\mathbf{P}_{u\widetilde{\pi}_u(t)}\Big)$$

$$\leq 2\mathsf{S}\|\mathbf{P}\|_\infty + 2(\mathsf{T}-\mathsf{S})\rho\Pr(\text{Exploration succeeds}) + 2(\mathsf{T}-\mathsf{S})\|\mathbf{P}\|_\infty\Pr(\text{Exploration fails})$$

$$\leq 2\mathsf{S}\|\mathbf{P}\|_\infty + 2\mathsf{T}\rho + 2\mathsf{T}\delta\|\mathbf{P}\|_\infty.$$

We use Lemma 1 to set $\mathsf{S} = O\Big(\mathsf{N}p + \sqrt{\mathsf{N}p\log\mathsf{M}\delta^{-1}}\Big)$ such that $\rho = O\Big(\frac{\sigma r}{\sqrt{d_2}}\sqrt{\frac{\mu^3\log d_2}{p}}\Big)$ and $\nu = 1 - \delta - O(d_2^{-3})$. We have

$$\mathrm{Reg}(\mathsf{T}) \leq O\Big(\mathsf{N}p + \sqrt{\mathsf{N}p\log\mathsf{M}\delta^{-1}}\Big)\|\mathbf{P}\|_\infty + \mathsf{T}O\Big(\frac{\sigma r}{\sqrt{d_2}}\sqrt{\frac{\mu^3\log d_2}{p}}\Big) + \mathsf{T}(\delta + d_2^{-3})\|\mathbf{P}\|_\infty.$$

For simplicity, we ignore the logarithmic and lower order terms and attempt to minimize $\mathsf{N}p\|\mathbf{P}\|_\infty + \mathsf{T}\Big(\frac{\sigma r}{\sqrt{d_2}}\sqrt{\frac{\mu^3\log d_2}{p}}\Big)$ (*Simplified Expression*) by choosing $p = (\mathsf{N}\|\mathbf{P}\|_\infty)^{-2/3}\Big(\frac{\mathsf{T}\sigma r}{\sqrt{d_2}}\sqrt{\mu^3\log d_1}\Big)^{2/3}$, $\delta = \mathsf{T}^{-4}$. If $p \geq C\mu^2 d_2^{-1}\log^3 d_2$, then notice that $\mathsf{N}p \geq 1$ and therefore $\mathsf{N}p + \sqrt{\mathsf{N}p\log\mathsf{M}\delta^{-1}} = O(\mathsf{N}p\sqrt{\log\mathsf{M}\delta^{-1}})$. Subsequently, we have

$$\mathrm{Reg}(\mathsf{T}) = O\Big(\mathsf{T}^{2/3}(\sigma^2 r^2\|\mathbf{P}\|_\infty)^{1/3}\Big(\frac{\mu^3\mathsf{N}\log d_2}{d_2}\Big)^{1/3}\log\sqrt{\mathsf{MT}} + \|\mathbf{P}\|_\infty\mathsf{T}^{-2}\Big).$$

There exists an edge case when the value of $p$ that minimizes the *simplified expression* satisfies $p \leq C\mu^2 d_2^{-1}\log^3 d_2$. Then we can substitute $p = C\mu^2 d_2^{-1}\log^3 d_2$. In that case, the second term in the *simplified expression* will still be bounded as before. On the other hand the first term in the *simplified expression* will now be bounded by $O(\frac{\mathsf{N}\mu^2}{d_2}\log^3 d_2\log^2(\mathsf{MNT})\|\mathbf{P}\|_\infty)$. Hence, our regret will be bounded by

$$\mathrm{Reg}(\mathsf{T}) = O\Big(\mathsf{T}^{2/3}(\sigma^2 r^2\|\mathbf{P}\|_\infty)^{1/3}\Big(\frac{\mu^3\mathsf{N}\log d_2}{d_2}\Big)^{1/3}\log\sqrt{\mathsf{MT}} + \frac{\mathsf{N}\mu^2}{d_2}\log^5(\mathsf{MNT})\|\mathbf{P}\|_\infty + \|\mathbf{P}\|_\infty\mathsf{T}^{-2}\Big)$$

$$= \widetilde{O}\Big(\mu\mathsf{T}^{2/3}\|\mathbf{P}\|_\infty^{1/3}\max\Big(1,\frac{\mathsf{N}}{\mathsf{M}}\Big)^{1/3} + \mu^2\|\mathbf{P}\|_\infty\max\Big(1,\frac{\mathsf{N}}{\mathsf{M}}\Big) + \|\mathbf{P}\|_\infty\mathsf{T}^{-2}\Big).$$

$\square$

## C  Experiments

We conduct detailed synthetic experiments in order to validate the theoretical guarantees/properties of our algorithm. For this purpose we use a simplified version of B-LATTICE described in Alg. 6 (all experiments have been performed on a Google Colab instance with 12GB RAM) :

In PB-LATTICE(Alg. 6), we do not use the *exploit* component for simplicity. Intuitively, if the user is recommended a *golden item* during the *explore* component, it is a good event in any case. Furthermore, in Step 14, we use $k$-means to cluster the users. Since we obtain a significantly large embedding vector for each user in Step 13, $k-$means is quite practical.

We run PB-LATTICEon several synthetic datasets. There are two main baselines for us to consider 1) Greedy Algorithm namely Alg. 5 2) In the setting where the user $u$, on being recommended item $j$ provides a like ($+1$) with probablity $\mathbf{P}_{uj}$ and a dislike ($-1$) with probability $1 - \mathbf{P}_{uj}$, we also compare with *Collaborative-Greedy* in [4]. However, recall that [4], even in the restricted setting, only provides theoretical guarantees on the number of likeable items (items with probability of liking $> 0.5$) recommended during the course of $\mathsf{T}$ rounds.

We generate three synthetic datasets to validate our algorithms. For each of them, we take $\mathsf{M} = 150$ users, $\mathsf{N} = 150$ items and $\mathsf{T} = 60$ rounds (hence the total items recommended will be 9000). The reward matrix $\mathbf{P} = \mathbf{U}\mathbf{V}^\mathsf{T}$ ($\mathbf{U}\in\mathbb{R}^{\mathsf{M}\times\mathsf{C}}$ and $\mathbf{V}\in\mathbb{R}^{\mathsf{N}\times\mathsf{C}}$) is generated as in [25] - in the $i^\text{th}$ row of $\mathbf{U}$, the $(i\%\mathsf{C})^\text{th}$ entry is set to be 1 while the other entries are 0; the entries of $\mathbf{V}$ are sampled in the following way for the three datasets.

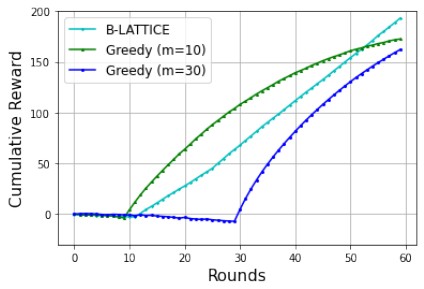 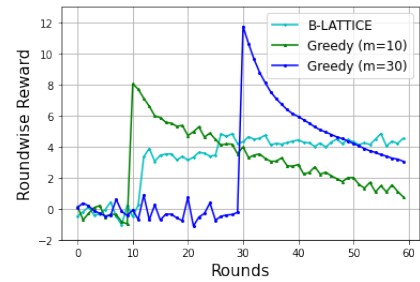

(a) $\mathbf{V}$ has entries distributed according to $\mathcal{N}(0, 25)$. We (b) $\mathbf{V}$ has entries generated according to $\mathcal{N}(0, 25)$. We observe $\mathbf{R}^{(t)}_{u\rho_u(t)} = \mathcal{N}(\mathbf{P}_{u\rho_u(t)}), 0.25)$ observe $\mathbf{R}^{(t)}_{u\rho_u(t)} = \mathcal{N}(\mathbf{P}_{u\rho_u(t)}), 0.25)$

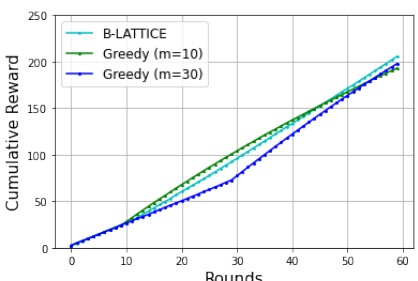 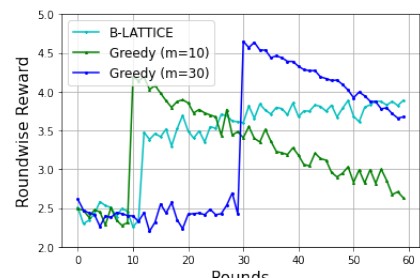

(c) $\mathbf{V}$ has entries distributed according to $\mathcal{U}(0, 5)$. We (d) $\mathbf{V}$ has entries distributed according to $\mathcal{U}(0, 5)$. We observe $\mathbf{R}^{(t)}_{u\rho_u(t)} = \mathcal{N}(\mathbf{P}_{u\rho_u(t)}), 0.25)$ observe $\mathbf{R}^{(t)}_{u\rho_u(t)} = \mathcal{N}(\mathbf{P}_{u\rho_u(t)}), 0.25)$

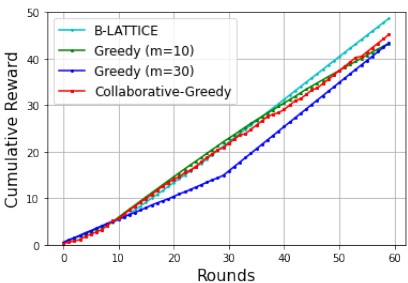 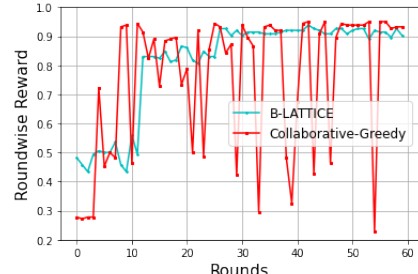

(e) $\mathbf{V}$ has entries distributed in $[0.05, 0.95]$ with equal (f) $\mathbf{V}$ has entries distributed in $[0.05, 0.95]$ with equal probability. We observe $\mathbf{R}^{(t)}_{u\rho_u(t)} = 2\mathrm{Ber}(\mathbf{P}_{u\rho_u(t)}) - 1$ probability. We observe $\mathbf{R}^{(t)}_{u\rho_u(t)} = 2\mathrm{Ber}(\mathbf{P}_{u\rho_u(t)}) - 1$

Figure 1: Cumulative Regret of the greedy algorithm (Alg. 5) and the Blocked LATTICE algorithm (simplified version in Alg. 6). In the setting where our observations are in $\{+1, -1\}$ i.e. the user likes $(+1)$ an item with probability $p$ and dislikes with probability $1 - p$, we also compare with the algorithm provided in [4] named Collaborative-Greedy. In all our settings, we have $\mathsf{M} = 150$ users, $\mathsf{N} = 150$ items, $\mathsf{C} = 4$ clusters and $\mathsf{T} = 60$ rounds. The ground truth reward matrix $\mathbf{P} = \mathbf{U}\mathbf{V}^\mathsf{T}$ is generated in the following way: each row of $\mathbf{U}$ is a standard basis vector while each entry of $\mathbf{V}$ is sampled independently from $\mathcal{N}(0, 25)$ in (a) , each entry of $\mathbf{V}$ is sampled independently from $\mathcal{U}(0, 5)$ in (b) and each entry of $\mathbf{U}$ is sampled independently from $\mathcal{U}(0, 1)$ in (c). In (a) and (b), gaussian noise with variance $0.25$ is added to the expected observation and in (c), we observe $+1$ (probability is expected reward) or $-1$. Notice that `PB-LATTICE` has 1) a small cold-start period 2) always makes good recommendations 3) better empirical rewards than other baselines.

**Algorithm 6** PB-LATTICE (Practical Blocked Latent bAndiTs via maTrIx ComplEtion with blocking constraint B = 1)

---

**Require:** Phase index $\ell$, phase length $m_\ell$, List of disjoint nice subsets of users $\mathcal{M}^{(\ell)}$, list of corresponding subsets of active items $\mathcal{N}^{(\ell)}$, clusters C, rounds T, noise $\sigma^2 > 0$, round index $t_0$, gap factor $\nu_\ell$.

1: Set $\lambda = 10\sigma\sqrt{(m_\ell/\mathsf{M})}$.
2: **for** rounds $t_0 + 1, t_0 + 2, \ldots, \min(t_0 + m, \mathsf{T})$ **do**
3:     **for** $i^{\text{th}}$ nice subset of users $\mathcal{M}^{(\ell,i)} \in \mathcal{M}^{(\ell)}$ with active items $\mathcal{N}^{(\ell,i)}$ ($i^{\text{th}}$ set in list $\mathcal{N}^{(\ell)}$) **do**
4:         Initialize $\Omega^{(i)} = \Phi$
5:         **for** user $u$ in $\mathcal{M}^{(\ell,i)}$ **do**
6:             Choose a random item $j \in \mathcal{N}^{(\ell,i)}$ for recommendation to user $u$. If $j$ is not blocked, recommend $j$ to user $u$ and observe $\mathbf{Z}_{uj}$ to be the feedback from user $u$ for item $j$. If $j$ is blocked, then recommend any $j' \in \mathcal{N}^{(\ell,i)}$ that is unblocked for user $u \in \mathcal{M}^{(\ell,i)}$. Observe $\mathbf{Z}_{uj'}$ to be the feedback from user $u$ for item $j'$.
7:             Update $\Omega^{(i)} = \Omega^{(i)} \cup \{(u, j)\}$
8:         **end for**
9:     **end for**
10: **end for**
11: **for** $i^{\text{th}}$ nice subset of users $\mathcal{M}^{(\ell,i)} \in \mathcal{M}^{(\ell)}$ with active items $\mathcal{N}^{(\ell,i)}$ ($i^{\text{th}}$ set in list $\mathcal{N}^{(\ell)}$) **do**
12:     Initialize $\mathcal{M}^{(\ell+1)} = []$ and $\mathcal{N}^{(\ell+1)} = []$.
13:     Compute $\mathbf{T} \in \mathbb{R}^{\mathsf{M} \times \mathsf{N}}$ by solving the convex program

$$\min_{\mathbf{T} \in \mathbb{R}^{\mathsf{N} \times \mathsf{M}}} \frac{1}{2} \sum_{(i,j) \in \Omega^{(i)}} \left( \mathbf{Z}_{ij} - \mathbf{T}_{ij} \right)^2 + \lambda \|\mathbf{T}_{\mathcal{M}^{(\ell,i)}, \mathcal{N}^{(\ell,i)}}\|_\star, \tag{7}$$

14:     Solve $k$-means for users in $\mathcal{M}^{(\ell,i)}$ using the vector embedding formed by the rows in $\mathbf{T}_{\mathcal{M}^{(\ell,i)}, \mathcal{N}^{(\ell,i)}}$. Choose best $k \leq \mathsf{C}$ by using ELBOW method. Denote the cluster of users by $\{\mathcal{M}^{(\ell,i,j)}\}_j$.
15:     **for** each cluster of users $\mathcal{M}^{(\ell,i,j)}$ **do**
16:         Compute set of active arms $\mathcal{N}^{(\ell,i,j)}$ as $\{s \in \mathcal{N}^{(\ell,i)} \mid \mathbf{T}_{us} \geq \mathbf{T}_{u\widetilde{\pi}_u(\mathsf{T})} - \nu_\ell$ for some $u \in \mathcal{M}^{(\ell,i,j)}\}$. Here $\widetilde{\pi}_u$ is the permutation of the surviving items in $\mathcal{N}^{(\ell,i)}$ in descending order of estimated reward for user $u$.
17:         Append $\mathcal{M}^{(\ell,i,j)}$ to $\mathcal{M}^{(\ell+1)}$ and $\mathcal{N}^{(\ell,i,j)}$ to $\mathcal{N}^{(\ell+1)}$.
18:     **end for**
19:     Compute $m_{\ell+1}, \nu_{\ell+1}$ as a function of $\ell$. Set $t = \min(t_0 + m, \mathsf{T})$.
20:     If $t < \mathsf{T}$, invoke Algorithm PB-LATTICE(phase $\ell+1$, phase length $m_{\ell+1}$, list of users $\mathcal{M}^{(\ell+1)}$, list of items $\mathcal{N}^{(\ell+1)}$, clusters C, rounds T, noise $\sigma^2$, round index $t_0 + m$, gap factor $\nu_{\ell+1}$).
21: **end for**

---

1. (*D1*:) Each entry of $\mathbf{V}$ is sampled from a gaussian distribution $\mathcal{N}(0, 25)$ with mean zero and variance 25. User $u$ on being recommended item $j$ provides a random feedback distributed according to $\mathcal{N}(\mathbf{P}_{uj}, 0.25)$.
2. (*D2*:) Each entry of $\mathbf{V}$ is sampled from a uniform distribution $\mathcal{U}(0, 5)$ with range in $[0, 5]$. User $u$ on being recommended item $j$ provides a random feedback distributed according to $\mathcal{N}(\mathbf{P}_{uj}, 0.25)$.
3. (*D3*:) Each entry of $\mathbf{V}$ is sampled from a uniform distribution $[0.05, 0.95]$ with equal probability. User $u$ on being recommended item $j$ provides a like $(+1)$ with probability $\mathbf{P}_{uj}$ and a dislike $(-1)$ with probability $1 - \mathbf{P}_{uj}$.

For the PB-LATTICEalgorithm (Alg. 6), we choose the hyper-parameters on the phase lengths and gap parameters as $m_\ell = 10 + 2\ell$ and $\nu_\ell = \|\mathbf{P}\|_\infty/(8 \cdot 2^\ell)$. Since PB-LATTICEis a recursive algorithm, we initialize PB-LATTICEwith phase index 1, phase length 12, list of users $[[\mathsf{M}]]$, list of items $[[\mathsf{N}]]$, clusters C, rounds T, noise $\sigma^2$, round index 1, gap factor $\|\mathbf{P}\|_\infty/16$. For the greedy algorithm (Alg. 5), we experiment with two exploration periods ($m = 10$ and $m = 30$). Finally, for the Collaborative-Greedy algorithm in [4], we choose $\theta = 0.5$ and $\alpha = 0.5$.

**Results and Insights:** The cumulative reward until round $t$ (defined as $M^{-1} \sum_{j=1}^{t} \sum_{u=1}^{M} \mathbf{P}_{u\rho_u(j)}$) and the round wise reward at round $t$ (defined as $M^{-1} \sum_{u=1}^{M} \mathbf{P}_{u\rho_u(t)}$) is plotted for all the three datasets D1, D2, D3. in Figures 1a, 1b, 1c, 1d, 1e and 1f. Apart from obtaining good empirical rewards, `PB-LATTICE`has several practically relevant properties which are demonstrated through our experiments:

1. Note from Figures 1b and 1d that the greedy algorithm has a large cold-start period for good exploitation properties. If the exploration is too small, then the estimation becomes poor. On the other hand, `PB-LATTICE`improves in phases - therefore, it has a small cold-start period i.e. it starts recommending relevant items very quickly and also has good estimation guarantees

2. Collaborative-greedy in [4] proposes exploration rounds throughout the entire course of $T$ rounds. This is often impractical as users will demand good recommendations throughout. This is demonstrated in Fig. 1f where the red curve (round-wise reward of Collaborative-Greedy) has several dips but the round-wise regret of `PB-LATTICE`stays high.

# D   Detailed Proof of Theorem 1

Next, we characterize some properties namely the condition number and incoherence of sub-matrices of $\mathbf{P}$ restricted to a *nice* subset of users. The following Lemmas 2 and 3 can be found in [25]

**Lemma 2.** *Suppose Assumption 1 is true. Consider a sub-matrix $\mathbf{P}_{\mathsf{sub}}$ of $\mathbf{P}$ having non-zero singular values $\lambda_1' > \cdots > \lambda_{C'}'$ (for $C' \leq C$). Then, if the rows of $\mathbf{P}_{\mathsf{sub}}$ correspond to a nice subset of users, we have $\frac{\lambda_1'}{\lambda_{C'}'} \leq \frac{\lambda_1}{\lambda_C} \sqrt{\tau}$.*

**Lemma 3.** *Suppose Assumption 1 is true. Consider a sub-matrix $\mathbf{P}_{\mathsf{sub}} \in \mathbb{R}^{\mathbf{B}' \times \mathbf{A}'}$ (with SVD decomposition $\mathbf{P}_{\mathsf{sub}} = \widetilde{\mathbf{U}}\widetilde{\boldsymbol{\Sigma}}\widetilde{\mathbf{V}}$) of $\mathbf{P}$ whose rows correspond to a nice subset of users. Then, $\left\|\widetilde{\mathbf{U}}\right\|_{2,\infty} \leq \sqrt{\frac{C\tau}{N'}}$ and $\left\|\widetilde{\mathbf{V}}\right\|_{2,\infty} \leq \sqrt{\frac{\mu C}{\alpha M'}}$.*

Lemmas 2 and 3 allow us to apply low rank matrix completion (Lemma 1) to relevant sub-matrices of the reward matrix $\mathbf{P}$.

## D.1   Main Analysis

Blocked LATTICE is run in phases consisting of *exploit* component and *explore* component indexed by $\ell = 1, 2, \dots$. Note that the *exploit* component of a phase is followed by the *explore* component. However, for the first phase, the *exploit* component has a length of zero rounds. Importantly note that any phase (say $\ell$) for distinct *nice* subsets of users $\mathcal{M}^{(\ell,i)}, \mathcal{M}^{(\ell,j)}$ will be run asynchronously as reward observations corresponding to one subset is not used to determine the policy for any other subset of users. At the beginning of the *explore* component of each phase $\ell$, we have the following set of desirable properties:

(A) We will run the *explore* component of phase $\ell$ asynchronously and separately for a list of disjoint nice subsets of users $\mathcal{M}^{(\ell)} \equiv \{\mathcal{M}^{(\ell,1)}, \dots, \mathcal{M}^{(\ell,a_\ell)}\}$ and respective sets of arms $\mathcal{N}^{(\ell)} \equiv \{\mathcal{N}^{(\ell,1)}, \dots, \mathcal{N}^{(\ell,a_\ell)}\}$ where $a_\ell \leq C$ such that $\cup_{i \in [a_\ell]}\mathcal{M}^{(\ell,i)} \subseteq [M]$, $\mathcal{M}^{(\ell,i)} \cap \mathcal{M}^{(\ell,j)} = \emptyset$ for any $i, j \in [a_\ell]$ (i.e. the groups of users are *nice* subsets of $[M]$ with no overlap) and $\mathcal{N}^{(\ell,i)} \subseteq [N]$ (i.e. the active set of items $\mathcal{N}^{(\ell,i)}$ for users in $\mathcal{M}^{(\ell,i)}$ are a subset of $[N]$ but the active sets can overlap). The sets $\{(\mathcal{M}^{(\ell,i)}, \mathcal{N}^{(\ell,i)}\}_{i \in [a_\ell]}$ remain unchanged during the *explore* component of phase $\ell$. As mentioned before, the round at which the *explore* component of the phase $\ell$ starts is different for each subset of *nice* users $\mathcal{M}^{(\ell,i)}$.

(B) At any round $t \in [T]$ in a particular phase $\ell$, let us denote by $\mathcal{O}_{\mathcal{M}^{(\ell,i)}}^{(\ell,t)}$ to be the set of items chosen for recommendation (see Step 5 in Alg. 3) in *exploit* components so far (from round $t = 1$) for users in $\mathcal{M}^{(\ell,i)}$. In other words, the items in $\mathcal{O}_{\mathcal{M}^{(\ell,i)}}^{(\ell,t)}$ have already been recommended to **all users** in $\mathcal{M}^{(\ell,i)}$ up to $\ell^{\text{th}}$ phase. In that case, we also maintain that

$$\mathcal{O}_{\mathcal{M}^{(\ell,i)}}^{(\ell,t)} \subseteq \{\pi_u(t)\}_{t=1}^{T/B}.$$

the best $\mathsf{T}/\mathsf{B}$ items *(golden items)* not chosen for recommendation so far in the exploit phases for users in $\mathcal{M}^{(\ell,i)}$ are contained in the set of surviving items i.e. if $\mathcal{Z} = [\mathsf{N}] \setminus \mathcal{O}^{(\ell,t)}_{\mathcal{M}^{(\ell,i)}}$, then it must happen that

$$\mathcal{N}^{(\ell,i)} \supseteq \bigcup_{u \in \mathcal{M}^{(\ell,i)}} \{\pi_u(t') \mid \mathcal{Z}\}^{\mathsf{T}/\mathsf{B} - \left|\mathcal{O}^{(\ell,t)}_{\mathcal{M}^{(\ell,i)}}\right|}_{t'=1} \text{ for all } i \in [a_\ell] \tag{8}$$

Note that eq. 8 implies that for every user $u \in \mathcal{M}^{(\ell,i)}$, there are **sufficient golden items in $\mathcal{N}^{(\ell,i)}$ to be recommended for the remaining rounds**. To see this, note that there are $\mathsf{T}/\mathsf{B}$ golden items at the beginning and no golden items which are unblocked are eliminated. Hence the number of remaining rounds at any point must be smaller than the number of possible allowed recommendations of golden items belonging to the surviving set of items.

(C) Furthermore, for all $i \in [a_\ell]$, the set $\mathcal{N}^{(\ell,i)}$ must also satisfy the following:

$$\left| \mathbf{P}_{u\pi_u(1)|\mathcal{N}^{(\ell,i)}} - \min_{j \in \mathcal{N}^{(\ell,i)}} \mathbf{P}_{uj} \right| \leq \epsilon_\ell \text{ for all } u \in \mathcal{M}^{(\ell,i)} \tag{9}$$

where $\epsilon_\ell$ is a fixed exponentially decreasing sequence in $\ell$ (in particular, we choose $\epsilon_1 = ||\mathbf{P}||_\infty$ and $\epsilon_\ell = C' 2^{-\ell} \min\left( ||\mathbf{P}||_\infty, \frac{\sigma\sqrt{\mu}}{\log \mathsf{N}} \right)$ for $\ell > 1$ for some constant $C' > 0$).

Next, at the beginning of the *exploit* component of phase $\ell + 1$, we will have the following set of desirable properties:

(a) At every round $t$ in the *exploit* component of phase $\ell + 1$, we maintain a list of disjoint *nice subsets* of users $\mathcal{M}^{(\ell+1)} \equiv \{\mathcal{M}^{(\ell+1,1)}, \ldots, \mathcal{M}^{(\ell+1,a_{\ell+1})}\}$ (where $\cup_{i \in [a_{\ell+1}]} \mathcal{M}^{(\ell+1,i)} \subseteq [\mathsf{M}]$) and corresponding sets of items $\mathcal{N}^{(\ell+1,t)} \equiv \{\mathcal{N}^{(\ell+1,t,1)}, \ldots, \mathcal{N}^{(\ell+1,t,a_{\ell+1})}\}$ where $a_{\ell+1} \leq \mathsf{C}$ and $\cup_{i \in [a_{\ell+1}]} \mathcal{N}^{(\ell+1,t,i)} \subseteq [\mathsf{N}]$. Note that in the exploit component, we also use the round index in the superscript for item subsets as they can change during the *exploit* component (unlike the *explore* component).

(b) We ensure that for any user $u \in \mathcal{M}^{(\ell+1,i)}$, the set of items chosen for recommendation in the *exploit* component of phase $\ell + 1$ belongs to the set of best $\mathsf{T}$ items i.e. $\{\pi_u(t)\}^{\mathsf{T}}_{t=1}$.

Since LATTICE is random, we will say that our algorithm is $(\epsilon_\ell, \ell)-good$ if at the beginning of the *explore* component of the $\ell^{\text{th}}$ phase the algorithm can maintain a list of users and items satisfying properties A-C. Let us also define the event $\mathcal{E}^{(\ell)}_2$ to be true if properties (A-C) are satisfied at the beginning of the *explore* component of phase $\ell$ by the phased elimination algorithm. We can show that if our algorithm is $(\epsilon_\ell, \ell)-good$ then the low rank matrix completion step (denoted by the event $\mathcal{E}^{(\ell)}_3$) in the *explore* component of phase $\ell$ is successful (i.e. the event $\mathcal{E}^{(\ell)}_3$ is true) with high probability.

Conditioned on the aforementioned two events $\mathcal{E}^{(\ell)}_2, \mathcal{E}^{(\ell)}_3$, with probability 1, during all rounds $t$ of the exploit component in phase $\ell + 1$, properties $a - b$ are satisfied and the event $\mathcal{E}^{(\ell+1)}_2$ is going to be true. We are going to prove inductively that our algorithm is $(\epsilon_\ell, \ell)$-good for all phases indexed by $\ell$ for our choice of $\{\epsilon_\ell\}_\ell$ with high probability.

**Base Case:** For $\ell = 1$ (the first phase), the number of rounds in the *exploit* component is zero and we start with the *explore* component. We initialize $\mathcal{M}^{(1,1)} = [\mathsf{N}], \mathcal{N}^{(1,1)} = [\mathsf{M}]$ and therefore, we have

$$\left| \max_{j \in \mathcal{N}^{(\ell,1)}} \mathbf{P}_{uj} - \min_{j \in \mathcal{N}^{(\ell,1)}} \mathbf{P}_{uj} \right| \leq ||\mathbf{P}||_\infty \text{ for all } u \in [\mathsf{M}].$$

Clearly, $[\mathsf{M}]$ is a *nice subset* of users and finally for every user $u \in [\mathsf{M}]$, the best $\mathsf{T}/\mathsf{B}$ items *(golden items)* $\{\pi_u(t)\}^{\mathsf{T}/\mathsf{B}}_{t=1}$ belong to the entire set of items. Thus for $\ell = 1$, conditions A-C are satisfied at the beginning of the *explore* component and therefore the event $\mathcal{E}^{(1)}_2$ is true. Hence, our initialization makes the algorithm $(||\mathbf{P}||_\infty, 1)$-good.

**Inductive Argument:** Suppose, at the beginning of the phase $\ell$, we condition on the events $\cap^\ell_{j=1} \mathcal{E}^{(j)}_2$ that Algorithm is $(\epsilon_j, j)-good$ for all $j \leq \ell$. This means that conditions (A-C) are satisfied at the beginning of the *explore* component of all phases up to and including that of $\ell$ for each

reward sub-matrix (indexed by $i \in [a_\ell]$) corresponding to the users in $\mathcal{M}^{(\ell,i)}$ and items in $\mathcal{N}^{(\ell,i)}$. Next, our goal is to run low rank matrix completion in order to estimate each of the sub-matrices corresponding to $\{(\mathcal{M}^{(\ell,i)}, \mathcal{N}^{(\ell,i)})\}_{i \in [a_\ell]}$.

**Exploration Strategy:** Consider a particular nice subset of users $\mathcal{M}^{(\ell,i)}$ and corresponding active items $\mathcal{N}^{(\ell,i)}$ (s.t. $|\mathcal{N}^{(\ell,i)}| \geq \mathsf{T}^{1/3}$) at the beginning of the *explore* component of phase $\ell$ for $\mathcal{M}^{(\ell,i)}$. For each user $u \in \mathcal{M}^{(\ell,i)}$, we are going to sample each item $j \in \mathcal{N}^{(\ell,i)}$ with probability $p$ (to be determined based on the desired error guarantee). Suppose the set of indices sampled in the explore component of phase $\ell$ is denoted by $\Omega^{(\ell)} \subseteq \mathcal{M}^{(\ell,i)} \times \mathcal{N}^{(\ell,i)}$. Now, for each user $u \in \mathcal{M}^{(\ell,i)}$ we recommend all the unblocked items in the set $\mathcal{A}_u^{(\ell)} \equiv \{j \in \mathcal{N}^{(\ell,i)} \mid (u,j) \mid \Omega^{(\ell)}\}$ and obtain the corresponding noisy reward (note that for blocked items in the aforementioned set, we have already obtained the corresponding noisy rewards).

For simplicity, we intend to complete the *explore* component at the same round for all users in $\mathcal{M}^{(\ell,i)}$. If, for two users $u, v \in \mathcal{M}^{(\ell,i)}$, it happens that recommendation of all items in $\mathcal{A}_u^{(\ell)}$ is complete for user $u$ but recommendation of all items in $\mathcal{A}_v^{(\ell)}$ is incomplete for user $v$, then for the remaining rounds we recommend to user $u$ arbitrary unblocked items from $\mathcal{N}^{(\ell,i)}$. Note that this is always possible since the set $\mathcal{N}^{(\ell,i)}$ has sufficiently many unblocked items allowing recommendations in the remaining rounds at beginning of explore component of phase $\ell$. We start with the following lemma to characterize the round complexity of estimating the sub-matrix $\mathbf{P}_{\mathcal{M}^{(\ell,i)}, \mathcal{N}^{(\ell,i)}}$ up to entry-wise error $\Delta_{\ell+1}$ with high probability using the noisy observations corresponding to the subset of indices $\Omega^{(\ell)}$:

**Lemma 4.** *Consider a particular subset of nice users $\mathcal{M}^{(\ell,i)}$ and corresponding active items $\mathcal{N}^{(\ell,i)}$ (such that $\min\left(\mathsf{M}/(\tau\mathsf{C}), |\mathcal{N}^{(\ell,i)}|\right) \geq \mathsf{T}^{1/3}$) at the beginning of the explore component of phase $\ell$. Suppose $d_1 = \max(|\mathcal{M}^{(\ell,i)}|, |\mathcal{N}^{(\ell,i)}|)$ and $d_2 = \min(|\mathcal{M}^{(\ell,i)}|, |\mathcal{N}^{(\ell,i)}|)$. Let us fix $\Delta_{\ell+1} = \Omega(\sigma\sqrt{\mu^3 \log d_1}/\sqrt{d_2})$ and condition on the event $\mathcal{E}_2^{(\ell)}$. Suppose Assumptions 1 and 2 are satisfied. In that case, in explore component of phase $\ell$, by choosing $1 \geq p = c\left(\frac{\sigma^2 \widetilde{\mu}^3 \log d_1}{\Delta_{\ell+1}^2 d_2}\right)$ (for some constant $c > 0$) and using*

$$m_\ell = O\left(\frac{\sigma^2 \widetilde{\mu}^3 \log(\mathsf{M} \bigvee \mathsf{N})}{\Delta_{\ell+1}^2} \max\left(1, \frac{\mathsf{N}\tau}{\mathsf{M}}\right) \log \mathsf{T})\right)$$

*rounds under the blocked constraint, we can compute an estimate $\widetilde{\mathbf{P}}^{(\ell)} \in \mathbb{R}^{\mathsf{M} \times \mathsf{N}}$ such that with probability $1 - O(\mathsf{T}^{-3})$, we have*

$$\left\|\widetilde{\mathbf{P}}_{\mathcal{M}^{(\ell,i)}, \mathcal{N}^{(\ell,i)}}^{(\ell)} - \mathbf{P}_{\mathcal{M}^{(\ell,i)}, \mathcal{N}^{(\ell,i)}}\right\|_\infty \leq \Delta_{\ell+1}. \tag{10}$$

*where $\sigma^2$ is the noise variance, $\mu$ is the incoherence of reward matrix $\mathbf{P}$ and $\widetilde{\mu}$ is the incoherence factor of reward sub-matrix $\mathbf{P}_{\mathcal{M}^{(\ell,i)}, \mathcal{N}^{(\ell,i)}}$.*

*Proof of Lemma 4.* We are going to use Lemma 1 in order to compute an estimate $\widetilde{\mathbf{P}}_{\mathcal{M}^{(\ell,i)}, \mathcal{N}^{(\ell,i)}}^{(\ell)}$ of the sub-matrix $\mathbf{P}_{\mathcal{M}^{(\ell,i)}, \mathcal{N}^{(\ell,i)}}$ satisfying $\left\|\widetilde{\mathbf{P}}_{\mathcal{M}^{(\ell,i)}, \mathcal{N}^{(\ell,i)}}^{(\ell)} - \mathbf{P}_{\mathcal{M}^{(\ell,i)}, \mathcal{N}^{(\ell,i)}}\right\|_\infty \leq \Delta_{\ell+1}$. Since $\mathcal{M}^{(\ell,i)}$ is a *nice* subset of users, the cardinality of $|\mathcal{M}^{(\ell,i)}|$ must be larger than $\mathsf{M}/(\tau\mathsf{C})$. Recall that $\tau$ is the ratio of the maximum cluster size and the minimum cluster size; $\mathsf{M}/\mathsf{C}$ being the average cluster size implies that the minimum cluster size is bounded from above by $\mathsf{M}/(\tau\mathsf{C})$. From Lemma 1, we know that by using $m_\ell = O\left(p|\mathcal{N}^{(\ell,i)}| + \sqrt{|\mathcal{N}^{(\ell,i)}| p \log(|\mathcal{M}^{(\ell,i)}|\delta^{-1})}\right)$ rounds (see Lemma 1) restricted to users in $\mathcal{M}^{(\ell,i)}$ such that with probability at least $1 - (\delta + d_2^{-12})$ (see Remark 5),

$$\left\|\widetilde{\mathbf{P}}_{\mathcal{M}^{(\ell,i)}, \mathcal{N}^{(\ell,i)}}^{(\ell)} - \mathbf{P}_{\mathcal{M}^{(\ell,i)}, \mathcal{N}^{(\ell,i)}}\right\|_\infty = O\left(\frac{\sigma\sqrt{\widetilde{\mu}^3 \log d_1}}{\sqrt{pd_2}}\right).$$

where $d_1 = \max(|\mathcal{M}^{(\ell,i)}|, |\mathcal{N}^{(\ell,i)}|)$, $d_2 = \min(|\mathcal{M}^{(\ell,i)}|, |\mathcal{N}^{(\ell,i)}|)$ and $\widetilde{\mu}$ is the incoherence factor of the matrix $\mathbf{P}_{\mathcal{M}^{(\ell,i)}, \mathcal{N}^{(\ell,i)}}$. In order for the right hand side to be less than $\Delta_{\ell+1}$, we can set

$p = c\left(\frac{\sigma^2 \widetilde{\mu}^3 \log d_1}{\Delta_{\ell+1}^2 d_2}\right)$ for some appropriate constant $c > 0$. Since the event $\mathcal{E}_2^{(\ell)}$ is true, we must have that $\left|\mathcal{M}^{(\ell,i)}\right| \geq \mathsf{M}/(\tau\mathsf{C})$; hence $d_2 \geq \min\left(\mathsf{M}/(\tau\mathsf{C}), \left|\mathcal{N}^{(\ell,i)}\right|\right)$. Therefore, we must have that

$$m_\ell = O\left(\frac{\sigma^2 \widetilde{\mu}^3 \log(\mathsf{M} \vee \mathsf{N})}{\Delta_{\ell+1}^2} \max\left(1, \frac{\mathsf{N}\tau}{\mathsf{M}}\right) \log \mathsf{T})\right)$$

where we substitute $\delta^{-1} = 1/\text{poly}(\mathsf{T})$ and furthermore, the condition of the Lemma statement implies that $d_2^{-12} = O(\mathsf{T}^{-3})$. Hence, we complete the proof of the lemma.

**Remark 5** (Remark 1 in [8])**.** *The error probability can be reduced from $O(\delta + d_2^{-3})$ to $O(\delta + d_2^{-c})$ for any constant $c > 0$ with only constant factor changes in the round complexity $m$ and the estimation guarantees $\|\mathbf{P} - \widetilde{\mathbf{P}}\|_\infty$. We will use $c = 12$ in rest of the paper.*

$\square$

Note that although $\widetilde{\mu}$, the incoherence factor of the sub-matrix $\mathbf{P}_{\mathcal{M}^{(\ell,i)}, \mathcal{N}^{(\ell,i)}}$ is unknown, from Lemma 3, we know that $\widetilde{\mu}$ is bounded from above by $O(\mu)$ (recall that $C, \alpha, \tau = O(1)$).

**Exploration Strategy continued:** In particular, we choose $\Delta_{\ell+1} = \epsilon_\ell / 176\mathsf{C}$ at the beginning of the *explore* component of phase $\ell$ for the set of users $\mathcal{M}^{(\ell,i)}$ with active items $\mathcal{N}^{(\ell,i)}$. One edge case scenario is when for a particular set of *nice* users $\mathcal{M}^{(\ell,i)}$, at the beginning of the explore component of phase $\ell$, we have $\left|\mathcal{N}^{(\ell,i)}\right| \leq \mathsf{T}^{1/3}$. In this case, **this set of *nice* users $\mathcal{M}^{(\ell,i)}$ do not progress to the next phase** and in the *explore* component, we simply recommend arbitrary items in $\mathcal{N}^{(\ell,i)}$ to users in $\mathcal{M}^{(\ell,i)}$ for the remaining rounds. Another interesting edge case scenario is when $d_2 = \min(\left|\mathcal{M}^{(\ell,i)}\right|, \left|\mathcal{N}^{(\ell,i)}\right|, )$ is so small that the requisite error guarantee $\Delta_{\ell+1}$ (eq. 10) cannot be achieved even by setting $p = 1$ i.e. we recommend all the items in the active set. Recall that by our induction assumption, $\mathcal{N}^{(\ell,i)}$ is sufficiently large so that it is possible to recommend unblocked items for the number of remaining rounds (say $\mathsf{T} - t_\ell$ with $t_\ell$ being the round at which the *explore* component of phase $\ell$ starts). In that case, **the set of users $\mathcal{M}^{(\ell,i)}$ do not progress to the subsequent phase**; we simply recommend arbitrary unblocked items in the set $\mathcal{N}^{(\ell,i)}$ to the users in $\mathcal{M}^{(\ell,i)}$ for the remaining rounds. In both the above edge case scenarios, the *explore* component of phase $\ell$ for users in $\mathcal{M}^{(\ell,i)}$ would last for the remaining rounds from where it starts. We can now show the following lemma:

**Lemma 5.** *Consider a particular subset of nice users $\mathcal{M}^{(\ell,i)}$ and corresponding surviving items $\mathcal{N}^{(\ell,i)}$ at the beginning of the explore component of phase $\ell$. Suppose $d_1 = \max(\left|\mathcal{M}^{(\ell,i)}\right|, \left|\mathcal{N}^{(\ell,i)}\right|)$ and $d_2 = \min(\left|\mathcal{M}^{(\ell,i)}\right|, \left|\mathcal{N}^{(\ell,i)}\right|)$. Now $\Delta_{\ell+1} = \epsilon_\ell/176\mathsf{C}$ is such that there does not exist any $p \in [0,1]$ for which RHS in eq. 3 can be $\Delta_{\ell+1}$ for estimation of matrix $\mathbf{P}$ restricted to the rows in $\mathcal{M}^{(\ell,i)}$ and columns in $\mathcal{N}^{(\ell,i)}$. In that case, we must have for all users $u \in \mathcal{M}^{(\ell,i)}$,*

$$(\mathsf{T} - t_\ell) \max_{y \in \mathcal{N}^{(\ell,i)}} \left(\mathbf{P}_{u\pi_u(1)|\mathcal{N}^{(\ell,i)}} - \mathbf{P}_{uy}\right) = \widetilde{O}\left(\sigma\mathsf{C}\sqrt{\mu^3 \log d_1} \max\left(\sqrt{\mathsf{T}}, \mathsf{T}\sqrt{\frac{\mathsf{C}}{\mathsf{M}\tau}}\right)\right)$$

*where $t_\ell$ is the round at which the explore component of phase $\ell$ starts.*

*Proof.* Note that by our induction hypothesis, we must have $\mathsf{B}\left|\mathcal{N}^{(\ell,i)}\right| \geq \mathsf{T} - t_\ell$ because the set of surviving items must contain sufficient unblocked items (for recommendation in the remaining rounds) for every user $u \in \mathcal{M}^{(\ell,i)}$. Hence, by setting $p = 1$ in eq. 3, with a certain number of rounds, we can obtain an estimate $\mathbf{Q}$ of $\mathbf{P}$ satisfying

$$\left\|\mathbf{Q}_{\mathcal{M}^{(\ell,i)}, \mathcal{N}^{(\ell,i)}} - \mathbf{P}_{\mathcal{M}^{(\ell,i)}, \mathcal{N}^{(\ell,i)}}\right\|_\infty = O\left(\frac{\sigma\sqrt{\mu^3 \log d_1}}{\sqrt{d_2}}\right).$$

Hence, this implies that our choice of $\Delta_{\ell+1}$ satisfies $\Delta_{\ell+1} = O\left(\frac{\sigma\sqrt{\mu^3 \log d_1}}{\sqrt{d_2}}\right)$ implying that $\epsilon_\ell = O\left(\frac{\sigma\mathsf{C}\sqrt{\mu^3 \log d_1}}{\sqrt{d_2}}\right)$. Now, there are two possibilities: 1) either $d_2 = \left|\mathcal{M}^{(\ell,i)}\right|$ implying that $\mathsf{M}/(\tau\mathsf{C}) \leq$

$\left|\mathcal{M}^{(\ell,i)}\right| \leq \left|\mathcal{N}^{(\ell,i)}\right|$. In that case, we have that $\epsilon_\ell = O\left(\frac{\sigma\mathsf{C}^{1.5}\sqrt{\mu^3\log d_1}}{\sqrt{\mathsf{M}\tau}}\right)$. Now, because of our induction assumption, we will have

$$\max_{y\in\mathcal{N}^{(\ell,i)}}\left(\mathbf{P}_{u\pi_u(1)|\mathcal{N}^{(\ell,i)}} - \mathbf{P}_{uy}\right) \leq \epsilon_\ell = O\left(\frac{\sigma\mathsf{C}^{1.5}\sqrt{\mu^3\log d_1}}{\sqrt{\mathsf{M}\tau}}\right).$$

Therefore, for any user $u \in \mathcal{M}^{(\ell,i)}$, if we recommend arbitrary unblocked items in $\mathcal{N}^{(\ell,i)}$ for the remaining rounds then we can bound the following quantity

$$(\mathsf{T}-t_\ell)\max_{y\in\mathcal{N}^{(\ell,i)}}\left(\mathbf{P}_{u\pi_u(1)|\mathcal{N}^{(\ell,i)}} - \mathbf{P}_{uy}\right) = O\left(\frac{\sigma\mathsf{T}\mathsf{C}^{1.5}\sqrt{\mu^3\log d_1}}{\sqrt{\mathsf{M}\tau}}\right).$$

2) The second possibility is the following: $d_2 = \left|\mathcal{N}^{(\ell,i)}\right| \geq \mathsf{B}^{-1}(\mathsf{T}-t_\ell)$. In that case, from our induction assumption, we have that $(\mathsf{B} = \Theta(\log\mathsf{T}))$

$$\max_{y\in\mathcal{N}^{(\ell,i)}}\left(\mathbf{P}_{u\pi_u(1)|\mathcal{N}^{(\ell,i)}} - \mathbf{P}_{uy}\right) \leq \epsilon_\ell = \widetilde{O}\left(\frac{\sigma\mathsf{C}\sqrt{\mu^3\log d_1}}{\sqrt{\mathsf{T}-t_\ell}}\right).$$

and therefore

$$(\mathsf{T}-t_\ell)\max_{y\in\mathcal{N}^{(\ell,i)}}\left(\mathbf{P}_{u\pi_u(1)|\mathcal{N}^{(\ell,i)}} - \mathbf{P}_{uy}\right) \leq \epsilon_\ell(\mathsf{T}-t_\ell) = \widetilde{O}\left(\frac{\sigma\mathsf{C}\sqrt{\mu^3\log d_1}}{\sqrt{\mathsf{T}-t_\ell}}\cdot(\mathsf{T}-t_\ell)\right)$$
$$= \widetilde{O}\left(\sigma\mathsf{C}\sqrt{\mu^3\mathsf{T}\log d_1}\right).$$

$\square$

Consider $\mathcal{M}'^{(\ell)} \subseteq \mathcal{M}^{(\ell)}$ to be the family of *nice* subsets of users which do not fall into the edge case scenarios i.e. 1) there exists $0 \leq p \leq 1$ for which the theoretical bound in RHS in eq. 3 can be smaller than $\Delta_{\ell+1}$ 2) we have $\left|\mathcal{N}^{(\ell,i)}\right| \geq \mathsf{T}^{1/3}$. More precisely $\mathcal{M}'^{(\ell)}$ corresponds to the set

$$\left\{\mathcal{M}^{(\ell,i)} \in \mathcal{M}^{(\ell)} \text{ with active items } \mathcal{N}^{(\ell,i)} \mid \Delta_{\ell+1} = \Omega\left(\frac{\sigma\sqrt{\mu^3\log\max(\left|\mathcal{M}^{(\ell,i)}\right|,\left|\mathcal{N}^{(\ell,i)}\right|)}}{\sqrt{\min(\left|\mathcal{M}^{(\ell,i)}\right|,\left|\mathcal{N}^{(\ell,i)}\right|)}}\right)\right.$$

$$\left. \text{and } \left|\mathcal{N}^{(\ell,i)}\right| \geq \mathsf{T}^{1/3}\right\}$$

Suppose $\mathsf{M}/(\tau\mathsf{C}) = \Omega(\mathsf{T}^{1/3})$. As mentioned before, the event $\mathcal{E}_3^{(\ell)}$ is true if the algorithm has successfully computed an estimate $\widehat{\mathbf{P}}^{(\ell)} \in \mathbb{R}^{\mathsf{M}\times\mathsf{N}}$ such that for all $\mathcal{M}^{(\ell,i)} \in \mathcal{M}'^{(\ell)}$

$$\left\|\widetilde{\mathbf{P}}_{\mathcal{M}^{(\ell,i)},\mathcal{N}^{(\ell,i)}}^{(\ell)} - \mathbf{P}_{\mathcal{M}^{(\ell,i)},\mathcal{N}^{(\ell,i)}}\right\|_\infty \leq \Delta_{\ell+1} \text{ for all } \mathcal{M}^{(\ell,i)} \in \mathcal{M}'^{\ell} \quad (11)$$

implying that for each of the distinct nice subsets $\mathcal{M}^{(\ell,i)} \in \mathcal{M}'^{(\ell)}$, after the *explore* component of phase $\ell$, the algorithm finds a good entry-wise estimate of the sub-matrix $\mathbf{P}_{\mathcal{M}^{(\ell,i)},\mathcal{N}^{(\ell,i)}}$. In the following part of the analysis, we will repeatedly condition on the events $\mathcal{E}_2^{(\ell)}$ (conditions A-C are satisfied at the beginning of the *explore* component of phase $\ell$) and the event $\mathcal{E}_3^{(\ell)}$ (eq. 10 is true for all nice subsets $\mathcal{M}^{(\ell,i)} \in \mathcal{M}'^{(\ell)}$ in the explore component of phase $\ell$). Note that the event $\mathcal{E}_2^{(\ell)}$ is true due to the induction hypothesis and conditioned on the event $\mathcal{E}_2^{(\ell)}$, the event $\mathcal{E}_3^{(\ell)}$ is true with probability at least $1 - O(\mathsf{T}^{-3}\mathsf{C})$ (after taking a union bound over $\mathsf{C}$ clusters).

Fix any $\mathcal{M}^{(\ell,i)} \subseteq \mathcal{M}'^{(\ell)}$ and condition on the events $\mathcal{E}_2^{(\ell)}, \mathcal{E}_3^{(\ell)}$. For each user $u \in \mathcal{M}^{(\ell,i)}$, once the algorithm has computed the estimate $\widetilde{\mathbf{P}}_{\mathcal{M}^{(\ell,i)},\mathcal{N}^{(\ell,i)}}^{(\ell)}$ in eq. 11, let us denote a set of good items for the user $u$ by

$$\mathcal{T}_u^{(\ell)} \equiv \{j \in \mathcal{N}^{(\ell,i)} \mid \widetilde{\mathbf{P}}_{uj} \geq \widetilde{\mathbf{P}}_{u\widetilde{\pi}_u(\mathsf{TB}^{-1}-\left|\mathcal{O}_{\mathcal{M}^{(\ell,i)}}^{(\ell,t)}\right|)|\mathcal{N}^{(\ell,i)}} - 2\Delta_{\ell+1}\} \quad (12)$$

$$\mathcal{R}_u^{(\ell)} \equiv \{j \in \mathcal{N}^{(\ell,i)} \mid \widetilde{\mathbf{P}}_{uj} \geq \widetilde{\mathbf{P}}_{u\widetilde{\pi}_u(1)|\mathcal{N}^{(\ell,i)}} - 2\Delta_{\ell+1}\} \quad (13)$$

where $t$ is the round index at the end of the *explore* component of phase $\ell$ and $|\mathcal{O}^{(\ell,t)}_{\mathcal{M}^{(\ell,i)}}|$ is the number of rounds in *exploit* components of phases that the user $u \in \mathcal{M}^{(\ell,i)}$ has encountered so far. Note from our algorithm, that users in the same nice subset $\mathcal{M}^{(\ell,i)}$ have encountered exactly the same number of rounds in *exploit* components of phases until the end of phase $\ell$. Recall that users in $\mathcal{M}^{(\ell,i)}$ have been recommended the same set of items until blocked (unless blocked already for some user in which case the item in question has already been recommended B times) in the *exploit* components of phases until the end of phase $\ell$ (this set of items chosen for recommendation in the *exploit* components up to end of phase $\ell$ is denoted by $\mathcal{O}^{(\ell,t)}_{\mathcal{M}^{(\ell,i)}}$). If we condition on the event $\mathcal{E}^{(\ell)}_3$, then we can show the following statement to be true (in the following lemma, we remove the superscript $t$ in $\mathcal{O}^{(\ell,t)}_{\mathcal{M}^{(\ell,i)}}$ for simplicity - the round index $t$ corresponds to the end of phase $\ell$ for users in $\mathcal{M}^{(\ell,i)}$):

**Lemma 6.** *Condition on the events $\mathcal{E}^{(\ell)}_2, \mathcal{E}^{(\ell)}_3$ being true. Consider a nice subset of users $\mathcal{M}^{(\ell,i)} \in \mathcal{M}'^{(\ell)}$ and their corresponding set of active items $\mathcal{N}^{(\ell,i)}$ for which guarantees in eq. 11 holds. Let $\mathcal{O}^{(\ell)}_{\mathcal{M}^{(\ell,i)}}$ be the set of items that have been chosen for recommendation to users in $\mathcal{M}^{(\ell,i)}$ in exploit components of the first $\ell$ phases. Denote $\mathcal{Z} = [\mathsf{N}] \setminus \mathcal{O}^{(\ell)}_{\mathcal{M}^{(\ell,i)}}$. In that case, for every user $u \in \mathcal{M}^{(\ell,i)}$, the items $\pi_u(s) \mid \mathcal{Z}$ for all $s \in [\mathsf{TB}^{-1} - \left|\mathcal{O}^{(\ell)}_{\mathcal{M}^{(\ell,i)}}\right|]$ must belong to the set $\mathcal{T}^{(\ell)}_u$.*

*Proof.* Let us fix a user $u \in \mathcal{M}^{(\ell,i)}$ with active set of arms $\mathcal{N}^{(\ell,i)}$. Let us also fix an item $a \equiv \pi_u(t) \mid \mathcal{Z}$ for $t \in [\mathsf{TB}^{-1} - \left|\mathcal{O}^{(\ell)}_{\mathcal{M}^{(\ell,i)}}\right|]$. Recall that $\widetilde{\pi} \mid \mathcal{Z}$ is the permutation of the items sorted in descending order according to their estimated reward in $\widetilde{\mathbf{P}}^{(\ell)}_{\mathcal{M}^{(\ell,i)}, \mathcal{N}^{(\ell,i)}}$. Now there are two possibilities regarding the position of the arm $a$ in the permutation $\widetilde{\pi} \mid \mathcal{Z}$ - 1) $\pi_u(t) \mid \mathcal{Z} \equiv \widetilde{\pi}_u(t_1) \mid \mathcal{Z}$ for some $t_1 \leq t$ which implies that in the permutation $\widetilde{\pi} \mid \mathcal{Z}$, the position of the arm $a$ is $t_1$. In that case, the arm $a$ survives in the set $\mathcal{T}^{(\ell)}_u$ by definition (see eq. 12) 2) Now, suppose that $\pi_u(t) \mid \mathcal{Z} \equiv \widetilde{\pi}_u(t_1) \mid \mathcal{Z}$ for some $t_1 \leq t$ which implies that in the permutation $\widetilde{\pi} \mid \mathcal{Z}$, the arm $a$ has been shifted to the right (from $\pi \mid \mathcal{Z}$). In that case, there must exist another item $b \equiv \pi_u(t_2) \mid \mathcal{Z}$ for $t_2 > t$ such that $\pi_u(t_2) \mid \mathcal{Z} \equiv \widetilde{\pi}_u(t_3) \mid \mathcal{Z}$ for $t_3 \leq t$. Now, we will have

$$\widetilde{\mathbf{P}}^{(\ell)}_{ub} - \widetilde{\mathbf{P}}^{(\ell)}_{ua} = \widetilde{\mathbf{P}}^{(\ell)}_{ub} - \mathbf{P}_{ub} + \mathbf{P}_{ub} - \mathbf{P}_{ua} + \mathbf{P}_{ua} - \widetilde{\mathbf{P}}^{(\ell)}_{ua} \leq 2\Delta_{\ell+1}$$

where we used the following facts a) conditioned on the event $\mathcal{E}^{(\ell)}_3$, we have $\widetilde{\mathbf{P}}^{(\ell)}_{us} - \mathbf{P}_{us} \leq \Delta_{\ell+1}$ for all $u \in \mathcal{M}^{(\ell,i)}, s \in \mathcal{N}^{(\ell,i)}$ b) $\mathbf{P}_{ub} - \mathbf{P}_{ua} \leq 0$ by definition. Hence, this implies that $a \in \mathcal{T}^{(\ell)}_u$. $\qquad \square$

**Corollary 1.** *Condition on the events $\mathcal{E}^{(\ell)}_2, \mathcal{E}^{(\ell)}_3$ being true. Consider a nice subset of users $\mathcal{M}^{(\ell,i)} \in \mathcal{M}'^{(\ell)}$ and their corresponding set of active items $\mathcal{N}^{(\ell,i)}$ for which guarantees in eq. 11 holds. Suppose $t_\ell$ is the final round of the explore component of phase $\ell$ for users in $\mathcal{M}^{(\ell,i)}$. In that case, for user $u$, the set $\mathcal{T}^{(\ell)}_u$ contains sufficient items that are unblocked and can be recommended for the remaining $\mathsf{T} - t_\ell$ rounds.*

*Proof.* From Lemma 11, we showed that $\mathcal{T}^{(\ell)}_u$ comprises the set of items $\pi_u(s) \mid \mathcal{Z}$ for all $s \in [\mathsf{TB}^{-1} - \left|\mathcal{O}^{(\ell)}_{\mathcal{M}^{(\ell,i)}}\right|]$ where $\mathcal{Z} = [\mathsf{N}] \setminus \mathcal{O}^{(\ell)}_{\mathcal{M}^{(\ell,i)}}$. Now, suppose the $k^{\text{th}}$ item in the set $\mathcal{H}_u \equiv \{\pi_u(t') \mid \mathcal{Z}\}^{\mathsf{T} - \left|\mathcal{O}^{(\ell)}_{\mathcal{M}^{(\ell,i)}}\right|}_{t'=1}$ has been recommended to user $u$ $b_k < \mathsf{B}$ times in previous phases. In that case, the number of allowed recommendations of items in $\mathcal{T}^{(\ell)}_u$ is at least $\mathsf{TB}^{-1} - \mathsf{B}\left|\mathcal{O}^{(\ell)}_{\mathcal{M}^{(\ell,i)}}\right| - \sum_{k \in \mathcal{H}_u} b_k$ which is more than the remaining rounds that is at most $\mathsf{TB}^{-1} - \mathsf{B}\left|\mathcal{O}^{(\ell)}_{\mathcal{M}^{(\ell,i)}}\right| - \sum_{k \in \mathcal{H}_u} b_k$ ($\mathsf{B}\left|\mathcal{O}^{(\ell)}_{\mathcal{M}^{(\ell,i)}}\right|$ rounds have already been used up when the *golden items* were identified and recommended). $\qquad \square$

Consider a *nice* subset of users $\mathcal{M}^{(\ell,i)} \in \mathcal{M}'^{(\ell)}$ and its corresponding active set of items $\mathcal{N}^{(\ell,i)}$ for which eq. 11 holds true. At the end of the *explore* component of phase $\ell$, based on the estimate $\widetilde{\mathbf{P}}^{(\ell)}_{\mathcal{M}^{(\ell,i)}, \mathcal{N}^{(\ell,i)}}$, we construct a graph $\mathcal{G}^{(\ell,i)}$ whose nodes are given by the users in $\mathcal{M}^{(\ell,i)}$. Now, we

draw an edge between two users $u, v \in \mathcal{M}^{(\ell,i)}$ if $\left|\widetilde{\mathbf{P}}^{(\ell)}_{ux} - \widetilde{\mathbf{P}}^{(\ell)}_{vx}\right| \le 2\Delta_{\ell+1}$ for all considered items $x \in \mathcal{N}^{(\ell,i)}$.

**Lemma 7.** *Condition on the events $\mathcal{E}_2^{(\ell)}, \mathcal{E}_3^{(\ell)}$ being true. Consider a nice subset of users $\mathcal{M}^{(\ell,i)} \in \mathcal{M}'^{(\ell)}$ and their corresponding set of active items $\mathcal{N}^{(\ell,i)}$ for which guarantees in eq. 11 holds. Consider the graph $\mathcal{G}^{(\ell,i)}$ formed by the users in $\mathcal{M}^{(\ell,i)}$ such that an edge exists between two users $u, v \in \mathcal{M}^{(\ell,i)}$ if $\left|\widetilde{\mathbf{P}}^{(\ell)}_{ux} - \widetilde{\mathbf{P}}^{(\ell)}_{vx}\right| \le 2\Delta_{\ell+1}$ for all considered items $x \in \mathcal{N}^{(\ell,i)}$. Nodes in $\mathcal{G}^{(\ell,i)}$ corresponding to users in the same cluster form a clique. Also, users in each connected component of the graph $\mathcal{G}^{(\ell,i)}$ form a nice subset of users.*

*Proof.* For any two users $u, v \in \mathcal{M}^{(\ell,i)}$ belonging to the same cluster, consider an arm $x \in \mathcal{N}^{(\ell,i)}$. We must have

$$\widetilde{\mathbf{P}}^{(\ell)}_{ux} - \widetilde{\mathbf{P}}^{(\ell)}_{vx} = \widetilde{\mathbf{P}}^{(\ell)}_{ux} - \mathbf{P}_{ux} + \mathbf{P}_{ux} - \mathbf{P}_{vx} + \mathbf{P}_{vx} - \widetilde{\mathbf{P}}^{(\ell)}_{vx} \le 2\Delta_{\ell+1}.$$

Now, consider two users $u, v \in \mathcal{M}^{(\ell,i)}$ that belongs to different clusters $\mathcal{P}, \mathcal{Q}$ respectively. Note that since the event $\mathcal{E}^{(\ell)}$ is true, $\mathcal{M}^{(\ell,i)}$ is a union of clusters comprising $\mathcal{P}, \mathcal{Q}$. Furthermore, we have already established that nodes in $\mathcal{G}^{(\ell,i)}$ (users in $\mathcal{M}^{(\ell,i)}$) restricted to the same cluster form a clique. There every connected component of the graph $\mathcal{G}^{(\ell,i)}$ can be represented as a union of a subset of clusters.

$\square$

**Lemma 8.** *Condition on the events $\mathcal{E}_2^{(\ell)}, \mathcal{E}_3^{(\ell)}$ being true. Consider a nice subset of users $\mathcal{M}^{(\ell,i)} \in \mathcal{M}'^{(\ell)}$ and their corresponding set of active items $\mathcal{N}^{(\ell,i)}$ for which guarantees in eq. 11 holds. In that case, for any subset $\mathcal{Y} \subseteq \mathcal{N}^{(\ell,i)}$ and any $s \in [|\mathcal{Y}|]$, for every user $u \in \mathcal{M}^{(\ell,i)}$, we must have*

$$\widetilde{\mathbf{P}}_{u\widetilde{\pi}_u(1)|\mathcal{Y}} - \widetilde{\mathbf{P}}_{u\widetilde{\pi}_u(s)|\mathcal{Y}} - 6\Delta_{\ell+1} \le \mathbf{P}_{u\pi_u(1)|\mathcal{Y}} - \mathbf{P}_{u\pi_u(s)|\mathcal{Y}} \le \widetilde{\mathbf{P}}_{u\widetilde{\pi}_u(1)|\mathcal{Y}} - \widetilde{\mathbf{P}}_{u\widetilde{\pi}_u(s)|\mathcal{Y}} + 6\Delta_{\ell+1}.$$

*Proof.* Since, we condition on the event $\mathcal{E}_3^{(\ell)}$, we must have computed an estimate $\widetilde{\mathbf{P}}^{(\ell)}$, an estimate of $\mathbf{P}$ restricted to users in $\mathcal{M}^{(\ell,i)}$ and items in $\mathcal{N}^{(\ell,i)}$ such that

$$\left\|\widetilde{\mathbf{P}}^{(\ell)}_{\mathcal{M}^{(\ell,i)},\mathcal{N}^{(\ell,i)}} - \mathbf{P}_{\mathcal{M}^{(\ell,i)},\mathcal{N}^{(\ell,i)}}\right\|_\infty \le \Delta_{\ell+1}.$$

We can decompose the term being studied in the following manner:

$$\widetilde{\mathbf{P}}^{(\ell)}_{u\pi_u(1)|\mathcal{Y}} - \widetilde{\mathbf{P}}^{(\ell)}_{u\pi_u(s)|\mathcal{Y}}$$
$$= \widetilde{\mathbf{P}}^{(\ell)}_{u\pi_u(1)|\mathcal{Y}} - \widetilde{\mathbf{P}}^{(\ell)}_{u\widetilde{\pi}_u(1)|\mathcal{Y}} + \widetilde{\mathbf{P}}^{(\ell)}_{u\widetilde{\pi}_u(1)|\mathcal{Y}} - \widetilde{\mathbf{P}}^{(\ell)}_{u\widetilde{\pi}_u(s)|\mathcal{Y}} + \widetilde{\mathbf{P}}^{(\ell)}_{u\widetilde{\pi}_u(s)|\mathcal{Y}} - \widetilde{\mathbf{P}}^{(\ell)}_{u\pi_u(s)|\mathcal{Y}}.$$

Let us bound the quantity $\widetilde{\mathbf{P}}^{(\ell)}_{u\widetilde{\pi}_u(s)|\mathcal{Y}} - \widetilde{\mathbf{P}}^{(\ell)}_{u\pi_u(s)|\mathcal{Y}}$. In order to analyze this quantity, we will consider a few cases. In the first case, suppose that $\pi_u(s) \mid \mathcal{Y} = \widetilde{\pi}_u(t_1) \mid \mathcal{Y}$ for $t_1 \ge s$. In that case, we will have that $\widetilde{\mathbf{P}}^{(\ell)}_{u\widetilde{\pi}_u(s)|\mathcal{Y}} - \widetilde{\mathbf{P}}^{(\ell)}_{u\pi_u(s)|\mathcal{Y}} \ge 0$. In the second case, suppose $\pi_u(s) \mid \mathcal{Y} = \widetilde{\pi}_u(t_2) \mid \mathcal{Y}$ for $t_2 < s$ and $\pi_u(t_3) \mid \mathcal{Y} = \widetilde{\pi}_u(s) \mid \mathcal{Y}$ for $t_3 < s$. In that case, we have

$$\widetilde{\mathbf{P}}^{(\ell)}_{u\widetilde{\pi}_u(s)|\mathcal{Y}} - \widetilde{\mathbf{P}}^{(\ell)}_{u\pi_u(s)|\mathcal{Y}}$$
$$= \widetilde{\mathbf{P}}^{(\ell)}_{u\widetilde{\pi}_u(s)|\mathcal{Y}} - \mathbf{P}_{u\widetilde{\pi}_u(s)|\mathcal{Y}} + \mathbf{P}_{u\widetilde{\pi}_u(s)|\mathcal{Y}} - \mathbf{P}_{u\pi_u(s)|\mathcal{Y}} + \mathbf{P}_{u\pi_u(s)|\mathcal{Y}} - \widetilde{\mathbf{P}}^{(\ell)}_{u\pi_u(s)|\mathcal{Y}}$$
$$= \widetilde{\mathbf{P}}^{(\ell)}_{u\widetilde{\pi}_u(s)|\mathcal{Y}} - \mathbf{P}_{u\widetilde{\pi}_u(s)|\mathcal{Y}} + \mathbf{P}_{u\pi_u(t_3)|\mathcal{Y}} - \mathbf{P}_{u\pi_u(s)|\mathcal{Y}} + \mathbf{P}_{u\pi_u(s)|\mathcal{Y}} - \widetilde{\mathbf{P}}^{(\ell)}_{u\pi_u(s)|\mathcal{Y}} \ge -2\Delta_{\ell+1}$$

where we used the fact $\mathbf{P}_{u\pi_u(t_3)|\mathcal{Y}} - \mathbf{P}_{u\pi_u(s)|\mathcal{Y}} \ge 0$. In the final case, we assume that $\pi_u(s) \mid \mathcal{Y} = \widetilde{\pi}_u(t_2) \mid \mathcal{Y}$ for $s > t_2$ and $\pi_u(t_3) \mid \mathcal{Y} = \widetilde{\pi}_u(s) \mid \mathcal{Y}$ for $t_3 > s$. This means that both the items $\pi_u(s) \mid \mathcal{Y}, \pi_u(t_3) \mid \mathcal{Y}$ have been shifted to the left in the permutation $\widetilde{\pi}_u \mid \mathcal{Y}$. Hence,

$$\widetilde{\mathbf{P}}^{(\ell)}_{u\widetilde{\pi}_u(s)|\mathcal{Y}} - \widetilde{\mathbf{P}}^{(\ell)}_{u\pi_u(s)|\mathcal{Y}}$$
$$= \widetilde{\mathbf{P}}^{(\ell)}_{u\widetilde{\pi}_u(s)|\mathcal{Y}} - \mathbf{P}_{u\widetilde{\pi}_u(s)|\mathcal{Y}} + \mathbf{P}_{u\widetilde{\pi}_u(s)|\mathcal{Y}} - \mathbf{P}_{u\pi_u(s)|\mathcal{Y}}$$
$$+ \mathbf{P}_{u\pi_u(s)|\mathcal{Y}} - \widetilde{\mathbf{P}}^{(\ell)}_{u\pi_u(s)|\mathcal{Y}} \ge -2\Delta_{\ell+1} + \mathbf{P}_{u\widetilde{\pi}_u(s)|\mathcal{Y}} - \mathbf{P}_{u\pi_u(s)|\mathcal{Y}} = -2\Delta_{\ell+1} + \mathbf{P}_{u\pi_u(t_3)|\mathcal{Y}} - \mathbf{P}_{u\pi_u(s)|\mathcal{Y}}.$$

Hence there must exist an element $\pi_u(t_4) \mid \mathcal{Y}$ such that $t_4 < s$ and $\widetilde{\pi}_u(t_5) \mid \mathcal{Y} = \pi_u(t_4) \mid \mathcal{Y}$ for $t_5 > s$. In that case, we must have

$$\mathbf{P}_{u\pi_u(t_3)|\mathcal{Y}} - \mathbf{P}_{u\pi_u(s)|\mathcal{Y}} \geq \mathbf{P}_{u\pi_u(t_3)|\mathcal{Y}} - \mathbf{P}_{u\pi_u(t_4)|\mathcal{Y}}$$

$$= \mathbf{P}_{u\pi_u(t_3)|\mathcal{Y}} - \widetilde{\mathbf{P}}^{(\ell)}_{u\pi_u(t_3)|\mathcal{Y}} + \widetilde{\mathbf{P}}^{(\ell)}_{u\pi_u(t_3)|\mathcal{Y}} - \widetilde{\mathbf{P}}^{(\ell)}_{u\pi_u(t_4)|\mathcal{Y}} + \widetilde{\mathbf{P}}^{(\ell)}_{u\pi_u(t_4)|\mathcal{Y}} - \mathbf{P}_{u\pi_u(t_4)|\mathcal{Y}}$$

$$\geq -2\Delta_{\ell+1} + \widetilde{\mathbf{P}}^{(\ell)}_{u\pi_u(t_3)|\mathcal{Y}} - \widetilde{\mathbf{P}}^{(\ell)}_{u\pi_u(t_4)|\mathcal{Y}} = -2\Delta_{\ell+1} + \widetilde{\mathbf{P}}^{(\ell)}_{u\widetilde{\pi}_u(s)|\mathcal{Y}} - \widetilde{\mathbf{P}}^{(\ell)}_{u\widetilde{\pi}_u(t_5)|\mathcal{Y}} \geq -2\Delta_{\ell+1}.$$

Therefore, in this case, we get that $\widetilde{\mathbf{P}}^{(\ell)}_{u\widetilde{\pi}_u(s)} - \widetilde{\mathbf{P}}^{(\ell)}_{u\pi_u(s)} \geq -4\Delta_{\ell+1}$. Again, we will have that

$$\widetilde{\mathbf{P}}^{(\ell)}_{u\pi_u(1)|\mathcal{Y}} - \widetilde{\mathbf{P}}^{(\ell)}_{u\widetilde{\pi}_u(1)|\mathcal{Y}} = \widetilde{\mathbf{P}}^{(\ell)}_{u\pi_u(1)|\mathcal{Y}} - \mathbf{P}^{(\ell)}_{u\pi_u(1)|\mathcal{Y}}$$

$$+ \mathbf{P}^{(\ell)}_{u\pi_u(1)|\mathcal{Y}} - \mathbf{P}^{(\ell)}_{u\widetilde{\pi}_u(1)|\mathcal{Y}} + \mathbf{P}^{(\ell)}_{u\widetilde{\pi}_u(1)|\mathcal{Y}} - \widetilde{\mathbf{P}}^{(\ell)}_{u\widetilde{\pi}_u(1)|\mathcal{Y}} \geq -2\Delta_{\ell+1}.$$

By combining the above arguments, we have that

$$\widetilde{\mathbf{P}}^{(\ell)}_{u\pi_u(1)|\mathcal{Y}} - \widetilde{\mathbf{P}}^{(\ell)}_{u\pi_u(s)|\mathcal{Y}} \geq \widetilde{\mathbf{P}}^{(\ell)}_{u\widetilde{\pi}_u(1)|\mathcal{Y}} - \widetilde{\mathbf{P}}^{(\ell)}_{u\widetilde{\pi}_u(s)|\mathcal{Y}} - 6\Delta_{\ell+1}.$$

By a similar set of arguments involving triangle inequalities, we will also have

$$\widetilde{\mathbf{P}}^{(\ell)}_{u\pi_u(1)|\mathcal{Y}} - \widetilde{\mathbf{P}}^{(\ell)}_{u\pi_u(s)|\mathcal{Y}} \leq \widetilde{\mathbf{P}}^{(\ell)}_{u\widetilde{\pi}_u(1)|\mathcal{Y}} - \widetilde{\mathbf{P}}^{(\ell)}_{u\widetilde{\pi}_u(s)|\mathcal{Y}} + 6\Delta_{\ell+1}.$$

This completes the proof of the lemma. $\qquad\square$

We now show the following lemma characterizing the union of good items for a connected component of the graph $\mathcal{G}^{(\ell,i)}$. Recall that $\mathsf{T} - \left|\mathcal{O}^{(\ell)}_{\mathcal{M}^{(\ell,i)}}\right|$ counts the number of rounds excluding the ones used up in *exploit* component so far up to the $\ell^{\text{th}}$ phase.

**Lemma 9.** *Condition on the events $\mathcal{E}_2^{(\ell)}, \mathcal{E}_3^{(\ell)}$ being true. Consider a nice subset of users $\mathcal{M}^{(\ell,i)} \in \mathcal{M}'^{(\ell)}$ and their corresponding set of active items $\mathcal{N}^{(\ell,i)}$ for which guarantees in eq. 11 holds. Consider a subset of users $\mathcal{G} \in \mathcal{M}^{(\ell,i)}$ forming a connected component. Fix any set $\mathcal{Y} = \bigcup_{u\in\mathcal{G}} \mathcal{T}_g^{(\ell)} \setminus \mathcal{J}$ for some $\mathcal{J}$ such that for every user $u \in \mathcal{G}$, we have $\widetilde{\pi}_u(s') \mid \mathcal{N}^{(\ell,i)} \equiv \widetilde{\pi}_u(s) \mid \mathcal{Y}$ for $s' = \mathsf{TB}^{-1} - \left|\mathcal{O}^{(\ell)}_{\mathcal{M}^{(\ell,i)}}\right|$ and some common index $s$. In that case, we must have*

$$\max_{v\in\mathcal{G}} \left( \max_{x,y\in\mathcal{Y}} \widetilde{\mathbf{P}}_{vx} - \widetilde{\mathbf{P}}_{vy} \right) \leq \max_{u\in\mathcal{G}} \left( \widetilde{\mathbf{P}}_{u\widetilde{\pi}_u(1)|\mathcal{Y}} - \widetilde{\mathbf{P}}_{u\widetilde{\pi}_u(s')|\mathcal{N}^{(\ell,i)}} \right) + 24\mathsf{C}\Delta_{\ell+1}$$

*Proof.* Let us fix a user $v \in \mathcal{G}$. We have $\max_{x,y\in\mathcal{Y}} \widetilde{\mathbf{P}}_{vx} - \widetilde{\mathbf{P}}_{vy} = \widetilde{\mathbf{P}}_{v\widetilde{\pi}_v(1)|\mathcal{Y}} - \min_{y\in\mathcal{Y}} \widetilde{\mathbf{P}}_{vy}$. Now, there are two possibilities for any $y \in \mathcal{Y}$: first, suppose that $y \in \mathcal{T}_v^{(\ell)}$. In that case, we have

$$\widetilde{\mathbf{P}}_{v\widetilde{\pi}_v(1)|\mathcal{Y}} - \widetilde{\mathbf{P}}_{vy} = \widetilde{\mathbf{P}}_{v\widetilde{\pi}_v(1)|\mathcal{Y}} - \widetilde{\mathbf{P}}_{v\widetilde{\pi}_v(s)|\mathcal{Y}} + \widetilde{\mathbf{P}}_{v\widetilde{\pi}_v(s)|\mathcal{Y}} - \widetilde{\mathbf{P}}_{v\widetilde{\pi}_v(s')|\mathcal{T}_v^{(\ell)}} + \widetilde{\mathbf{P}}_{v\widetilde{\pi}_v(s')|\mathcal{T}_v^{(\ell)}} - \widetilde{\mathbf{P}}_{vy}$$

Recall that the set $\mathcal{T}_v^{(\ell)}$ was constructed as

$$\mathcal{T}_v^{(\ell)} \equiv \{j \in \mathcal{N}^{(\ell,i)} \mid \widetilde{\mathbf{P}}_{vj} \geq \widetilde{\mathbf{P}}_{v\widetilde{\pi}_v(s')|\mathcal{N}^{(\ell,i)}} - 2\Delta_{\ell+1}\} \text{ where } s' = \mathsf{TB}^{-1} - \left|\mathcal{O}^{(\ell)}_{\mathcal{M}^{(\ell,i)}}\right| \quad (14)$$

Since $y \in \mathcal{T}_v^{(\ell)}$, we can bound $\widetilde{\mathbf{P}}_{v\widetilde{\pi}_v(s')|\mathcal{T}_v^{(\ell)}} - \widetilde{\mathbf{P}}_{vy} \leq 2\Delta_{\ell+1}$. Also, from the construction of $\mathcal{T}_v^{(\ell)}$, $\{\widetilde{\pi}_v(r) \mid \mathcal{N}^{(\ell,i)}\}_{r=1}^s$ is present in the set $\mathcal{T}_v^{(\ell)}$. Also, since $\mathcal{T}_u^{(\ell)}$ is just a subset of $\mathcal{N}^{(\ell,i)}$, the positions of the items $\{\widetilde{\pi}_v(r) \mid \mathcal{N}^{(\ell,i)}\}_{r=1}^s$ do not change in the permutation corresponding to the items in $\mathcal{T}_v^{(\ell)}$ sorted by estimated reward for $v$ in decreasing order i.e. $\widetilde{\pi}_v(r) \mid \mathcal{N}^{(\ell,i)} = \widetilde{\pi}_v(r) \mid \mathcal{T}_v^{(\ell)}$ for any $r \in [s]$. Hence $\widetilde{\mathbf{P}}_{v\widetilde{\pi}_v(s)|\mathcal{Y}} - \widetilde{\mathbf{P}}_{v\widetilde{\pi}_v(s')|\mathcal{T}_v^{(\ell)}} = 0$. Hence, by combining the above, we have

$$\widetilde{\mathbf{P}}_{v\widetilde{\pi}_v(1)|\mathcal{Y}} - \widetilde{\mathbf{P}}_{vy} \leq \widetilde{\mathbf{P}}_{v\widetilde{\pi}_v(1)|\mathcal{Y}} - \widetilde{\mathbf{P}}_{v\widetilde{\pi}_v(s)|\mathcal{Y}} + 2\Delta_{\ell+1}.$$

Next, consider the case when $y \notin \mathcal{T}_v^{(\ell)}$. Hence, there must exist an user $u$ such that $y \in \mathcal{T}_u^{(\ell)} \setminus \mathcal{T}_v^{(\ell)}$ and $u$ is connected to $v$ via a path of length $\mathsf{L} \leq 2\mathsf{C} - 1$ (since there are $\mathsf{C}$ clusters and users in the same cluster form a clique as proved in Lemma 7). In that case, we have the following decomposition:

$$\widetilde{\mathbf{P}}_{v\widetilde{\pi}_v(1)|\mathcal{Y}} - \widetilde{\mathbf{P}}_{vy} = \widetilde{\mathbf{P}}_{v\widetilde{\pi}_v(1)|\mathcal{Y}} - \widetilde{\mathbf{P}}_{v\widetilde{\pi}_u(1)|\mathcal{Y}} + \widetilde{\mathbf{P}}_{v\widetilde{\pi}_u(1)|\mathcal{Y}} - \widetilde{\mathbf{P}}_{v\widetilde{\pi}_u(s)|\mathcal{Y}} + \widetilde{\mathbf{P}}_{v\widetilde{\pi}_u(s)|\mathcal{Y}} - \widetilde{\mathbf{P}}_{u\widetilde{\pi}_u(s)|\mathcal{Y}}$$

$$+ \widetilde{\mathbf{P}}_{u\widetilde{\pi}_u(s)|\mathcal{Y}} - \widetilde{\mathbf{P}}_{u\widetilde{\pi}_u(s')|\mathcal{T}_u^{(\ell)}} + \widetilde{\mathbf{P}}_{u\widetilde{\pi}_u(s')|\mathcal{T}_u^{(\ell)}} - \widetilde{\mathbf{P}}_{uy} + \widetilde{\mathbf{P}}_{uy} - \widetilde{\mathbf{P}}_{vy}$$

Now, let us consider the terms pairwise. Due to our construction of the graph $\mathcal{G}^{(\ell,i)}$, for two users $u, v$ connected via a path of length L, we must have $\left|\widetilde{\mathbf{P}}_{ux}^{(\ell)} - \widetilde{\mathbf{P}}_{vx}^{(\ell)}\right| \leq 2\mathsf{L}\Delta_{\ell+1}$. Note that

$$\widetilde{\mathbf{P}}_{v\widetilde{\pi}_v(1)|\mathcal{Y}}^{(\ell)} - \widetilde{\mathbf{P}}_{v\widetilde{\pi}_u(1)|\mathcal{Y}}^{(\ell)} = \widetilde{\mathbf{P}}_{v\widetilde{\pi}_v(1)|\mathcal{Y}}^{(\ell)} - \mathbf{P}_{u\widetilde{\pi}_v(1)|\mathcal{Y}}^{(\ell)} + \widetilde{\mathbf{P}}_{u\widetilde{\pi}_v(1)|\mathcal{Y}}^{(\ell)}$$

$$- \widetilde{\mathbf{P}}_{u\widetilde{\pi}_u(1)|\mathcal{Y}}^{(\ell)} + \mathbf{P}_{u\widetilde{\pi}_u(1)|\mathcal{Y}}^{(\ell)} - \widetilde{\mathbf{P}}_{v\widetilde{\pi}_u(1)|\mathcal{Y}}^{(\ell)} \leq 4\mathsf{L}\Delta_{\ell+1}$$

where we used that $\widetilde{\mathbf{P}}_{u\widetilde{\pi}_v(1)|\mathcal{Y}}^{(\ell)} - \widetilde{\mathbf{P}}_{u\widetilde{\pi}_u(1)|\mathcal{Y}}^{(\ell)} \leq 0$. Next, we have $\widetilde{\mathbf{P}}_{v\widetilde{\pi}_u(1)|\mathcal{Y}} - \widetilde{\mathbf{P}}_{v\widetilde{\pi}_u(s)|\mathcal{Y}} \leq \widetilde{\mathbf{P}}_{u\widetilde{\pi}_u(1)|\mathcal{Y}} - \widetilde{\mathbf{P}}_{u\widetilde{\pi}_u(s)|\mathcal{Y}} + 4\mathsf{L}\Delta_{\ell+1}$. Furthermore, again we have $\widetilde{\pi}_u(r) \mid \mathcal{N}^{(\ell,i)} = \widetilde{\pi}_u(r) \mid \mathcal{T}_u^{(\ell)}$ for any $r \in [s]$ and therefore $\widetilde{\mathbf{P}}_{u\widetilde{\pi}_u(s)|\mathcal{Y}} - \widetilde{\mathbf{P}}_{u\widetilde{\pi}_u(s')|\mathcal{T}_u^{(\ell)}} = 0$. Finally, $\widetilde{\mathbf{P}}_{u\widetilde{\pi}_u(s')|\mathcal{T}_u^{(\ell)}} - \widetilde{\mathbf{P}}_{uy} + \widetilde{\mathbf{P}}_{uy} - \widetilde{\mathbf{P}}_{vy} \leq 4\mathsf{L}\Delta_{\ell+1}$ (since $\widetilde{\mathbf{P}}_{u\widetilde{\pi}_u(s')|\mathcal{T}_u^{(\ell)}} - \widetilde{\mathbf{P}}_{uy} \leq 2\Delta_{\ell+1}$). Hence by combining, we have that

$$\widetilde{\mathbf{P}}_{v\widetilde{\pi}_v(1)|\mathcal{Y}} - \widetilde{\mathbf{P}}_{vy} \leq \widetilde{\mathbf{P}}_{u\widetilde{\pi}_u(1)|\mathcal{Y}} - \widetilde{\mathbf{P}}_{u\widetilde{\pi}_u(s)|\mathcal{Y}} + 12\mathsf{L}\Delta_{\ell+1}.$$

Hence, we complete the proof of the lemma (by substituting $\mathsf{L} \leq 2\mathsf{C}$). $\qquad\square$

**Lemma 10.** *Condition on the events $\mathcal{E}_2^{(\ell)}, \mathcal{E}_3^{(\ell)}$ being true. Consider a nice subset of users $\mathcal{M}^{(\ell,i)} \in \mathcal{M}'^{(\ell)}$ and their corresponding set of active items $\mathcal{N}^{(\ell,i)}$ for which guarantees in eq. 11 holds. Consider two users $u, v \in \mathcal{M}^{(\ell,i)}$ having an edge i.e. $\left|\widetilde{\mathbf{P}}_{ux}^{(\ell)} - \widetilde{\mathbf{P}}_{vx}^{(\ell)}\right| \leq 2\Delta_{\ell+1}$ for all $x \in \mathcal{N}^{(\ell,i)}$. In that case, for any subset $\mathcal{Y} \subseteq \mathcal{N}^{(\ell,i)}$ and any $s \in [|\mathcal{Y}|]$, for every user $u \in \mathcal{M}^{(\ell,i)}$, we must have*

$$\widetilde{\mathbf{P}}_{v\widetilde{\pi}_v(1)|\mathcal{Y}} - \widetilde{\mathbf{P}}_{v\widetilde{\pi}_v(s)|\mathcal{Y}} - 12\Delta_{\ell+1} \leq \widetilde{\mathbf{P}}_{u\widetilde{\pi}_u(1)|\mathcal{Y}} - \widetilde{\mathbf{P}}_{u\widetilde{\pi}_u(s)|\mathcal{Y}} \leq \widetilde{\mathbf{P}}_{v\widetilde{\pi}_v(1)|\mathcal{Y}} - \widetilde{\mathbf{P}}_{v\widetilde{\pi}_v(s)|\mathcal{Y}} + 12\Delta_{\ell+1}$$

*Proof.* We can decompose the term being studied in the following manner:

$$\widetilde{\mathbf{P}}_{v\widetilde{\pi}_u(1)|\mathcal{Y}}^{(\ell)} - \widetilde{\mathbf{P}}_{v\widetilde{\pi}_u(s)|\mathcal{Y}}^{(\ell)}$$
$$= \widetilde{\mathbf{P}}_{v\widetilde{\pi}_u(1)|\mathcal{Y}}^{(\ell)} - \widetilde{\mathbf{P}}_{v\widetilde{\pi}_v(1)|\mathcal{Y}}^{(\ell)} + \widetilde{\mathbf{P}}_{v\widetilde{\pi}_v(1)|\mathcal{Y}}^{(\ell)} - \widetilde{\mathbf{P}}_{v\widetilde{\pi}_v(s)|\mathcal{Y}}^{(\ell)} + \widetilde{\mathbf{P}}_{v\widetilde{\pi}_v(s)|\mathcal{Y}}^{(\ell)} - \widetilde{\mathbf{P}}_{v\widetilde{\pi}_u(s)|\mathcal{Y}}^{(\ell)}.$$

Let us bound the quantity $\widetilde{\mathbf{P}}_{v\widetilde{\pi}_v(s)|\mathcal{Y}}^{(\ell)} - \widetilde{\mathbf{P}}_{v\widetilde{\pi}_u(s)|\mathcal{Y}}^{(\ell)}$. In order to analyze this quantity, we will consider a few cases. In the first case, suppose that $\widetilde{\pi}_u(s) \mid \mathcal{Y} = \widetilde{\pi}_v(t_1) \mid \mathcal{Y}$ for $t_1 \geq s$. In that case, we will have that $\widetilde{\mathbf{P}}_{v\widetilde{\pi}_v(s)|\mathcal{Y}}^{(\ell)} - \widetilde{\mathbf{P}}_{v\widetilde{\pi}_u(s)|\mathcal{Y}}^{(\ell)} \geq 0$. In the second case, suppose $\widetilde{\pi}_u(s) \mid \mathcal{Y} = \widetilde{\pi}_v(t_2) \mid \mathcal{Y}$ for $t_2 < s$ and $\widetilde{\pi}_u(t_3) \mid \mathcal{Y} = \widetilde{\pi}_v(s) \mid \mathcal{Y}$ for $t_3 < s$. In that case, we have

$$\widetilde{\mathbf{P}}_{v\widetilde{\pi}_v(s)|\mathcal{Y}}^{(\ell)} - \widetilde{\mathbf{P}}_{v\widetilde{\pi}_u(s)|\mathcal{Y}}^{(\ell)}$$
$$= \widetilde{\mathbf{P}}_{v\widetilde{\pi}_v(s)|\mathcal{Y}}^{(\ell)} - \widetilde{\mathbf{P}}_{u\widetilde{\pi}_v(s)|\mathcal{Y}}^{(\ell)} + \widetilde{\mathbf{P}}_{u\widetilde{\pi}_v(s)|\mathcal{Y}}^{(\ell)} - \widetilde{\mathbf{P}}_{u\widetilde{\pi}_u(s)|\mathcal{Y}}^{(\ell)} + \widetilde{\mathbf{P}}_{u\widetilde{\pi}_u(s)|\mathcal{Y}}^{(\ell)} - \widetilde{\mathbf{P}}_{v\widetilde{\pi}_u(s)|\mathcal{Y}}^{(\ell)} \geq -4\Delta_{\ell+1}$$

where we used the fact $\widetilde{\mathbf{P}}_{u\widetilde{\pi}_v(s)|\mathcal{Y}}^{(\ell)} - \widetilde{\mathbf{P}}_{u\widetilde{\pi}_u(s)|\mathcal{Y}}^{(\ell)} = \widetilde{\mathbf{P}}_{u\widetilde{\pi}_u(t_3)|\mathcal{Y}}^{(\ell)} - \widetilde{\mathbf{P}}_{u\widetilde{\pi}_u(s)|\mathcal{Y}}^{(\ell)} \geq 0$. In the final case, we assume that $\widetilde{\pi}_u(s) \mid \mathcal{Y} = \widetilde{\pi}_v(t_2) \mid \mathcal{Y}$ for $s > t_2$ and $\widetilde{\pi}_u(t_3) \mid \mathcal{Y} = \widetilde{\pi}_v(s) \mid \mathcal{Y}$ for $t_3 > s$. This means that both the items $\widetilde{\pi}_u(s) \mid \mathcal{Y}, \widetilde{\pi}_u(t_3) \mid \mathcal{Y}$ have been shifted to the left in the permutation $\widetilde{\pi}_v \mid \mathcal{Y}$. Hence,

$$\widetilde{\mathbf{P}}_{v\widetilde{\pi}_v(s)|\mathcal{Y}}^{(\ell)} - \widetilde{\mathbf{P}}_{v\widetilde{\pi}_u(s)|\mathcal{Y}}^{(\ell)}$$
$$= \widetilde{\mathbf{P}}_{v\widetilde{\pi}_v(s)|\mathcal{Y}}^{(\ell)} - \widetilde{\mathbf{P}}_{u\widetilde{\pi}_v(s)|\mathcal{Y}} + \widetilde{\mathbf{P}}_{u\widetilde{\pi}_v(s)|\mathcal{Y}} - \widetilde{\mathbf{P}}_{u\widetilde{\pi}_u(s)|\mathcal{Y}}$$
$$+ \widetilde{\mathbf{P}}_{u\widetilde{\pi}_u(s)|\mathcal{Y}} - \widetilde{\mathbf{P}}_{v\widetilde{\pi}_u(s)|\mathcal{Y}}^{(\ell)} \geq -4\Delta_{\ell+1} + \widetilde{\mathbf{P}}_{u\widetilde{\pi}_v(s)|\mathcal{Y}} - \widetilde{\mathbf{P}}_{u\widetilde{\pi}_u(s)|\mathcal{Y}}$$
$$= -4\Delta_{\ell+1} + \widetilde{\mathbf{P}}_{u\widetilde{\pi}_u(t_3)|\mathcal{Y}} - \widetilde{\mathbf{P}}_{u\widetilde{\pi}_u(s)|\mathcal{Y}}.$$

Hence there must exist an element $\widetilde{\pi}_u(t_4) \mid \mathcal{Y}$ such that $t_4 < s$ and $\widetilde{\pi}_v(t_5) \mid \mathcal{Y} = \widetilde{\pi}_u(t_4) \mid \mathcal{Y}$ for $t_5 > s$. In that case, we must have

$$\widetilde{\mathbf{P}}_{u\widetilde{\pi}_u(t_3)|\mathcal{Y}} - \widetilde{\mathbf{P}}_{u\widetilde{\pi}_u(s)|\mathcal{Y}} \geq \widetilde{\mathbf{P}}_{u\widetilde{\pi}_u(t_3)|\mathcal{Y}} - \widetilde{\mathbf{P}}_{u\widetilde{\pi}_u(t_4)|\mathcal{Y}}$$
$$= \widetilde{\mathbf{P}}_{u\widetilde{\pi}_u(t_3)|\mathcal{Y}} - \widetilde{\mathbf{P}}_{v\widetilde{\pi}_u(t_3)|\mathcal{Y}}^{(\ell)} + \widetilde{\mathbf{P}}_{v\widetilde{\pi}_u(t_3)|\mathcal{Y}}^{(\ell)} - \widetilde{\mathbf{P}}_{v\widetilde{\pi}_v(t_4)|\mathcal{Y}}^{(\ell)} + \widetilde{\mathbf{P}}_{v\widetilde{\pi}_v(t_4)|\mathcal{Y}}^{(\ell)} - \widetilde{\mathbf{P}}_{u\widetilde{\pi}_u(t_4)|\mathcal{Y}}$$
$$\geq -4\Delta_{\ell+1} + \widetilde{\mathbf{P}}_{v\widetilde{\pi}_u(t_3)|\mathcal{Y}}^{(\ell)} - \widetilde{\mathbf{P}}_{v\widetilde{\pi}_v(t_4)|\mathcal{Y}}^{(\ell)} = -4\Delta_{\ell+1} + \widetilde{\mathbf{P}}_{v\widetilde{\pi}_v(s)|\mathcal{Y}}^{(\ell)} - \widetilde{\mathbf{P}}_{v\widetilde{\pi}_v(t_5)|\mathcal{Y}}^{(\ell)} \geq -4\Delta_{\ell+1}.$$

Therefore, in this case, we get that $\widetilde{\mathbf{P}}^{(\ell)}_{v\widetilde{\pi}_v(s)} - \widetilde{\mathbf{P}}^{(\ell)}_{v\pi_u(s)} \geq -8\Delta_{\ell+1}$. Hence we get

$$
\begin{aligned}
\widetilde{\mathbf{P}}^{(\ell)}_{v\widetilde{\pi}_v(1)|\mathcal{Y}} - \widetilde{\mathbf{P}}^{(\ell)}_{v\widetilde{\pi}_u(s)|\mathcal{Y}} &= \widetilde{\mathbf{P}}^{(\ell)}_{v\widetilde{\pi}_v(1)|\mathcal{Y}} - \widetilde{\mathbf{P}}^{(\ell)}_{v\widetilde{\pi}_v(s)|\mathcal{Y}} \\
&+ \widetilde{\mathbf{P}}^{(\ell)}_{v\widetilde{\pi}_v(s)|\mathcal{Y}} - \widetilde{\mathbf{P}}^{(\ell)}_{v\widetilde{\pi}_u(s)|\mathcal{Y}} \geq \widetilde{\mathbf{P}}^{(\ell)}_{v\widetilde{\pi}_v(1)|\mathcal{Y}} - \widetilde{\mathbf{P}}^{(\ell)}_{v\widetilde{\pi}_v(s)|\mathcal{Y}} - 8\Delta_{\ell+1}.
\end{aligned}
$$

Also, we will have that

$$
\begin{aligned}
\widetilde{\mathbf{P}}^{(\ell)}_{v\widetilde{\pi}_u(1)|\mathcal{Y}} - \widetilde{\mathbf{P}}^{(\ell)}_{v\widetilde{\pi}_v(1)|\mathcal{Y}} &= \widetilde{\mathbf{P}}^{(\ell)}_{v\widetilde{\pi}_u(1)|\mathcal{Y}} - \mathbf{P}^{(\ell)}_{u\widetilde{\pi}_u(1)|\mathcal{Y}} \\
&+ \widetilde{\mathbf{P}}^{(\ell)}_{u\widetilde{\pi}_u(1)|\mathcal{Y}} - \widetilde{\mathbf{P}}^{(\ell)}_{u\widetilde{\pi}_v(1)|\mathcal{Y}} + \mathbf{P}^{(\ell)}_{u\widetilde{\pi}_v(1)|\mathcal{Y}} - \widetilde{\mathbf{P}}^{(\ell)}_{v\widetilde{\pi}_v(1)|\mathcal{Y}} \geq -4\Delta_{\ell+1}.
\end{aligned}
$$

By combining the above arguments, we have that

$$
\widetilde{\mathbf{P}}^{(\ell)}_{u\widetilde{\pi}_u(1)|\mathcal{Y}} - \widetilde{\mathbf{P}}^{(\ell)}_{u\widetilde{\pi}_u(s)|\mathcal{Y}} \geq \widetilde{\mathbf{P}}^{(\ell)}_{v\widetilde{\pi}_u(1)|\mathcal{Y}} - \widetilde{\mathbf{P}}^{(\ell)}_{v\widetilde{\pi}_u(s)|\mathcal{Y}} - 4\Delta_{\ell+1} \geq \widetilde{\mathbf{P}}^{(\ell)}_{v\widetilde{\pi}_v(1)|\mathcal{Y}} - \widetilde{\mathbf{P}}^{(\ell)}_{v\widetilde{\pi}_v(s)|\mathcal{Y}} - 12\Delta_{\ell+1}.
$$

By a similar set of arguments involving triangle inequalities, we will also have

$$
\widetilde{\mathbf{P}}^{(\ell)}_{u\widetilde{\pi}_u(1)|\mathcal{Y}} - \widetilde{\mathbf{P}}^{(\ell)}_{u\widetilde{\pi}_u(s)|\mathcal{Y}} \leq \widetilde{\mathbf{P}}^{(\ell)}_{v\widetilde{\pi}_v(1)|\mathcal{Y}} - \widetilde{\mathbf{P}}^{(\ell)}_{v\widetilde{\pi}_v(s)|\mathcal{Y}} + 12\Delta_{\ell+1}.
$$

This completes the proof of the lemma. $\qquad\square$

Now, for any fixed subset $\mathcal{Y} \subseteq \mathcal{N}^{(\ell,i)}$, let us define the set $\mathcal{R}^{(\ell)}_u \mid \mathcal{Y}$ for user $u$ below:

$$
\mathcal{R}^{(\ell)}_u \mid \mathcal{Y} = \{j \in \mathcal{Y} \mid \widetilde{\mathbf{P}}_{uj} \geq \widetilde{\mathbf{P}}_{u\widetilde{\pi}_u(1)|\mathcal{Y}} - 2\Delta_{\ell+1}\}
$$

Hence, $\mathcal{R}^{(\ell)}_u \mid \mathcal{Y}$ corresponds to the set of items for user $u$ that is close to the item with the highest estimated reward for user $u$ restricted to the set $\mathcal{Y}$ at the end of the *explore* component of phase $\ell$.

**Lemma 11.** *Condition on the events $\mathcal{E}^{(\ell)}_2, \mathcal{E}^{(\ell)}_3$ being true. Consider a nice subset of users $\mathcal{M}^{(\ell,i)} \in \mathcal{M}'^{(\ell)}$ and their corresponding set of active items $\mathcal{N}^{(\ell,i)}$ for which guarantees in eq. 11 holds. Fix any subset $\mathcal{Y} \subseteq \mathcal{N}^{(\ell,i)}$. In that case, for every user $u \in \mathcal{M}^{(\ell,i)}$, the item with the highest reward $\pi_u(1) \mid \mathcal{Y}$ in the set $\mathcal{Y}$ must belong to the set $\mathcal{R}^{(\ell)}_u \mid \mathcal{Y}$. Moreover, $\max_{s,s' \in \mathcal{R}^{(\ell)}_u \mid \mathcal{Y}} |\mathbf{P}_{us} - \mathbf{P}_{us'}| \leq 4\Delta_{\ell+1}$.*

*Proof.* Let us fix a user $u \in \mathcal{M}^{(\ell,i)}$ with active set of arms $\mathcal{N}^{(\ell,i)}$. Now, we will have

$$
\begin{aligned}
\widetilde{\mathbf{P}}_{u\widetilde{\pi}_u(1)|\mathcal{Y}} - \widetilde{\mathbf{P}}^{(\ell)}_{u\pi_u(1)|\mathcal{Y}} &= \widetilde{\mathbf{P}}^{(\ell)}_{u\widetilde{\pi}_u(1)|\mathcal{Y}} \\
&- \mathbf{P}_{u\widetilde{\pi}_u(1)|\mathcal{Y}} + \mathbf{P}_{u\widetilde{\pi}_u(1)|\mathcal{Y}} - \mathbf{P}_{u\pi_u(1)|\mathcal{Y}} + \mathbf{P}_{u\pi_u(1)|\mathcal{Y}} - \widetilde{\mathbf{P}}^{(\ell)}_{u\pi_u(1)|\mathcal{Y}} \leq 2\Delta_{\ell+1}
\end{aligned}
$$

which implies that $\pi_u(1) \mid \mathcal{Y} \in \mathcal{R}^{(\ell)}_u \mid \mathcal{Y}$. Here we used the fact that $\widetilde{\mathbf{P}}^{(\ell)}_{u\widetilde{\pi}_u(1)|\mathcal{Y}} - \mathbf{P}_{u\widetilde{\pi}_u(1)|\mathcal{Y}} \leq \Delta_{\ell+1}$, $\mathbf{P}_{u|\mathcal{Y}} - \widetilde{\mathbf{P}}^{(\ell)}_{u\pi_u(1)|\mathcal{Y}} \leq \Delta_{\ell+1}$ and $\mathbf{P}_{u\widetilde{\pi}_u(1)|\mathcal{Y}} - \mathbf{P}_{u\pi_u(1)|\mathcal{Y}} \leq 0$. Next, notice that for any $s, s' \in \mathcal{R}^{(\ell)}_u \mid \mathcal{Y}$

$$
\mathbf{P}_{us} - \mathbf{P}_{us'} = \mathbf{P}_{us} - \widetilde{\mathbf{P}}^{(\ell)}_{us} + \widetilde{\mathbf{P}}^{(\ell)}_{us} - \widetilde{\mathbf{P}}^{(\ell)}_{ut_1} + \widetilde{\mathbf{P}}^{(\ell)}_{ut_1} - \widetilde{\mathbf{P}}^{(\ell)}_{us'} + \widetilde{\mathbf{P}}^{(\ell)}_{us'} - \mathbf{P}_{us'} \leq 4\Delta_{\ell+1}.
$$

$\qquad\square$

**Lemma 12.** *Condition on the events $\mathcal{E}^{(\ell)}_2, \mathcal{E}^{(\ell)}_3$ being true. Consider a nice subset of users $\mathcal{M}^{(\ell,i)} \in \mathcal{M}'^{(\ell)}$ and their corresponding set of active items $\mathcal{N}^{(\ell,i)}$ for which guarantees in eq. 11 holds. Fix any subset $\mathcal{Y} \subseteq \mathcal{N}^{(\ell,i)}$. Consider two users $u, v \in \mathcal{M}^{(\ell,i)}$ having an edge in the graph $\mathcal{G}^{(\ell,i)}$. Conditioned on the events $\mathcal{E}^{(\ell)}_2, \mathcal{E}^{(\ell)}_3$, we must have*

$$
\max_{x \in \mathcal{R}^{(\ell)}_u \mid \mathcal{Y}, y \in \mathcal{R}^{(\ell)}_v \mid \mathcal{Y}} |\mathbf{P}_{ux} - \mathbf{P}_{uy}| \leq 16\Delta_{\ell+1} \text{ and } \max_{x \in \mathcal{R}^{(\ell)}_u \mid \mathcal{Y}, y \in \mathcal{R}^{(\ell)}_v \mid \mathcal{Y}} |\mathbf{P}_{vx} - \mathbf{P}_{vy}| \leq 16\Delta_{\ell+1}
$$

*Proof.* From the construction of $\mathcal{G}^{(\ell,i)}$, we know that users $u, v \in \mathcal{M}^{(\ell,i)}$ have an edge if $\left|\widetilde{\mathbf{P}}^{(\ell)}_{ux} - \widetilde{\mathbf{P}}^{(\ell)}_{vx}\right| \leq 2\Delta_{\ell+1}$ (and therefore $|\mathbf{P}_{ux} - \mathbf{P}_{vx}| \leq \left|\widetilde{\mathbf{P}}^{(\ell)}_{ux} - \mathbf{P}_{ux}\right| + \left|\mathbf{P}_{vx} - \widetilde{\mathbf{P}}^{(\ell)}_{vx}\right| + \left|\widetilde{\mathbf{P}}^{(\ell)}_{ux} - \widetilde{\mathbf{P}}^{(\ell)}_{vx}\right| \leq 4\Delta_{\ell+1}$) for all $x \in \mathcal{N}^{(\ell,i)}$. For simplicity of notation, let us denote $\mathcal{R}_u$ to

be the set $\mathcal{R}_u^{(\ell)} \mid \mathcal{Y}$. Suppose $\mathbf{P}_{u\pi_u(1)\mid\mathcal{Y}} = a$ and $\mathbf{P}_{v\pi_v(1)\mid\mathcal{Y}} = b$. Consider any pair of items $x \in \mathcal{R}_u, y \in \mathcal{R}_v$ respectively. Therefore, for $x \in \mathcal{R}_u$ this must mean that $\mathbf{P}_{ux} \geq a - 4\Delta_{\ell+1}$ (note that $\pi_u(1) \mid \mathcal{Y} \in \mathcal{R}_u$). For $y \in \mathcal{R}_v$, we must similarly have $\mathbf{P}_{vy} \geq b - 4\Delta_{\ell+1}$. Since $u, v$ are connected by an edge, we must have $\mathbf{P}_{vx} \geq a - 8\Delta_{\ell+1}$ and $\mathbf{P}_{uy} \geq b - 8\Delta_{\ell+1}$. Now, we have $\mathbf{P}_{v\pi_v(1)\mid\mathcal{Y}} \geq \mathbf{P}_{vx}$ implying that $b \geq a - 8\Delta_{\ell+1}$; hence

$$\mathbf{P}_{ux} - \mathbf{P}_{uy} \leq \mathbf{P}_{u\pi_u(1)\mid\mathcal{R}_u} - \mathbf{P}_{uy} \leq a - (b - 8\Delta_{\ell+1}) \leq 16\Delta_{\ell+1}.$$

A similar analysis for $v$ shows that $\mathbf{P}_{vy} - \mathbf{P}_{vx} \leq 16\Delta_{\ell+1}$. This completes the proof of the lemma. $\square$

**Lemma 13.** *Condition on the events $\mathcal{E}_2^{(\ell)}, \mathcal{E}_3^{(\ell)}$ being true. Consider a nice subset of users $\mathcal{M}^{(\ell,i)} \in \mathcal{M}'^{(\ell)}$ and their corresponding set of active items $\mathcal{N}^{(\ell,i)}$ for which guarantees in eq. 11 holds. Fix any subset $\mathcal{Y} \subseteq \mathcal{N}^{(\ell,i)}$. Consider two users $u, v \in \mathcal{M}^{(\ell,i)}$ having a path of length $\mathsf{L}$ in the graph $\mathcal{G}^{(\ell,i)}$. Conditioned on the events $\mathcal{E}_2^{(\ell)}, \mathcal{E}_3^{(\ell)}$, we must have*

$$\max_{x \in \mathcal{R}_u^{(\ell)}\mid\mathcal{Y}, y \in \mathcal{R}_v^{(\ell)}\mid\mathcal{Y}} |\mathbf{P}_{ux} - \mathbf{P}_{uy}| \leq 8(\mathsf{L}+1)\Delta_{\ell+1}$$

$$\text{and} \quad \max_{x \in \mathcal{R}_u^{(\ell)}\mid\mathcal{Y}, y \in \mathcal{R}_v^{(\ell)}\mid\mathcal{Y}} |\mathbf{P}_{vx} - \mathbf{P}_{vy}| \leq 8(\mathsf{L}+1)\Delta_{\ell+1}$$

*Proof.* Recall from the construction of $\mathcal{G}^{(\ell,i)}$ that conditioned on $\mathcal{E}_3^{(\ell)}$ users $u, v \in \mathcal{M}^{(\ell,i)}$ have an edge if $\left|\widetilde{\mathbf{P}}_{ux}^{(\ell)} - \widetilde{\mathbf{P}}_{vx}^{(\ell)}\right| \leq 2\Delta_{\ell+1}$ (and therefore $|\mathbf{P}_{ux} - \mathbf{P}_{vx}| \leq 4\Delta_{\ell+1}$) for all $x \in \mathcal{N}^{(\ell,i)}$. Again, for simplicity of notation, let us denote $\mathcal{R}_u$ to be the set $\mathcal{R}_u^{(\ell)} \mid \mathcal{Y}$. Suppose $\mathbf{P}_{u\pi_u(1)\mid\mathcal{Y}} = a$ and $\mathbf{P}_{v\pi_v(1)\mid\mathcal{Y}} = b$. Consider any pair of items $x \in \mathcal{R}_u, y \in \mathcal{R}_v$ respectively. Therefore, for $x \in \mathcal{R}_u$ this must mean that $\mathbf{P}_{ux} \geq a - 4\Delta_{\ell+1}$. For $y \in \mathcal{R}_v$, we must similarly have $\mathbf{P}_{vy} \geq b - 4\Delta_{\ell+1}$. Since $u, v$ are connected by an path of length $\mathsf{L}$ (say $a_1, a_2, \ldots, a_{(\mathsf{L}-1)}$), we must have $\mathbf{P}_{a_1 x} \geq a - 8\Delta_{\ell+1}$, $\mathbf{P}_{a_2 x} \geq a - 12\Delta_{\ell+1}$ and finally $\mathbf{P}_{vx} \geq a - 4(\mathsf{L}+1)\Delta_{\ell+1}$. By a similar analysis $\mathbf{P}_{uy} \geq b - 4(\mathsf{L}+1)\Delta_{\ell+1}$. Now, we have $\mathbf{P}_{v\pi_v(1)\mid\mathcal{Y}} \geq \mathbf{P}_{vx}$ implying that $b \geq a - 4(\mathsf{L}+1)\Delta_{\ell+1}$; hence

$$\mathbf{P}_{ux} - \mathbf{P}_{uy} \leq \mathbf{P}_{u\pi_u(1)\mid\mathcal{R}_u} - \mathbf{P}_{uy} \leq a - (b - 4(\mathsf{L}+1)\Delta_{\ell+1}) \leq 8(\mathsf{L}+1)\Delta_{\ell+1}.$$

Again, a similar analysis for $v$ shows that $\mathbf{P}_{vy} - \mathbf{P}_{vx} \leq 8(\mathsf{L}+1)\Delta_{\ell+1}$. This completes the proof of the lemma. $\square$

**Corollary 2.** *Condition on the events $\mathcal{E}_2^{(\ell)}, \mathcal{E}_3^{(\ell)}$ being true. Consider a nice subset of users $\mathcal{M}^{(\ell,i)} \in \mathcal{M}'^{(\ell)}$ and their corresponding set of active items $\mathcal{N}^{(\ell,i)}$ for which guarantees in eq. 11 holds. Fix any subset $\mathcal{Y} \subseteq \mathcal{N}^{(\ell,i)}$. Consider two users $u, v \in \mathcal{M}^{(\ell,i)}$ having a path in the graph $\mathcal{G}^{(\ell,i)}$. Conditioned on the events $\mathcal{E}_2^{(\ell)}, \mathcal{E}_3^{(\ell)}$, we must have*

$$\max_{x \in \mathcal{R}_u^{(\ell)}\mid\mathcal{Y}, y \in \mathcal{R}_v^{(\ell)}\mid\mathcal{Y}} |\mathbf{P}_{ux} - \mathbf{P}_{uy}| \leq 16\mathsf{C}\Delta_{\ell+1} \text{ and } \max_{x \in \mathcal{R}_u^{(\ell)}\mid\mathcal{Y}, y \in \mathcal{R}_v^{(\ell)}\mid\mathcal{Y}} |\mathbf{P}_{vx} - \mathbf{P}_{vy}| \leq 16\mathsf{C}\Delta_{\ell+1}$$

*Proof.* The proof follows from the fact that any two users $u, v \in \mathcal{M}^{(\ell,i)}$ connected via a path must have a shortest path of length at most $2\mathsf{C} - 1$ conditioned on the events $\mathcal{E}_2^{\ell}, \mathcal{E}_3^{(\ell)}$. This is because, from Lemma 13, we know that users in the same cluster form a clique and since there are at most $\mathsf{C}$ clusters, the shortest path must be of length at most $2\mathsf{C} - 1$. $\square$

Hence, for a particular set of users $\mathcal{M}^{(\ell,i)} \in \mathcal{M}'^{(\ell)}$, consider the $j^{\text{th}}$ connected component of $\mathcal{G}^{(\ell,i)}$ comprising of users $\mathcal{M}^{(\ell,i,j)}$. If we consider the union of items $\mathcal{S} \equiv \cup_{u \in \mathcal{M}^{(\ell,i,j)}} \mathcal{R}_u^{(\ell)} \mid \mathcal{Y}$ for any subset $\mathcal{Y} \mid \mathcal{N}^{(\ell,i)}$, then from Corollary 2, we must have that for any user $u \in \mathcal{M}^{(\ell,i,j)}$,

$$\max_{x,y \in \mathcal{S}} |\mathbf{P}_{ux} - \mathbf{P}_{uy}| \leq 16\mathsf{C}\Delta_{\ell+1}. \tag{15}$$

This follows from the fact that every element $s \in \mathcal{S}$ must exist in $\mathcal{R}_v^{(\ell)} \mid \mathcal{Y}$ for some $v \in \mathcal{M}^{(\ell,i,j)}$; moreover, $v$ is connected to $u$ and therefore the shortest path joining them must be of length at most $2\mathsf{C} - 1$. Finally we use Corollary 2 to conclude equation 15.

**Lemma 14.** *Condition on the events $\mathcal{E}_2^{(\ell)}, \mathcal{E}_3^{(\ell)}$ being true. Consider a nice subset of users $\mathcal{M}^{(\ell,i)} \in \mathcal{M}'^{(\ell)}$ and their corresponding set of active items $\mathcal{N}^{(\ell,i)}$ for which guarantees in eq. 11 holds. Consider the $j^{\text{th}}$ connected component of the graph $\mathcal{G}^{(\ell,i)}$ comprising of users $\mathcal{M}^{(\ell,i,j)}$. Let $\mathcal{N}^{(\ell,i,j)} \equiv \cup_{u \in \mathcal{M}^{(\ell,i,j)}} \mathcal{T}_u^{(\ell)}$ denote the union of good items for users in $\mathcal{M}^{(\ell,i,j)}$. In that case,*

1. *For every user $u \in \mathcal{M}^{(\ell,i,j)}$, the items $\pi_u(s) \mid [N] \backslash \mathcal{O}_{\mathcal{M}^{(\ell,i)}}^{(\ell)}$ for $s \in [\mathsf{TB}^{-1} - \left| \mathcal{O}_{\mathcal{M}^{(\ell,i)}}^{(\ell)} \right|]$ must belong to the set $\mathcal{N}^{(\ell,i,j)}$ i.e. the golden items that have not been chosen for recommendation to users in $\mathcal{M}^{(\ell,i,j)}$ in exploit components of previous phases must belong to the surviving set of items $\mathcal{N}^{(\ell,i,j)}$.*

2. *For any subset $\mathcal{Y} \subseteq \mathcal{N}^{(\ell,i,j)}$ and any $s \le |\mathcal{Y}|$, we must have the following for any $\mathsf{A} > 0$:*

$$\text{If } \widetilde{\mathbf{P}}_{u\widetilde{\pi}_u(1)|\mathcal{Y}} - \widetilde{\mathbf{P}}_{u\widetilde{\pi}_u(s)|\mathcal{Y}} \ge \mathsf{A} \text{ for some } u \in \mathcal{M}^{(\ell,i,j)}$$

$$\text{then } \mathbf{P}_{v\pi_v(1)|\mathcal{Y}} - \mathbf{P}_{v\pi_v(s)|\mathcal{Y}}$$

$$\ge \widetilde{\mathbf{P}}_{v\widetilde{\pi}_v(1)|\mathcal{Y}} - \widetilde{\mathbf{P}}_{v\widetilde{\pi}_v(s)|\mathcal{Y}} - 4\Delta_{\ell+1} \ge \mathsf{A} - (2\mathsf{C} - 1)4\Delta_{\ell+1} \text{ for all } v \in \mathcal{M}^{(\ell,i,j)}.$$

*Proof.* The proof of the first part follows directly from Lemma 11 where we showed that for a particular user $u \in \mathcal{M}^{(\ell,i)}$, the items $\mathbf{P}_{u\pi_u(s)|[N] \backslash \mathcal{O}_{\mathcal{M}^{(\ell,i)}}^{(\ell)}}$ for $s \in [\mathsf{TB}^{-1} - \left| \mathcal{O}_{\mathcal{M}^{(\ell,i)}}^{(\ell)} \right|]$ must belong to the set $\mathcal{T}_u^{(\ell)}$ (and the fact that $\mathcal{N}^{(\ell,i,j)} \supseteq \mathcal{T}_u^{(\ell)}$).

We move on to the proof of the second part of the lemma. Recall that any two users $u, v \in \mathcal{M}^{(\ell,i)}$ have an edge in the graph $\mathcal{G}^{(\ell,i)}$ if $\left| \widetilde{\mathbf{P}}_{ux}^{(\ell)} - \widetilde{\mathbf{P}}_{vx}^{(\ell)} \right| \le 2\Delta_{\ell+1}$ for all $x \in \mathcal{N}^{(\ell,i)}$. In that case, we have

$$\widetilde{\mathbf{P}}_{v\widetilde{\pi}_v(1)|\mathcal{Y}} - \widetilde{\mathbf{P}}_{v\widetilde{\pi}_v(s)|\mathcal{Y}} \ge \widetilde{\mathbf{P}}_{v\widetilde{\pi}_u(1)|\mathcal{Y}} - \widetilde{\mathbf{P}}_{v\widetilde{\pi}_u(s)|\mathcal{Y}}$$

$$\ge \widetilde{\mathbf{P}}_{u\widetilde{\pi}_u(1)|\mathcal{Y}} - \widetilde{\mathbf{P}}_{u\widetilde{\pi}_u(s)|\mathcal{Y}} - 4\Delta_{\ell+1} \ge \mathsf{A} - 4\Delta_{\ell+1}.$$

We are going to prove the Lemma statement by induction on length of the shortest path joining the users $u, v$. The base case (when the path length is 1 i.e. $u, v$ are joined by an edge) is proved above. Suppose the statement is true when length of the shortest path is $\mathsf{L} - 1$. In that case, we have the following set of inequalities (suppose $w$ is the neighbor of $v$ and the length of the shortest path joining $u, w$ is $\mathsf{L} - 1$)

$$\widetilde{\mathbf{P}}_{v\widetilde{\pi}_v(1)|\mathcal{Y}} - \widetilde{\mathbf{P}}_{v\widetilde{\pi}_v(s)|\mathcal{Y}} \ge \widetilde{\mathbf{P}}_{v\widetilde{\pi}_w(1)|\mathcal{Y}} - \widetilde{\mathbf{P}}_{v\widetilde{\pi}_w(s)|\mathcal{Y}}$$

$$\ge \widetilde{\mathbf{P}}_{w\widetilde{\pi}_w(1)|\mathcal{Y}} - \widetilde{\mathbf{P}}_{w\widetilde{\pi}_w(s)|\mathcal{Y}} - 4\mathsf{L}\Delta_{\ell+1} \ge \mathsf{A} - 4\mathsf{L}\Delta_{\ell+1}.$$

The lemma statement follows from the fact that the length of the shortest path between users $u, v$ in the same connected component is at most $2\mathsf{C} - 1$. This completes the proof of the lemma. $\square$

**Partition of $\mathcal{M}^{(\ell,i)}$ in phase $\ell + 1$ into nice subsets of users and their corresponding active subsets of items:** Consider a nice set of users $\mathcal{M}^{(\ell,i)}$ at the end of the *explore* component of phase $\ell$ i.e. the start of the subsequent phase $\ell + 1$ for the users in $\mathcal{M}^{(\ell,i)}$. At this point, conditioned on events $\mathcal{E}_2^{(\ell)}, \mathcal{E}_3^{(\ell)}$, our goal is to further partition the users in $\mathcal{M}^{(\ell,i)}$ into more nuanced *nice* subsets. Suppose the number of rounds is at least $\mathsf{T}^{1/3}/\mathsf{B} = \widetilde{O}(\mathsf{T}^{1/3})$ implying that the active set of items $\mathcal{N}^{(\ell,i)}$ is of size at least $\widetilde{\Omega}(\mathsf{T}^{1/3})$ (induction assumption property B). Suppose, we index the connected components of the graph $\mathcal{G}^{(\ell,i)}$ formed by the users in $\mathcal{M}^{(\ell,i)} \in \mathcal{M}'^{(\ell)}$. Now, for each index $j$, the set of users corresponding to the $j^{\text{th}}$ connected component of the graph $\{\mathcal{G}^{(\ell,i)}\}$ forms the $j^{\text{th}}$ nice subset of users (Lemma 7) stemming from users in $\mathcal{M}^{(\ell,i)} \in \mathcal{M}'^{(\ell)}$ - let us denote this set of users by $\mathcal{M}^{(\ell+1,z)}$ for some index $s > 0$. For this set of users $\mathcal{M}^{(\ell+1,z)}$, at the start of the *exploit* component of phase $\ell + 1$ (round $t$), we define the active set of items $\mathcal{N}^{(\ell+1,t,z)}$ to be $\bigcup_{u \in \mathcal{M}^{(\ell+1,z)}} \mathcal{T}_u^{(\ell)}$. Hence $\{(\mathcal{M}^{(\ell+1,z)}, \mathcal{N}^{(\ell+1,t,z)}\}_z$ forms the family of nice sets of users (that progress to the $(\ell + 1)^{\text{th}}$ phase) and their corresponding active set of items at the beginning (*exploit* component) of phase $\ell + 1$ for users stemming from $\mathcal{M}'^{(\ell)}$. Next, we discuss our recommendation strategy for $\mathcal{M}^{(\ell+1,z)}$ (a *nice* subset of users) in the *exploit* component of phase $\ell + 1$.

**Strategy in exploit component of phase $\ell+1$:** Note that in the exploit component of phase $\ell+1$ for users in $\mathcal{M}^{(\ell,i,j)} \equiv \mathcal{M}^{(\ell+1,z)}$ (new notation indicating that $\mathcal{M}^{(\ell,i,j)} \equiv \mathcal{M}^{(\ell+1,z)}$ is a *nice* subset of users at phase $\ell+1$) for some indices $i,j$ and $z$, we follow a recursive approach to identify and recommend items in $\{\pi_u(t)\}_{t=1}^{\mathsf{TB}^{-1}}$ for all users $u \in \mathcal{M}^{(\ell+1,z)}$. At the beginning of the *exploit* component of phase $\ell+1$ (say round $t$), we will also initialize $\mathcal{O}_{\mathcal{M}^{(\ell+1,z)}}^{(\ell+1,t)}$ to be the set of items $\mathcal{O}_{\mathcal{M}^{(\ell,i)}}^{(\ell)}$ (i.e. the *golden* items that have been chosen for recommendation in the *exploit* components of previous phases $(1-\ell)$ to users in $\mathcal{M}^{(\ell,i)}$ and recommended sufficiently enough number of times to be blocked). At start of the *exploit* component of phase $\ell+1$ (round $t$), recall that the active set of items is given by $\mathcal{N}^{(\ell+1,t,z)} \equiv \bigcup_{u \in \mathcal{M}^{(\ell+1,z)}} \mathcal{T}_u^{(\ell)}$. We reiterate here that $t$ corresponds to the index of the starting round in the *exploit* component of phase $\ell+1$ for users in the *nice subset* $\mathcal{M}^{(\ell+1,z)}$.

Let us denote $s = \mathsf{TB}^{-1} - \left|\mathcal{O}_{\mathcal{M}^{(\ell+1,z)}}^{(\ell+1,t)}\right|$ and $\mathcal{Y} = \mathcal{N}^{(\ell+1,t,z)}$. First of all, note that from Lemma 14, for all users $u \in \mathcal{M}^{(\ell+1,z)}$ the items $\pi_u(r) \mid [\mathsf{N}] \setminus \mathcal{O}_{\mathcal{M}^{(\ell+1,z)}}^{(\ell+1,t)}$ for $r \in [s]$ must belong to the set $\mathcal{Y}$ i.e. the best $s$ items among those that are not in the set $\mathcal{O}_{\mathcal{M}^{(\ell+1,z)}}^{(\ell+1,t)}$ must survive in $\mathcal{Y}$ (Lemma 14). Now, we look at two possibilities:

1. *(Possibility A):* For all users $u \in \mathcal{M}^{(\ell+1,z)}$, we have that $\widetilde{\mathbf{P}}_{u\widetilde{\pi}_u(1)|\mathcal{Y}} - \widetilde{\mathbf{P}}_{u\widetilde{\pi}_u(s)|\mathcal{Y}} \le 64\mathsf{C}\Delta_{\ell+1}$ In that case, we stop the *exploit* component of phase $\ell+1$ and move on to the *explore* component of phase $\ell+1$ for users in $\mathcal{M}^{(\ell+1,z)}$ and active items $\mathcal{Y}$. Conditioned on events $\mathcal{E}_2^{(\ell)}, \mathcal{E}_3^{(\ell)}$, in Lemma 15, we show that the above condition implies for every user $u \in \mathcal{M}^{(\ell+1,z)}$, we must have

$$\max_{x,y \in \mathcal{Y}} \le 88\mathsf{C}\Delta_{\ell+1}.$$

   Furthermore, for $\mathcal{Z} = [\mathsf{N}] \setminus \mathcal{O}_{\mathcal{M}^{(\ell+1,z)}}^{(\ell+1)}$, it must happen that $\mathcal{Y} \supseteq \{\pi_u(s) \mid \mathcal{Z}\}_{s=1}^{\mathsf{T}/\mathsf{B} - \left|\mathcal{O}_{\mathcal{M}^{(\ell+1,z)}}^{(\ell+1)}\right|}$ for every user $u \in \mathcal{M}^{(\ell+1,z)}$. In other words, for each user $u \in \mathcal{M}^{(\ell+1,z)}$, the set $\mathcal{Y}$ must contain all the top $\mathsf{TB}^{-1}$ golden items ( $\{\pi_u(r)\}_{r=1}^{\mathsf{TB}^{-1}}$ ) that were not recommended in the *exploit* components so far.

2. *(Possibility B):* For some user $u \in \mathcal{M}^{(\ell+1,z)}$, we have that $\widetilde{\mathbf{P}}_{u\widetilde{\pi}_u(1)|\mathcal{Y}} - \widetilde{\mathbf{P}}_{u\widetilde{\pi}_u(s)|\mathcal{Y}} \ge 64\mathsf{C}\Delta_{\ell+1}$. In that case, from Lemma 14, we know that for every user $v \in \mathcal{M}^{(\ell+1,z)}$, we must have $\mathbf{P}_{v\pi_v(1)|\mathcal{Y}} - \mathbf{P}_{v\pi_v(s)|\mathcal{Y}} \ge 56\mathsf{C}\Delta_{\ell+1}$. In that case, if we consider the set of items $\mathcal{S} \equiv \cup_{u \in \mathcal{M}^{(\ell+1,z)}} \mathcal{R}_u^{(\ell)} \mid \mathcal{Y}$, then from Lemma 13 (or see eq. 15), we must have that for every user $u \in \mathcal{M}^{(\ell+1,z)}$,

$$\max_{x,y \in \mathcal{S}} |\mathbf{P}_{ux} - \mathbf{P}_{uy}| \le 16\mathsf{C}\Delta_{\ell+1} \tag{16}$$

   For every user $u \in \mathcal{M}^{(\ell+1,z)}$, recall that in Lemma 11, we showed that $\pi_u(1) \mid \mathcal{Y} \in \mathcal{R}_u^{(\ell)} \mid \mathcal{Y}$ and in Lemma 14, we showed that $\pi_u(1) \mid \mathcal{Y} = \pi_u(1) \mid [\mathsf{N} \setminus \left|\mathcal{O}_{\mathcal{M}^{(\ell+1,z)}}^{(\ell+1,t)}\right|]$. Hence, $\pi_u(1) \mid \mathcal{Y} = \pi_u(1) \mid [\mathsf{N} \setminus \left|\mathcal{O}_{\mathcal{M}^{(\ell+1,z)}}^{(\ell+1,t)}\right|]$ belongs to the set $\mathcal{S}$ for every user $u \in \mathcal{M}^{(\ell+1,z)}$. Hence we must have that the items $\mathcal{S}$ is a subset of $\{\pi_u(r) \mid [\mathsf{N}] \setminus \mathcal{O}_{\mathcal{M}^{(\ell+1,z)}}^{(\ell+1,t)}\}_{r=1}^{s}$. Suppose we index the items in $\mathcal{S}$. For each of the subsequent $\mathsf{B}|\mathcal{S}|$ rounds (indexed by $b \in [\mathsf{B}\mathcal{S}]$), for every user $u \in \mathcal{M}^{(\ell+1,z)}$, we go to the $\lceil(b/\mathsf{B})\rceil^{\text{th}}$ item in $\mathcal{S}$ and recommend it to user $u$ if unblocked. On the other hand, if the $\lceil(b/\mathsf{B})\rceil^{\text{th}}$ item in $\mathcal{S}$ is blocked (or becomes blocked) for the user $u \in \mathcal{M}^{(\ell+1,z)}$, then we simply recommend any unblocked item in $\mathcal{N}^{(\ell+1,t,z)}$. This is always possible because we will prove via induction (see Lemma 15 and in particular eq. 23) that at every round in the *exploit* component of phase $\ell+1$, the number of unblocked items for any user $u \in \mathcal{M}^{(\ell+1,z)}$ in the set $\mathcal{N}^{(\ell+1,t,z)}$ (where $t$ is the previous decision round for $\mathcal{M}^{(\ell+1,z)}$ on whether possibility A or B is true) is always larger than the number

of remaining rounds. We make the following updates:

$$\mathcal{O}^{(\ell+1,t+|\mathcal{S}|)}_{\mathcal{M}^{(\ell+1,z)}} \leftarrow \mathcal{O}^{(\ell+1,t)}_{\mathcal{M}^{(\ell+1,z)}} \cup \mathcal{S} \tag{17}$$

$$\mathcal{N}^{(\ell+1,t+|\mathcal{S}|,z)} \leftarrow \mathcal{N}^{(\ell+1,t,z)} \setminus \mathcal{O}^{(\ell+1,t+|\mathcal{S}|)}_{\mathcal{M}^{(\ell+1,z)}} \tag{18}$$

$$t \leftarrow t + |\mathcal{S}| \text{ and } \mathcal{Y} \leftarrow \mathcal{N}^{(\ell+1,t,s)} \tag{19}$$

i.e we update the set $\mathcal{O}^{(\ell+1,t)}_{\mathcal{M}^{(\ell+1,z)}}$ by taking union with the set of $|\mathcal{S}|$ identified items in $\{\pi_u(t)\}_{t=1}^{\mathsf{T}}$ for all users $u \in \mathcal{M}^{(\ell+1,z)}$. After these $|\mathcal{S}|$ rounds, for the set of users $\mathcal{M}^{(\ell+1,z)}$, the set of active items $\mathcal{N}^{(\ell+1,t,z)}$ is pruned by removing the items in $\mathcal{S}$ and the time index is increased from $t$ to $t + |\mathcal{S}|$.

At this point, we repeat the same process again for users in $\mathcal{M}^{(\ell+1,z)}$ with the pruned set of active items $\mathcal{N}^{(\ell+1,t,s)}$ i.e. we check for *possibility A or possibility B*. If we encounter possibility $B$, then we again find the set of items $\mathcal{S} \equiv \cup_{u \in \mathcal{M}^{(\ell+1,z)}} \mathcal{R}_u^{(\ell)} \mid \mathcal{Y}$ and recommended it to all users in $\mathcal{M}$ in $|\mathcal{S}|$ B steps as outlined above. We do this step recursively until we encounter Step A for the users in $\mathcal{M}^{(\ell+1,z)}$ and at that point we exit the *exploit* component of phase $\ell + 1$ and enter the *explore* component of phase $\ell + 1$.

As before, at the beginning of the *explore* component of phase $\ell + 1$ for the *nice* subset of users $\mathcal{M}^{(\ell+1,z)}$, let us denote the set of active items by $\mathcal{N}^{(\ell+1,z)}$ and the set of items considered for recommendation in the *exploit* phases including the $(\ell + 1)^{\text{th}}$ one by $\mathcal{O}^{(\ell+1)}_{\mathcal{M}^{(\ell+1,z)}}$ (i.e. we remove the $t$ in the superscript for simplicity). Therefore, at the end of the *explore* component of phase $\ell + 1$ for the *nice* subset of users $\mathcal{M}^{(\ell+1,z)}$, the set of active items $\mathcal{N}^{(\ell+1,z)}$ satisfy the following:

**Lemma 15.** *Consider a nice subset of users $\mathcal{M}^{(\ell+1,z)}$ and their corresponding set of active items $\mathcal{N}^{(\ell+1,z)}$ at the end of the exploit stage of phase $\ell + 1$ i.e. for all users $u \in \mathcal{M}^{(\ell+1,z)}$, we have $\widetilde{\mathbf{P}}_{u\widetilde{\pi}_u(1)|\mathcal{N}^{(\ell+1,z)}} - \widetilde{\mathbf{P}}_{u\widetilde{\pi}_u(s)|\mathcal{N}^{(\ell+1,z)}} \le 64\mathsf{C}\Delta_{\ell+1}$ for $s = \mathsf{TB}^{-1} - \left|\mathcal{O}^{(\ell+1)}_{\mathcal{M}^{(\ell+1,z)}}\right|$. Suppose $\mathcal{M}^{(\ell+1,z)}$ is comprised of the users in a connected component of the graph $\mathcal{G}^{(\ell,i)}$ which in turn is formed by the users in $\mathcal{M}^{(\ell,i)}$ for which guarantees in eq. 11 holds true i.e. we condition on the events $\mathcal{E}_2^{(\ell)}, \mathcal{E}_3^{(\ell)}$ being true. In that case, we must have that*

1. *for all users $u \in \mathcal{M}^{(\ell+1,z)}$,*

$$\max_{x,y \in \mathcal{N}^{(\ell+1,z)}} |\mathbf{P}_{ux} - \mathbf{P}_{uy}| \le 88\mathsf{C}\Delta_{\ell+1}$$

   *i.e. the best and worst items in the set $\mathcal{N}^{(\ell+1,z)}$ for any user $u \in \mathcal{M}^{(\ell+1,z)}$ has close rewards.*

2. *Denote $\mathcal{Z} = [\mathsf{N}] \setminus \mathcal{O}^{(\ell+1)}_{\mathcal{M}^{(\ell+1,z)}}$ to be set of items not chosen for recommendation to users in $\mathcal{M}^{(\ell+1,z)}$ in the exploit component of phases until (and including) phase $\ell + 1$. Then it must happen that*

$$\mathcal{O}^{(\ell+1)}_{\mathcal{M}^{(\ell+1,z)}} \subseteq \{\pi_u(t)\}_{t=1}^{\mathsf{TB}^{-1}} \tag{20}$$

$$\mathcal{N}^{(\ell+1,z)} \supseteq \bigcup_{u \in \mathcal{M}^{(\ell+1,z)}} \{\pi_u(t') \mid \mathcal{Z}\}_{t'=1}^{\mathsf{TB}^{-1} - \left|\mathcal{O}^{(\ell+1)}_{\mathcal{M}^{(\ell+1,z)}}\right|}. \tag{21}$$

*Proof.* Suppose at the end of the *explore* component of phase $\ell$, $\mathcal{M}^{(\ell+1,z)} \subseteq \mathcal{M}^{(\ell,i)}$. We will prove a more general statement. Consider the rounds $t_1, t_2, \dots$ at which we check for *possibility A or possibility B* (this includes the starting and ending rounds of the *exploit* component of phase $\ell + 1$). At any such round $t_r$, for all users $u \in \mathcal{M}^{(\ell+1,z)}$, we must have that $\widetilde{\pi}_u(\mathsf{TB}^{-1} - \left|\mathcal{O}^{(\ell,t_r)}_{\mathcal{M}^{(\ell,i)}}\right|) \mid \mathcal{N}^{(\ell,i)} \in \mathcal{N}^{(\ell+1,t_r,z)}$ and

$$\max_{x,y \in \mathcal{N}^{(\ell+1,t_r,z)}} |\mathbf{P}_{ux} - \mathbf{P}_{uy}| \le \max_{v \in \mathcal{M}^{(\ell+1,z)}} \left(\widetilde{\mathbf{P}}_{u\widetilde{\pi}_u(1)|\mathcal{N}^{(\ell+1,t_r,z)}} - \widetilde{\mathbf{P}}_{u\widetilde{\pi}_u(s)|\mathcal{N}^{(\ell+1,t_r,z)}}\right) + 24\mathsf{C}\Delta_{\ell+1} \tag{22}$$

$$\mathcal{N}^{(\ell+1,t_r,z)} \supseteq \bigcup_{u \in \mathcal{M}^{(\ell+1,z)}} \{\pi_u(t') \mid \mathcal{Z}\}_{t'=1}^{s} \text{ where } \mathcal{Z} \equiv [\mathsf{N}] \setminus \mathcal{O}^{(\ell+1,t_r)}_{\mathcal{M}^{(\ell+1,z)}} \tag{23}$$

$$\mathcal{O}^{(\ell+1,t_r)}_{\mathcal{M}^{(\ell+1,z)}} \subseteq \{\pi_u(t)\}_{t=1}^{\mathsf{TB}^{-1}} \tag{24}$$

for $s = \mathsf{T} - \left|\mathcal{O}_{\mathcal{M}^{(\ell+1,z)}}^{(\ell+1,t_r)}\right|$ We will prove the statement above via induction on the recursions performed in the *exploit* component of the phase $\ell + 1$. For the base case, we consider the round $t_1$ which corresponds to the beginning of the *exploit* component of phase $\ell + 1$. At this round, recall that $\mathcal{N}^{(\ell+1,t_1,z)} = \cup_{u \in \mathcal{M}^{(\ell+1,z)}} \mathcal{T}_u^{(\ell)}$. Of course, with $s = \mathsf{T} - \left|\mathcal{O}_{\mathcal{M}^{(\ell,i)}}^{(\ell)}\right|$, for every user $u \in \mathcal{M}^{(\ell+1,z)}$, we must have $\{\widetilde{\pi}_u(r) \mid \mathcal{N}^{(\ell,i)}\}_{r=1}^{s} \in \mathcal{T}_u^{(\ell)}$ (Lemma 11) and since $\mathcal{N}^{(\ell+1,z)}$ is a union of the sets $\mathcal{T}_u^{(\ell)}$, $\widetilde{\pi}_u(r) \mid \mathcal{N}^{(\ell,i)} = \widetilde{\pi}_u(r) \mid \mathcal{N}^{(\ell+1,z)}$ for all $r \in [s]$. Hence by invoking Lemma 9 (in the statement of Lemma 9, we have $\mathcal{J} = \phi$ i.e. $\mathcal{Y} = \mathcal{N}^{(\ell+1,z)}$ and $\widetilde{\pi}_u(s) \mid \mathcal{N}^{(\ell,i)} = \widetilde{\pi}_u(s) \mid \mathcal{N}^{(\ell+1,z)}$ for $s = \mathsf{T} - \left|\mathcal{O}_{\mathcal{M}^{(\ell,i)}}^{(\ell,t_r)}\right|$), we obtain the statement of the Lemma for round $t_1$.

Suppose the induction statement is true for round $t_a$ and the second possibility i.e. possibility $B$ became true. In that case, we do not exit the recursion and our goal is to show that the lemma statement is true at the next decision round $t_{a+1}$. The induction hypothesis implies that for every user $u \in \mathcal{M}^{(\ell+1,z)}$, we have that $\widetilde{\pi}_u(\mathsf{TB}^{-1} - \left|\mathcal{O}_{\mathcal{M}^{(\ell,i)}}^{(\ell)}\right|)$ survives in the set of items $\mathcal{N}^{(\ell+1,t_a,z)}$ and furthermore, we have $\widetilde{\pi}_u(\mathsf{TB}^{-1} - \left|\mathcal{O}_{\mathcal{M}^{(\ell,i)}}^{(\ell)}\right|) \mid \mathcal{N}^{(\ell,i)} = \widetilde{\pi}_u(s) \mid \mathcal{N}^{(\ell+1,t_a,z)}$ for $s = \mathsf{TB}^{-1} - \left|\mathcal{O}_{\mathcal{M}^{(\ell+1,z)}}^{(\ell+1,t_a)}\right|$ (note that $s$ is common for all users in $\mathcal{M}^{(\ell+1,z)}$). The induction hypothesis also implies that

$$\mathcal{N}^{(\ell+1,t_r,z)} \supseteq \bigcup_{u \in \mathcal{M}^{(\ell+1,z)}} \{\pi_u(t') \mid \mathcal{Z}\}_{t'=1}^{s} \text{ where } \mathcal{Z} \equiv [\mathsf{N}] \setminus \mathcal{O}_{\mathcal{M}^{(\ell+1,z)}}^{(\ell+1,t_a)} \text{ and } s = \mathsf{T} - \left|\mathcal{O}_{\mathcal{M}^{(\ell+1,z)}}^{(\ell+1,t_a)}\right|$$

(25)

$$\mathcal{O}_{\mathcal{M}^{(\ell+1,z)}}^{(\ell+1,t_r)} \subseteq \{\pi_u(t)\}_{t=1}^{\mathsf{TB}^{-1}}$$

(26)

Again, at the decision round $t_a$, since the possibility $B$ was true, for one of the users $u \in \mathcal{M}^{(\ell+1,z)}$, we must have for $s = \mathsf{T} - \left|\mathcal{O}_{\mathcal{M}^{(\ell+1,z)}}^{(\ell+1,t_{a+1})}\right|$,

$$\widetilde{\mathbf{P}}_{u\widetilde{\pi}_u(1)|\mathcal{N}^{(\ell+1,t_a,z)}} - \widetilde{\mathbf{P}}_{u\widetilde{\pi}_u(s)|\mathcal{N}^{(\ell+1,t_a,z)}} \geq 64\mathsf{C}\Delta_{\ell+1}$$

implying that for every user $v \in \mathcal{M}^{(\ell+1,z)}$, we have

$$\widetilde{\mathbf{P}}_{v\widetilde{\pi}_v(1)|\mathcal{N}^{(\ell+1,t_a,z)}} - \widetilde{\mathbf{P}}_{v\widetilde{\pi}_v(s)|\mathcal{N}^{(\ell+1,t_a,z)}} \geq 56\mathsf{C}\Delta_{\ell+1}.$$

The above equation further implies that (Lemma 8), for every user $v \in \mathcal{M}^{(\ell+1,z)}$, we will have

$$\mathbf{P}_{v\pi_v(1)|\mathcal{N}^{(\ell+1,t_a,z)}} - \mathbf{P}_{v\pi_v(s)|\mathcal{N}^{(\ell+1,t_a,z)}} \geq 50\mathsf{C}\Delta_{\ell+1}.$$

Hence, as mentioned before, if we consider the set of items $\mathcal{S} \equiv \cup_{u \in \mathcal{M}^{(\ell+1,z)}} \mathcal{R}_u^{(\ell)} \mid \mathcal{N}^{(\ell+1,t_a,z)}$, then from Lemma 13 (or see eq. 15), we must have that for every user $u \in \mathcal{M}^{(\ell+1,z)}$, $\pi_u(1) \mid \mathcal{N}^{(\ell+1,t_a,z)} \in \mathcal{S}$ and

$$\max_{y \in \mathcal{S}} \left|\mathbf{P}_{u\pi_u(1)|\mathcal{N}^{(\ell+1,t_a,z)}} - \mathbf{P}_{uy}\right| \leq 16\mathsf{C}\Delta_{\ell+1}.$$

(27)

Hence, if we remove the set $\mathcal{S}$ to update $\mathcal{N}^{(\ell+1,t_a,z)}$ i.e. $\mathcal{N}^{(\ell+1,t_{a+1},z)} \leftarrow \mathcal{N}^{(\ell+1,t_a,z)} \setminus \mathcal{S}$, for every user $u \in \mathcal{M}^{(\ell+1,z)}$, we must have

$$\widetilde{\pi}_u(\mathsf{TB}^{-1} - \left|\mathcal{O}_{\mathcal{M}^{(\ell,i)}}^{(\ell)}\right|) \mid \mathcal{N}^{(\ell,i)} = \widetilde{\pi}_u(\mathsf{TB}^{-1} - \left|\mathcal{O}_{\mathcal{M}^{(\ell+1,z)}}^{(\ell+1,t_a)}\right|) \mid \mathcal{N}^{(\ell+1,t_a,z)}$$

$$= \widetilde{\pi}_u(\mathsf{TB}^{-1} - \left|\mathcal{O}_{\mathcal{M}^{(\ell+1,z)}}^{(\ell+1,t_{a+1})}\right|) \mid \mathcal{N}^{(\ell+1,t_{a+1},z)}.$$

This is because for each user $u$, only elements which have larger rewards than $\widetilde{\pi}_u(\mathsf{TB}^{-1} - \left|\mathcal{O}_{\mathcal{M}^{(\ell+1,z)}}^{(\ell+1,t_a)}\right|) \mid \mathcal{N}^{(\ell+1,t_a,z)}$ are removed which makes the aforementioned item survive in $\mathcal{N}^{(\ell+1,t_{a+1},z)}$ and also moves its position up (recall that $\mathcal{O}_{\mathcal{M}^{(\ell+1,z)}}^{(\ell+1,t_{a+1})} \leftarrow \mathcal{O}_{\mathcal{M}^{(\ell+1,z)}}^{(\ell+1,t_a)} \cup \mathcal{S}$) in the list of surviving items sorted in decreasing order by expected reward. Hence, we can apply Lemma

9 to conclude the first part of the induction proof. In order to show the final statement, with $s = \mathsf{TB}^{-1} - \left|\mathcal{O}^{(\ell+1,t_r)}_{\mathcal{M}^{(\ell+1,z)}}\right|$, we can simply substitute that

$$\max_{v \in \mathcal{M}^{(\ell+1,z)}} \left(\widetilde{\mathbf{P}}_{u\widetilde{\pi}_u(1)|\mathcal{N}^{(\ell+1,t_r,z)}} - \widetilde{\mathbf{P}}_{u\widetilde{\pi}_u(s)|\mathcal{N}^{(\ell+1,t_r,z)}}\right) \leq 64\mathsf{C}\Delta_{\ell+1}$$

when possibility $A$ became true at a decision round $t_r$ and we exit the *exploit* component to enter the *explore* component of phase $\ell + 1$ for users in $\mathcal{M}^{(\ell+1,z)}$.

Moreover, we will also have that

$$\mathcal{S} \subseteq \{\pi_u(r) \mid \mathcal{N}^{(\ell+1,t_a,z)}\}^s_{r=1} \text{ where } s = \mathsf{TB}^{-1} - \left|\mathcal{O}^{(\ell+1,t_a)}_{\mathcal{M}^{(\ell+1,z)}}\right| \tag{28}$$

$$\implies \mathcal{S} \subseteq \{\pi_u(r) \mid \mathcal{Z}\}^s_{r=1} \text{ where } \mathcal{Z} \equiv [\mathsf{N}] \setminus \mathcal{O}^{(\ell+1,t_a)}_{\mathcal{M}^{(\ell+1,z)}} \text{ and } s = \mathsf{T} - \left|\mathcal{O}^{(\ell+1,t_a)}_{\mathcal{M}^{(\ell+1,z)}}\right| \tag{29}$$

$$\implies \mathcal{S} \subseteq \{\pi_u(t)\}^{\mathsf{TB}^{-1}}_{t=1} \tag{30}$$

$$\implies \mathcal{N}^{(\ell+1,t_{a+1},z)} \supseteq \bigcup_{u \in \mathcal{M}^{(\ell+1,z)}} \{\pi_u(t') \mid \mathcal{Z}\}^{\mathsf{TB}^{-1}-s}_{t'=1} \tag{31}$$

$$\text{where } \mathcal{Z} \equiv [\mathsf{N}] \setminus \mathcal{O}^{(\ell+1,t_{a+1})}_{\mathcal{M}^{(\ell+1,z)}} \text{ and } s = \left|\mathcal{O}^{(\ell+1,t_{a+1})}_{\mathcal{M}^{(\ell+1,z)}}\right| \tag{32}$$

This first implication is due to our induction hypothesis (see eq.25) which implies that the best $s$ items in the smaller set $\mathcal{N}^{(\ell+1,t_a,z)}$ is same as the best $s = \mathsf{TB}^{-1} - \left|\mathcal{O}^{(\ell+1,t_a)}_{\mathcal{M}^{(\ell+1,z)}}\right|$ items in the larger set $[\mathsf{N}] \setminus \mathcal{O}^{(\ell+1,t_a)}_{\mathcal{M}^{(\ell+1,z)}}$. Hence, it is evident that the set $\mathcal{S}$ must also be a subset of the best $\mathsf{TB}^{-1}$ items (*golden items*) for user $u$ namely $\{\pi_u(t)\}_{t \in [\mathsf{TB}^{-1}]}$. Since the above facts are true for all users $u \in \mathcal{M}^{(\ell+1,z)}$, we can also conclude that the new pruned set of items $\mathcal{N}^{(\ell+1,t_{a+1},z)}$ at the next decision round $t_{a+1}$ is a superset of the best $\mathsf{TB}^{-1} - \left|\mathcal{O}^{(\ell+1,t_{a+1})}_{\mathcal{M}^{(\ell+1,z)}}\right|$ items for every user $u$. This completes the second part of the induction proof. □

**Lemma 16.** *Conditioned on the events $\mathcal{E}^{(\ell)}_2, \mathcal{E}^{(\ell)}_3$, with choice of $\Delta_{\ell+1} = \epsilon_{\ell+1}/88\mathsf{C}$, conditions A-C will be satisfied at the beginning of the explore component of phase $\ell$ for the different nice subsets $\{\mathcal{M}^{(\ell+1,z)}\}_z$ and their corresponding set of active arms $\{\mathcal{N}^{(\ell+1,z)}\}_z$ implying that the event $\mathcal{E}^{(\ell+1)}_2$ will be true.*

*Proof.* 1. *Proof of condition A:* Due to our induction hypothesis, condition A is true for the *explore* component of phase $\ell$. Hence, the *explore* component of phase $\ell$ is implemented separately and asynchronously for each disjoint *nice* subset of users $\mathcal{M}^{(\ell,i)} \in \mathcal{M}'^{(\ell)} \subseteq \mathcal{M}^{(\ell)}$ (recall that $\mathcal{M}'^{(\ell)}$ corresponds to the nice subsets of users which do not fall into the edge case scenarios). Let us fix one such nice subset of users $\mathcal{M}^{(\ell,i)} \in \mathcal{M}'^{(\ell)}$. After the successful low rank matrix completion step (event $\mathcal{E}^{(\ell)}_3$ is true), we find the connected components of a graph $\mathcal{G}^{(\ell,i)}$ which in turn correspond to nice subsets of users as well (see Lemma 7). Since the above facts are true for all nice subsets of users in $\mathcal{M}'^{(\ell)}$, the *nice* subsets of users that progress to the $(\ell + 1)^{\text{th}}$ phase are disjoint. Since these set of users are not modified during the *exploit* component of phase $\ell + 1$, condition A is true at the beginning of the *explore* component of phase $\ell + 1$.

2. *Proof of conditions B and C:* Again, let us fix a subset of nice users $\mathcal{M}^{(\ell+1,z)}$ that has progressed to phase $\ell + 1$ and was in turn a part of the *nice* subset of users $\mathcal{M}^{(\ell,i)}$ in phase $\ell$. In other words, the set of users $\mathcal{M}^{(\ell+1,z)}$ corresponds to a connected component of the graph $\mathcal{G}^{(\ell,i)}$. From Lemma 15, we can conclude that conditions B and C are true at the beginning of the *explore* component of phase $\ell + 1$ for users in $\mathcal{M}^{(\ell+1,z)}$ with choice of $\Delta_{\ell+1} = \epsilon_{\ell+1}/88\mathsf{C}$ where $\epsilon_{\ell+1}$ was pre-determined. Therefore, the conditions B and C hold for all nice subsets of users $\{\mathcal{M}^{(\ell+1,z)}\}_z$ that have progressed to phase $\ell + 1$.

Hence, conditioned on the events $\mathcal{E}^{(\ell)}_2, \mathcal{E}^{(\ell)}_3$, with choice of $\Delta_{\ell+1} = \epsilon_{\ell+1}/88\mathsf{C}$, the algorithm will be $(\epsilon_{\ell+1}, \ell + 1)-$good and the event $\mathcal{E}^{(\ell+1)}_2$ will be true. □

## D.2 Analyzing the regret guarantee

**Lemma 17.** *Consider a fixed decreasing sequence $\{\epsilon_\ell\}_{\ell \geq 1}$ where $\epsilon_1 = ||\mathbf{P}||_\infty$ and $\epsilon_\ell = C'2^{-\ell}\min\left(||\mathbf{P}||_\infty, \frac{\sigma\sqrt{\mu}}{\log N}\right)$ for $\ell > 1$ for some constant $C' > 0$. Let us denote the event $\mathcal{E} = \bigcap_\ell \mathcal{E}_2^{(\ell)} \bigcap_\ell \mathcal{E}_3^{(\ell)}$ to imply that our algorithm is $(\epsilon_\ell, \ell)-$good at all phases indexed by $\ell$ and the explore components of all phases are successful with the length of the explore component of phase $\ell$ being*

$$m_\ell = O\left(\frac{\sigma^2\widetilde{\mu}^3 \log(M \vee N)}{\Delta_{\ell+1}^2} \max\left(1, \frac{N\tau}{M}\right)\log T\right)\right).$$

*The above statement implies that for any nice subset of users $\mathcal{M}^{(\ell,i)}$ that has progressed to the $\ell^{\text{th}}$ phase with active items $\mathcal{N}^{(\ell,i)}$ at the beginning of the explore components, with $m_\ell$ rounds, we can compute an estimate $\widetilde{\mathbf{P}}_{\mathcal{M}^{(\ell,i)},\mathcal{N}^{(\ell,i)}}$ of $\mathbf{P}_{\mathcal{M}^{(\ell,i)},\mathcal{N}^{(\ell,i)}}$ satisfying*

$$\left\|\widetilde{\mathbf{P}}_{\mathcal{M}^{(\ell,i)},\mathcal{N}^{(\ell,i)}}^{(\ell)} - \mathbf{P}_{\mathcal{M}^{(\ell,i)},\mathcal{N}^{(\ell,i)}}\right\|_\infty \leq \Delta_{\ell+1}.$$

*In that case, the event $\mathcal{E}$ is true with probability at least $1 - \mathsf{C}\mathsf{T}^{-2}$.*

*Proof.* Notice that

$$\Pr(\mathcal{E}^c) = 1 - \Pr(\bigcup_\ell \mathcal{E}_2^{(\ell)c} \bigcup_\ell \mathcal{E}_3^{(\ell)c})$$

$$\geq 1 - (\Pr(\mathcal{E}_2^{(1)c}) + \Pr(\mathcal{E}_3^{(1)c} \mid \mathcal{E}_2^{(1)}))$$

$$- \sum_{\ell>1}\left(\Pr(\mathcal{E}_2^{(\ell)c} \mid \bigcap_{\ell'<\ell}(\mathcal{E}_2^{(\ell')} \cap \mathcal{E}_3^{(\ell)})) + \Pr(\mathcal{E}_3^{(\ell)c} \mid \mathcal{E}_2^{(\ell)c}, \bigcap_{\ell'<\ell}(\mathcal{E}_2^{(\ell')} \cap \mathcal{E}_3^{(\ell)}))\right)$$

$$\geq 1 - (\Pr(\mathcal{E}_2^{(1)c}) + \Pr(\mathcal{E}_3^{(1)c} \mid \mathcal{E}_2^{(1)}))$$

$$- \sum_{\ell>1}\left(\Pr(\mathcal{E}_2^{(\ell)c} \mid \mathcal{E}_2^{(\ell-1)}, \mathcal{E}_3^{(\ell-1)}) + \Pr(\mathcal{E}_3^{(\ell)c} \mid \mathcal{E}_2^{(\ell)})\right) \geq 1 - \mathsf{C}\mathsf{T}^{-2}$$

where we used the following facts 1) $\Pr(\mathcal{E}_2^{(1)c}) = 0$ and $\Pr(\mathcal{E}_2^{(\ell)c} \mid \mathcal{E}_2^{(\ell-1)}, \mathcal{E}_3^{(\ell-1)}) = 0$ for all $\ell$ (Lemma 16) 2) $\Pr(\mathcal{E}_3^{(\ell)c} \mid \mathcal{E}_2^{(\ell)}) \leq \mathsf{C}\mathsf{T}^{-2}$ implied from Lemma 4 with additional union bounds over the number of phases (at most the number of rounds $\mathsf{T}$) and the number of disjoint nice subsets of users that have progressed in each phase (at most the number of clusters $\mathsf{C}$). An important fact to keep in mind is that the above analysis is possible since the observations used to compute estimates are never repeated in Alg. 2 and we are able to avoid complex dependencies. $\square$

Now, we are ready to prove our main regret bound. Suppose we condition on the event $\mathcal{E}$ as defined in Lemma 17. Conditioned on the event $\mathcal{E}$, let us denote by $\rho_u$ to be some sequence of items recommended to the user $u \in [\mathsf{M}]$ by our algorithm. The probability of this sequence of items being recommended is $\Pr(\bigcap_{u\in[\mathsf{M}]}\rho_u \mid \mathcal{E})$.

### D.2.1 Swapping argument

Let us fix a particular user $u \in [\mathsf{M}]$ and a sequence of recommended items $\rho_u$ such that $\rho_u(t) \in [\mathsf{N}]$ is the item recommended to user $u$ at round $t$. For sake of analysis, we will construct a permutation $\theta_u : [\mathsf{T}] \to \{\rho_u(t)\}_{t\in[\mathsf{T}]}$ of the items $\{\rho_u(t)\}_{t\in[\mathsf{T}]}$ with sequential modifications ($\theta_u$ is initialized with $\rho_u$). For any phase indexed by $\ell$, consider the *exploit* and *explore* components for a *nice* subset of users $\mathcal{M}^{(\ell,i)}$. The sequence of items recommended to user $u$ during the *explore* component remain unchanged i.e. for any phase $\ell$, $\theta_u(t) = \rho_u(t)$ for all rounds $t \in [t]$ such that $t$ corresponds to the *explore* component of phase $\ell$ for the user $u$.

Now, for the exploit component in phase $\ell$, recall that $\mathcal{O}_{\mathcal{M}^{(\ell,i)}}^{(\ell)} \setminus \mathcal{O}_{\mathcal{M}^{(\ell,i)}}^{(\ell-1)}$ is the set of items chosen for recommendation particularly in the *exploit* component of phase $\ell$ (we remove the super-script $t$ since we refer to the end of the *exploit* components in phase $\ell$ and phase $\ell - 1$ respectively). Suppose at round $t$ in the *explore* component of phase $\ell$, the $b^{\text{th}}$ item in the set $\mathcal{O}_{\mathcal{M}^{(\ell,i)}}^{(\ell)} \setminus \mathcal{O}_{\mathcal{M}^{(\ell,i)}}^{(\ell-1)}$ was chosen

to be recommended to all users in $\mathcal{M}^{(\ell,i)}$ but was found to be blocked for user $u \in \mathcal{M}^{(\ell,i)}$ before it could be recommended B times. Let us also denote $\mathcal{N}^{(\ell,t_a,i)}$ to be set of active items for users in $\mathcal{M}^{(\ell,i)}$ at round $t$ (i.e. $t_a$ was the previous decision round where it was decided whether possibility A or possibility B was true). Since the $b^{\text{th}}$ item in the set $\mathcal{O}^{(\ell)}_{\mathcal{M}^{(\ell,i)}} \setminus \mathcal{O}^{(\ell-1)}_{\mathcal{M}^{(\ell,i)}}$ was blocked for user $u$, instead we recommend any unblocked item for user $u$ from the current set of active items $\mathcal{N}^{(\ell,t_a,i)}$. This is always possible; we showed in Lemma 15 (see eq. 23) that the active set of items always contain sufficient unblocked items for possible recommendations for remaining rounds for any user in the corresponding *nice* subset of users during the exploit component. Now there are two possibilities:

1. ($b^{\text{th}}$ *item in the set* $\mathcal{O}^{(\ell)}_{\mathcal{M}^{(\ell,i)}} \setminus \mathcal{O}^{(\ell-1)}_{\mathcal{M}^{(\ell,i)}}$ *was recommended in round $t'$ in the explore component of previous phase $\ell'$)* We consider the active set of items $\mathcal{N}^{(\ell',h)}$ (such that $u \in \mathcal{M}^{(\ell',h)}$ in phase $\ell'$) where $\ell' < \ell$ is the phase index (and $t'$ is the round index) when the $b^{\text{th}}$ item in $\mathcal{O}^{(\ell)}_{\mathcal{M}^{(\ell,i)}} \setminus \mathcal{O}^{(\ell-1)}_{\mathcal{M}^{(\ell,i)}}$ (say $a$) was recommended to the user $u$ during the *explore* component. Let us denote the item that we have recommended as replacement to user $u$ at round $t$ by $a'$. Note that since $a' \in \mathcal{N}^{(\ell,t_a,i)}$, it must happen that $a' \in \mathcal{N}^{(\ell',h)}$ since $\mathcal{N}^{(\ell,t_a,i)} \subseteq \mathcal{N}^{(\ell',h)}$. In that case, we have

$$\rho_u(t) = a' \text{ and } \rho_u(t') = a$$

We swap the above items so that in the modified sequence, we have

$$\theta_u(t) = a \text{ and } \theta_u(t') = a'$$

The goal of the swapping operation at round $t$ in the *exploit* component is to modify the sequence of items such that 1) the item $a$ chosen for recommendation at round $t$ is assigned to round $t$ in the *exploit* component 2) the item $a$ actually recommended in round $t' < t$ and phase $\ell' < \ell$ is replaced by another item $a'$ (actually recommended at round $t > t'$ in phase $\ell > \ell'$) such that both $a, a'$ belongs to the same set of active items $\mathcal{N}^{(\ell',h)}$ ($u \in \mathcal{M}^{(\ell',h)}$ for some $h$). We will also say that the item $a$ *chosen for recommendation is replaced by a swapping operation of length* 1. This is because the chosen element for recommendation $a$ was recommended in the *explore* component of a previous phase. A more precise definition of the *length of a swapping operation* is provided below.

2. ($b^{\text{th}}$ *item in the set* $\mathcal{O}^{(\ell)}_{\mathcal{M}^{(\ell,i)}} \setminus \mathcal{O}^{(\ell-1)}_{\mathcal{M}^{(\ell,i)}}$ *was recommended in the exploit component of previous phase):* This is the more difficult case. As before, let us denote the $b^{\text{th}}$ item in the set $\mathcal{O}^{(\ell)}_{\mathcal{M}^{(\ell,i)}} \setminus \mathcal{O}^{(\ell-1)}_{\mathcal{M}^{(\ell,i)}}$ by $a$. Of course, the item $a$ could not have been chosen for recommendation in the *exploit* component of a previous phase (i.e. $a$ cannot belong in the set $\mathcal{O}^{(\ell')}_{\mathcal{M}^{(\ell',j)}} \setminus \mathcal{O}^{(\ell'-1)}_{\mathcal{M}^{(\ell',j)}}$ for any $\ell' < \ell$ with $u \in \mathcal{M}^{(\ell',j)}$ in phase $\ell'$.) In that case, the item $a$ was recommended in the *exploit* component of phase $\ell$ as part of a swapping operation. With this intuition in mind, let us define *length of a swapping operation* precisely:

   **Definition 3** (Length of swapping operation). *For a user $u$, suppose $a_1 \in \mathcal{O}^{(\ell)}_{\mathcal{M}^{(\ell,i)}} \setminus \mathcal{O}^{(\ell-1)}_{\mathcal{M}^{(\ell,i)}}$ is an item chosen for recommendation in the exploit component of phase $\ell$ but found to be blocked. In that case, we will say that $a_1$ is replaced by a swapping operation of length $\mathsf{L}+1$ if there exists $\mathsf{L}+1$ items $a_1, a_2, \ldots, a_{\mathsf{L}}, a_{\mathsf{L}+1}$, $\mathsf{L}+1$ phases $p_1 > p_2 > \cdots > p_{\mathsf{L}} > p_{\mathsf{L}+1}$ and respective rounds $t_1, t_2, \ldots, t_{\mathsf{L}}, t_{\mathsf{L}+1}$ such that 1) $a_1$ is chosen for recommendation in exploit component of phase $p_1$ at round $t_1$ 2) for each $2 \leq i < \mathsf{L}$, $a_{i-1}$ has been recommended in the exploit component of phase $p_i$ at round $t_i$ when the intended item chosen for recommendation was $a_i$ 3) $a_{\mathsf{L}}$ has been recommended in the explore component of phase $p_{\mathsf{L}+1}$ at round $t_{\mathsf{L}+1}$ 3) $a_{\mathsf{L}+1}$ is the item recommended to user $u$ in place of $a_1$ in phase $p_1$ at round $t_1$.*

   As before, $a_{\mathsf{L}+1}$ belongs to current set of active items $\mathcal{N}^{(\ell,t_a,i)}$ at round $t_1$ for users in $\mathcal{M}^{(\ell,i)}$ (recall that $t_a$ is the decision round just prior to $t_1$). In the above definition, note that $\mathsf{L}$ must be finite since there are a finite number of components and the first phase only has an *explore* component (recall that *exploit* component in the first phase has zero rounds).

   For a user $u \in \mathcal{M}^{(\ell,i)}$, suppose $a_1 \in \mathcal{O}^{(\ell)}_{\mathcal{M}^{(\ell,i)}} \setminus \mathcal{O}^{(\ell-1)}_{\mathcal{M}^{(\ell,i)}}$ is an item chosen for recommendation in the *exploit* component of phase $\ell$ but found to be blocked. Moreover suppose $a_1$ must be *replaced by a swapping operation of length* $\mathsf{L}$ (see Definition 3 for notations). In that case, we make the following modifications to the permutation $\theta_u$:

$$\theta_u(t_i) = a_i \text{ for all } 1 \leq i \leq \mathsf{L}+1$$

**Lemma 18.** *Condition on the event $\mathcal{E}$. Consider the modified sequence of distinct items $\{\theta_u(t)\}_{t\in[\mathsf{T}]}$ for a certain fixed user $u$. For any phase $\ell$ to which the user $u$ has progressed as part of the nice subset $\mathcal{M}^{(\ell,i)}$, consider the exploit component starting from round index $t_{\mathsf{exploit,start},\ell}$ to $t_{\mathsf{exploit,end},\ell}$. In that case, the set of elements $\{\theta_u(t) \mid t \in [t_{\mathsf{exploit,start},\ell}, t_{\mathsf{exploit,end},\ell}]\}$ is equivalent to the set of elements $\mathcal{O}^{(\ell)}_{\mathcal{M}^{(\ell,i)}} \setminus \mathcal{O}^{(\ell-1)}_{\mathcal{M}^{(\ell,i)}}$ repeated $\mathsf{B}$ times i.e. the golden items **chosen for recommendation** in the exploit component of phase $\ell$ to users in $\mathcal{M}^{(\ell,i)}$. Similarly, consider the explore component starting from round index $t_{\mathsf{explore,start},\ell}$ to $t_{\mathsf{explore,end},\ell}$. In that case, for any $t \in [t_{\mathsf{explore,start},\ell}, t_{\mathsf{explore,end},\ell}]$, it must happen that $\theta_u(t) \in \mathcal{N}^{(\ell,i)}$ i.e. the set of active items for $\mathcal{M}^{(\ell,i)}$ during the explore component of phase $\ell$.*

*Proof.* We start with the following claim.

**Claim 1.** *For a user $u \in \mathcal{M}^{(\ell,i)}$, suppose $a_1 \in \mathcal{O}^{(\ell)}_{\mathcal{M}^{(\ell,i)}} \setminus \mathcal{O}^{(\ell-1)}_{\mathcal{M}^{(\ell,i)}}$ is an item chosen for recommendation in the exploit component of phase $\ell$ at round $t_1$ but found to be blocked for user $u$. Moreover suppose $a_1$ has been replaced by a swapping operation of length $\mathsf{L}+1$ (see Definition 3 for notations). In that case, we already have $\theta_u(t_i) = a_i$ for all $2 \leq i \leq \mathsf{L}$. We only need to modify the sequence by making the following two changes: 1) $\theta_u(t_1) = a_1$ 2) $\theta_u(t_{\mathsf{L}+1}) = a_{\mathsf{L}+1}$. Note that in the true sequence, $a_{\mathsf{L}+1}$ has been recommended in the round $t_1$ replacing the intended item $a_1$.*

*Proof.* We will prove the claim via induction. Notation-wise, suppose for any $\ell' \leq \ell$, the user $u$ belongs to the set $\mathcal{M}^{(\ell',i)}$ and the set of active items at round $t_1$ for users in $\mathcal{M}^{(\ell,i)}$ is denote by $\mathcal{N}$ (for simplicity, we remove the superscripts). The base case for $\mathsf{L} = 0$ is true by construction - here the item $a_1$ is recommended in the *explore* component of phase $p_2$ at round $t_2$. We find an unblocked item $a_2$ in the set $\mathcal{N} \subseteq \mathcal{N}^{(\ell,i)}$ (note that items are never added to the active set across phases and only pruned), recommend it at round $t_1$ according to our algorithm. For analysis, we modify

$$\theta_u(t_1) = a_1 \text{ and } \theta_u(t_2) = a_2$$

Now, suppose our claim is true for some $\mathsf{L} = l$. Now, for $\mathsf{L} = l + 1$, note that $a_1$ was recommended in the *exploit* component of phase $p_2$ at round $t_2$; this implies that $a_1$ must have been used to replace another item $a_2$ via a swapping operation of length $l$. By our induction hypothesis, we must have $\theta_u(t_i) = a_i$ for all $2 \leq i \leq \mathsf{L}$ (as a matter of fact, we will have $\theta_u(t_{\mathsf{L}+1}) = a_1$). Therefore, only the pair of modifications $\theta_u(t_1) = a_1$ and $\theta_u(t_{\mathsf{L}+1}) = a_{\mathsf{L}+1}$ suffice to bring the desired changes in $\theta_u$. $\qquad\square$

We can also conclude from Claim 1 that 1) at any round $t$ in the *exploit* component of some phase for user $u$, if the chosen item to be recommended is found to be blocked, then that chosen item is brought to the $t^{\mathsf{th}}$ position in the sequence $\theta_u$ 2) Once the chosen item is brought to its correct position in $\theta_u$, it will not be modified/moved in any future round. 3) All chosen items for recommendation to user $u$ corresponding to *exploit* components are moved to their correct position (i.e. the intended round for their recommendation) in the sequence $\theta_u$. Next we make the following claim:

**Claim 2.** *Consider the setting in Claim 1. It must happen that all the items $a_1, a_2, \ldots, a_{\mathsf{L}+1}$ must belong to the set $\mathcal{N}^{(p_{\mathsf{L}+1},i)}$ - the set of active items in the phase $p_{\mathsf{L}+1}$ for users in the nice subset $\mathcal{M}^{(p_{\mathsf{L}+1},i)}$ to which $u$ belongs and $a_{\mathsf{L}}$ was recommended in its explore component.*

*Proof.* Again, note that items are never added to the active set across phases and only pruned. If an item $a_1$ is replaced by $a_{\mathsf{L}+1}$ via a swapping operation of length $\mathsf{L} + 1$, it implies that $a_1$ was used to replace item $a_2$ (via a swapping operation of length $\mathsf{L}$), $a_2$ was used to replace $a_3$ (via a swapping operation of length $\mathsf{L} - 1$) and so on. The final golden item for user $u$ in this sequence $a_{\mathsf{L}}$ was recommended in phase $p_{\mathsf{L}+1}$ in the explore component and belonged to the set of active items $\mathcal{N}^{(p_{\mathsf{L}+1},i)}$. Hence, this implies that all the subsequent surviving items $a_1, a_2, a_3, \ldots, a_{\mathsf{L}-1}, a_{\mathsf{L}+1}$ must have belonged to the set of active items $\mathcal{N}^{(p_{\mathsf{L}+1},i)}$ as well. $\qquad\square$

From Claim 2, we can conclude that for any round $t$ in the *explore* component of some phase $\ell$ for user $u$, if $\rho_u(t)$ is replaced in the sequence $\theta_u$ at round $t$, then $\rho_u(t), \theta_u(t)$ belong to the same set of active items $\mathcal{N}^{(\ell,i)}$. With both these arguments, we complete the proof of the lemma. $\qquad\square$

### D.2.2 Final Regret analysis

Let us condition on the event $\mathcal{E}$. Consider any user $u \in [M]$ - recall that $\{\rho_u(t)\}_{t \in [T]}$ is the random variable denoting the sequence of $T$ items recommended to the user $u$, $\rho_u(t)$ is a realization of $\{\rho_u(t)\}_{t \in [T]}$ conditioned on the event $\mathcal{E}$. Furthermore, $\theta_u$ is the modified sequence - a permutation of the $T$ items $\{\rho_u(t)\}$ recommended for user $u$. For simplicity of notation, we will assume that $u \in \mathcal{M}^{(\ell,i)}$ for every phase $\ell$. Hence the active set of items during the *explore* component of phase $\ell$ for the *nice* subset of users $\mathcal{M}^{(\ell,i)}$ is denoted by $\mathcal{N}^{(\ell,i)}$. Also, in line with our previous usage of notations, let us denote $\mathcal{O}^{(\ell)}_{\mathcal{M}^{(\ell,i)}} \setminus \mathcal{O}^{(\ell-1)}_{\mathcal{M}^{(\ell,i)}}$ to be the set of items chosen for recommendation particularly in the *exploit* component of phase $\ell$. Moreover, $\mathcal{O}$ denotes the entire set of items chosen for recommendation in all the *exploit* components to user $u$. In other words, if $\ell_u$ is the final phase to which user $u$ has progressed then $\mathcal{O}_u = \mathcal{O}^{(\ell_u)}_{\mathcal{M}^{(\ell_u,i)}}$. We denote the regret for the user $u$ by

$$\mathsf{Reg}_u(\mathsf{T}) \mid \mathcal{E} \triangleq = \mathbb{E}\Big[\Big(\sum_{t \in [\mathsf{T}]} \mathbf{P}_{u\pi_u(t)} - \mathbf{P}_{u\rho_u(t)}\Big) \mid \mathcal{E}\Big]$$

where the expectation is over the randomness in the algorithm and the noise in the observations. Note that with this notation, we have $\mathsf{Reg}(\mathsf{T}) \mid \mathcal{E} = \mathsf{M}^{-1} \sum_{u \in [\mathsf{M}]} \mathsf{Reg}_u(\mathsf{T}) \mid \mathcal{E}$. We now have the following set of inequalities for regret of user $u$:

$$\mathsf{Reg}_u(\mathsf{T}) \mid \mathcal{E} \triangleq \mathbb{E}\Big[\Big(\sum_{t \in [\mathsf{T}]} \mathbf{P}_{u\pi_u(t)} - \mathbf{P}_{u\rho_u(t)}\Big) \mid \mathcal{E}\Big] = \mathbb{E}\Big[\Big(\sum_{t \in [\mathsf{T}]} \mathbf{P}_{u\pi_u(t)} - \mathbf{P}_{u\rho_u(t)}\Big) \mid \mathcal{E}\Big]$$

$$= \sum_{\rho_u(t)} \Pr(\rho_u(t) = \rho_u(t) \mid \mathcal{E})\Big(\sum_{t \in [\mathsf{T}]} \mathbf{P}_{u\pi_u(t)} - \mathbf{P}_{u\theta_u(t)}\Big)$$

$$= \sum_{\rho_u(t)} \Pr(\rho_u(t) = \rho_u(t) \mid \mathcal{E})\Big(\sum_{t \in [\mathsf{T}]} \mathbf{P}_{u\pi_u(t)} - \mathbf{P}_{u\widetilde{\theta}_u(t)}\Big).$$

Let us denote the set of rounds in the *exploit* component of $\ell^{\text{th}}$ phase by $\mathsf{exploit}(\ell, \mathcal{M}^{(\ell,i)})$ and the *explore* component of $\ell^{\text{th}}$ phase by $\mathsf{explore}(\ell, \mathcal{M}^{(\ell,i)})$. Therefore we can further decompose the regret for user $u$ as follows:

$$\mathsf{Reg}_u(\mathsf{T}) \mid \mathcal{E} = \sum_{\rho_u(t)} \Pr(\rho_u(t) = \rho_u(t) \mid \mathcal{E})\Big(\sum_{\ell \in [\ell_u]} \sum_{t \in \mathsf{exploit}(\ell, \mathcal{M}^{(\ell,i)})} \Big(\mathbf{P}_{u\pi_u(t)} - \mathbf{P}_{u\widetilde{\theta}_u(t)}\Big)$$

$$+ \sum_{\ell \in [\ell_u]} \sum_{t \in \mathsf{explore}(\ell, \mathcal{M}^{(\ell,i)})} \Big(\mathbf{P}_{u\pi_u(t)} - \mathbf{P}_{u\widetilde{\theta}_u(t)}\Big)\Big).$$

Next, our arguments are conditioned on the event $\mathcal{E}$ and any sequence $\rho_u(t)$ having a non-zero probability of appearing conditioned on event $\mathcal{E}$. Recall in Lemma 15, we proved by induction that $\mathcal{O}_u \subseteq \{\pi_u(t)\}_{t=1}^{\mathsf{T}}$. Moreover, in Lemma 18, we proved that items chosen for recommendation in the *exploit* components for user $u$ are in their correct positions (i.e. the round when they were intended to be recommended but might have been found to be blocked) in the sequence $\theta_u$. Consider a permutation of the best $\mathsf{T}$ items for user $u$ $\sigma_u : [\mathsf{T}] \to \{\pi_u(t)\}_{t=1}^{\mathsf{T}}$ such that

$$\sigma_u(t) = \theta_u(t) \text{ for all } t \in \cup_\ell \mathsf{exploit}(\ell, \mathcal{M}^{(\ell,i)})$$

$$\{\sigma_u(t)\}_{t \in \cup_\ell \mathsf{explore}(\ell, \mathcal{M}^{(\ell,i)})} \equiv \{\pi_u(t')\}_{t' \in [\mathsf{T}]} \setminus \mathcal{O}_u$$

where in a round in any *exploit* component, the permutation $\sigma$ maps the item chosen for recommendation (which we know to be among the best $\mathsf{T}$ items) for user $u$ to that round. For any round belonging to the *explore* component, the permutation $\sigma$ arbitrarily maps the remaining items among the best $\mathsf{T}$ items (namely the set $\{\pi_u(t')\}_{t' \in [\mathsf{T}]} \setminus \mathcal{O}_u$). Notice that $\sum_{t \in [\mathsf{T}]} \mathbf{P}_{u\pi_u(t)} = \sum_{t \in [\mathsf{T}]} \mathbf{P}_{u\sigma_u(t)}$. Therefore, we can further decompose the regret as

$$\mathsf{Reg}_u(\mathsf{T}) \mid \mathcal{E} = \sum_{\rho_u(t)} \Pr(\rho_u(t) = \rho_u(t) \mid \mathcal{E})\Big(\sum_{\ell \in [\ell_u]} \sum_{t \in \mathsf{exploit}(\ell, \mathcal{M}^{(\ell,i)})} \Big(\mathbf{P}_{u\sigma_u(t)} - \mathbf{P}_{u\widetilde{\theta}_u(t)}\Big)$$

$$+ \sum_{\ell \in [\ell_u]} \sum_{t \in \mathsf{explore}(\ell, \mathcal{M}^{(\ell,i)})} \Big(\mathbf{P}_{u\sigma_u(t)} - \mathbf{P}_{u\widetilde{\theta}_u(t)}\Big)\Big)$$

$$= \sum_{\rho_u(t)} \Pr(\rho_u(t) = \rho_u(t) \mid \mathcal{E})\Big(\sum_{\ell \in [\ell_u]} \sum_{t \in \mathsf{explore}(\ell, \mathcal{M}^{(\ell,i)})} \Big(\mathbf{P}_{u\sigma_u(t)} - \mathbf{P}_{u\widetilde{\theta}_u(t)}\Big)\Big)$$

For the explore component in the phase indexed by $\ell$, we have proved in Lemma 14 that the active set of items $\mathcal{N}^{(\ell,i)}$ is a superset of $\{\pi_u(t')\}_{t'\in[\mathsf{T}]} \setminus \mathcal{O}_u$. Therefore, we can bound (using Lemma 4 and condition C stated at beginning of sec. D.1) for any $\ell \neq \ell_u$ ($\ell_u$ denotes index of the final phase that user $u$ was part of)

$$\sum_{t\in\mathsf{explore}(\ell,\mathcal{M}^{(\ell,i)})} \left(\mathbf{P}_{u\sigma_u(t)} - \mathbf{P}_{u\widetilde{\theta}_u(t)}\right) \leq m_\ell \cdot \epsilon_\ell = O\Big(\frac{\sigma^2\widetilde{\mu}^3\log(\mathsf{M}\vee\mathsf{N})}{\Delta_{\ell+1}^2}\max\Big(1,\frac{\mathsf{N}\tau}{\mathsf{M}}\Big)\log\mathsf{T}\Big)\Big) \cdot \epsilon_\ell.$$

where $\widetilde{\mu}$ is the incoherence factor of the sub-matrix $\mathbf{P}_{\mathcal{M}^{(\ell,i)},\mathcal{N}^{(\ell,i)}}$. Next, using the facts that $\Delta_{\ell+1} = \epsilon_{\ell+1}/88\mathsf{C}$, $2\epsilon_\ell = \epsilon_{\ell+1}$, $\mathsf{C},\tau = O(1)$, we get that (after hiding log factors for simplicity)

$$\sum_{t\in\mathsf{explore}(\ell,\mathcal{M}^{(\ell,i)})} \left(\mathbf{P}_{u\sigma_u(t)} - \mathbf{P}_{u\widetilde{\theta}_u(t)}\right) = \widetilde{O}\Big(\frac{\sigma^2\widetilde{\mu}^3}{\epsilon_\ell}\max\Big(1,\frac{\mathsf{N}}{\mathsf{M}}\Big)\Big) = \widetilde{O}\Big(\frac{\sigma^2\mu^3}{\epsilon_\ell}\max\Big(1,\frac{\mathsf{N}}{\mathsf{M}}\Big)\Big).$$

In the above statement, from Lemmas 2 and 3, we also used the fact that $\widetilde{\mu} = O(\mu)$ and the condition number of the sub-matrix $\mathbf{P}_{\mathcal{M}^{(\ell,i)},\mathcal{N}^{(\ell,i)}}$ is $O(1)$. Finally, for the *explore* component of the final phase $\ell_u$, from Lemma 11, we will also have (in the final phase, one of the edge case scenarios might appear)

$$\sum_{t\in\mathsf{explore}(\ell,\mathcal{M}^{(\ell,i)})} \left(\mathbf{P}_{u\sigma_u(t)} - \mathbf{P}_{u\widetilde{\theta}_u(t)}\right) = \widetilde{O}\Big(\sigma\mu^{3/2}\max\Big(\sqrt{\mathsf{T}},\sqrt{\frac{\mathsf{T}^2}{\mathsf{M}}}\Big)\Big) + \widetilde{O}\Big(\frac{\sigma^2\mu^3}{\epsilon_{\ell_u}}\max\Big(1,\frac{\mathsf{N}}{\mathsf{M}}\Big)\Big)$$

Hence, we can put together everything to conclude that

$$\sum_{\ell\in[\ell_u]}\sum_{t\in\mathsf{explore}(\ell,\mathcal{M}^{(\ell,i)})} \left(\mathbf{P}_{u\sigma_u(t)} - \mathbf{P}_{u\widetilde{\theta}_u(t)}\right)$$

$$= \sum_\ell \widetilde{O}\Big(\frac{\sigma^2\mu^3}{\epsilon_\ell}\max\Big(1,\frac{\mathsf{N}}{\mathsf{M}}\Big)\Big) + \widetilde{O}\Big(\sigma\mu^{3/2}\max\Big(\sqrt{\mathsf{T}},\sqrt{\frac{\mathsf{T}^2}{\mathsf{M}}}\Big)\Big)$$

$$\leq \sum_{\ell:\epsilon_\ell\leq\Phi} m_\ell\Phi + \sum_{\ell:\epsilon_\ell\geq\Phi}\widetilde{O}\Big(\frac{\sigma^2\mu^3}{\Phi}\max\Big(1,\frac{\mathsf{N}}{\mathsf{M}}\Big)\Big) + \widetilde{O}\Big(\sigma\mu^{3/2}\max\Big(\sqrt{\mathsf{T}},\sqrt{\frac{\mathsf{T}^2}{\mathsf{M}}}\Big)\Big)$$

$$\leq \mathsf{T}\Phi + \mathsf{J}\cdot\widetilde{O}\Big(\frac{\sigma^2\mu^3}{\Phi}\max\Big(1,\frac{\mathsf{N}}{\mathsf{M}}\Big)\Big) + \widetilde{O}\Big(\sigma\mu^{3/2}\max\Big(\sqrt{\mathsf{T}},\sqrt{\frac{\mathsf{T}^2}{\mathsf{M}}}\Big)\Big)$$

where $\mathsf{J}$ is the number of phases with $\epsilon_\ell \geq \Phi$. By choosing $\Phi = \sqrt{\frac{\sigma^2\mu^3}{\mathsf{T}}\max\Big(1,\frac{\mathsf{N}}{\mathsf{M}}\Big)}$, we can bound

$$\sum_{\ell\in[\ell_u]}\sum_{t\in\mathsf{explore}(\ell,\mathcal{M}^{(\ell,i)})} \left(\mathbf{P}_{u\sigma_u(t)} - \mathbf{P}_{u\widetilde{\theta}_u(t)}\right) = \widetilde{O}\Big(\mathsf{J}\sigma\mu^{3/2}\sqrt{\mathsf{T}\max\Big(1,\frac{\mathsf{N}}{\mathsf{M}}\Big)}\Big)$$

where we used the fact that $\mathsf{N} \gg \mathsf{T}$ and therefore the last term in the previous equation is a lower order term compared to the first two ones. Next, recall that we choose $\epsilon_\ell = C'2^{-\ell}\min\Big(\|\mathbf{P}\|_\infty,\frac{\sigma\sqrt{\mu}}{\log\mathsf{N}}\Big)$ (so that the condition on $\sigma > 0$ in Lemma 1 is automatically satisfied for all $\ell$) for some constant $C' > 0$, the maximum number of phases $\ell$ for which $\epsilon_\ell > \Phi$ can be bounded from above by $\mathsf{J} = O\Big(\log\Big(\frac{1}{\Phi}\min\Big(\|\mathbf{P}\|_\infty,\frac{\sigma\sqrt{\mu}}{\log\mathsf{N}}\Big)\Big)\Big)$. Therefore, we can hide $\mathsf{J}$ inside $\widetilde{O}$ and obtain

$$\sum_{\ell\in[\ell_u]}\sum_{t\in\mathsf{explore}(\ell,\mathcal{M}^{(\ell,i)})} \left(\mathbf{P}_{u\sigma_u(t)} - \mathbf{P}_{u\widetilde{\theta}_u(t)}\right) = \widetilde{O}\Big(\sigma\mu^{3/2}\sqrt{\mathsf{T}\max\Big(1,\frac{\mathsf{N}}{\mathsf{M}}\Big)}\Big).$$

Therefore, we must have that

$$\mathsf{Reg}_u(\mathsf{T})\mid\mathcal{E} = \widetilde{O}\Big(\sigma\mu^{3/2}\sqrt{\mathsf{T}\max\Big(1,\frac{\mathsf{N}}{\mathsf{M}}\Big)}\Big) \implies \mathsf{Reg}(\mathsf{T})\mid\mathcal{E} = \widetilde{O}\Big(\sigma\mu^{3/2}\sqrt{\mathsf{T}\max\Big(1,\frac{\mathsf{N}}{\mathsf{M}}\Big)}\Big).$$

Finally, we use the fact that

$$\mathsf{Reg}(\mathsf{T}) \leq \mathsf{Reg}(\mathsf{T})\mid\mathcal{E} + \Pr(\mathcal{E}^c)(\mathsf{Reg}(\mathsf{T})\mid\mathcal{E}^c).$$

From Lemma 17, we know that $\Pr(\mathcal{E}^c) \leq \mathsf{C}\mathsf{T}^{-2}$, $\mathsf{Reg}(\mathsf{T})\mid\mathcal{E}^c \leq \mathsf{T}\|\mathbf{P}\|_\infty$ therefore,

$$\mathsf{Reg}(\mathsf{T}) \leq \mathsf{Reg}(\mathsf{T})\mid\mathcal{E} + \Pr(\mathcal{E}^c)(\mathsf{Reg}(\mathsf{T})\mid\mathcal{E}^c) = \widetilde{O}\Big(\sigma\mu^{3/2}\sqrt{\mathsf{T}\max\Big(1,\frac{\mathsf{N}}{\mathsf{M}}\Big)} + \mathsf{T}^{-1}\|\mathbf{P}\|_\infty\Big)$$

which completes the proof of our main result.

# E  Proof of Lower Bound (Theorem 2)

Consider our problem setting with $M$ users, $N$ items and $T$ rounds, blocking constraint $B$, noise variance proxy $\sigma^2 = 1$ and all expected rewards in $[0, 1]$. Here we consider $C = 1$ i.e. all users belong to the same cluster. Let us denote a particular policy chosen by the recommendation system as $\pi$ that belongs to the class of polices $\Pi$. Moreover, let us also denote by $\mathcal{E}$ the set of possible environments corresponding to the expected reward matrices that has all rows to be same (satisfies cluster structure for $C = 1$). In that case, the minimax regret is given by

$$\inf_{\pi \in \Pi} \sup_{\nu \in \mathcal{E}} \mathsf{Reg}(\mathsf{T}; \mathcal{E})$$

## E.1  Lower Bound via reduction

We can ease the problem by assuming that at each round $t = 1, 2, \ldots, \mathsf{T}$, the users come in a sequential fashion - the $j^{\text{th}}$ user is recommended an item based on all previous history of observations including the feedback obtained from recommending items from users $1, 2, \ldots, j - 1$ at round $t$. Furthermore, we also assume that the $N$ items can be partitioned into $NB/T$ known groups where each group has identical items - hence we have a simple multi-armed bandit problem (MAB) with $NB/T$ arms and $MT$ rounds with a single user. A lower bound on this simplified MAB problem will imply a lower bound on our setting i.e by appropriately normalizing, we have

$$\inf_{\pi \in \Pi} \sup_{\nu \in \mathcal{E}} \mathsf{Reg}(\mathsf{T}; \mathcal{E}) \geq \frac{1}{M} \inf_{\pi \in \Pi} \sup_{\nu \in \mathcal{E}} \mathsf{Reg}_{\mathsf{MAB}}(\mathsf{MT}; \mathcal{E}) = \Omega\Big(\frac{1}{M}\sqrt{\frac{NB}{T} \cdot MT}\Big) = \Omega\Big(\sqrt{\frac{NB}{M}}\Big)$$

where we simply used the standard regret lower bound in multi-armed bandits with $K$ arms and $T$ rounds which is $\Omega(\sqrt{KT})$ [19].

## E.2  Lower Bound via application of Fano's inequality

As before, we can ease the original problem by assuming that at each round $t = 1, 2, \ldots, \mathsf{T}$, the users come in a sequential fashion - the $j^{\text{th}}$ user is recommended an item based on all previous history of observations including the feedback obtained from recommending items from users $1, 2, \ldots, j - 1$ at round $t$. Clearly, a lower bound on the simplified problem will imply a lower bound on our setting i.e by appropriately normalizing, we have

$$\inf_{\pi \in \Pi} \sup_{\nu \in \mathcal{E}} \mathsf{Reg}(\mathsf{T}; \mathcal{E}) \geq \frac{1}{M} \inf_{\pi \in \Pi} \sup_{\nu \in \mathcal{E}} \mathsf{Reg}_{\mathsf{MAB}}(\mathsf{MT}; \mathcal{E}).$$

Furthermore, ignoring the normalizing factor by $M$, for $C = 1$, the simplified problem is equivalent to a standard Multi-armed bandit (MAB) problem with a single agent with $MT$ rounds and an additional hard constraint that each item can be pulled at most $MB$ times. We will construct $\binom{N}{TB^{-1}}$ environments in the following way: let $\mathcal{T} \equiv \{\mathcal{S} \subseteq [N] \mid |\mathcal{S}| = TB^{-1}\}$ be the set of all subsets of $[N]$ of size $TB^{-1}$. Now for each subset $\mathcal{S} \in \mathcal{T}$, we construct an environment by assuming that the agent on pulling any arm in the set $\mathcal{S}$ observes a random reward distributed according to $\mathcal{N}(\Delta, 1)$ and on pulling any arm outside the set $\mathcal{S}$ observes a random reward distributed according to $\mathcal{N}(0, 1)$. This corresponds to the the reward matrix $\mathbf{P}$ (in our original problem) having an entry $\Delta$ in the $TB^{-1}$ columns indexed in $\mathcal{S}$ and $0$ in the remaining columns. Let $\mathbb{E}_{\mathcal{S}}, \mathbb{P}_{\mathcal{S}}, \mathcal{E}_{\mathcal{S}}$ denote the expectation, probability measure and the environment if $\mathcal{S} \in \mathcal{T}$ is the set of chosen columns for constructing the environment.

Next we assume that the set $\mathcal{S}$ is chosen uniformly at random from $\mathcal{T}$. Hence we must have

$$\inf_{\pi \in \Pi} \sup_{\nu \in \mathcal{E}} \mathsf{Reg}_{\mathsf{MAB}}(\mathsf{MT}; \mathcal{E}) \geq \inf_{\pi \in \Pi} \mathbb{E}_{\mathcal{S} \sim \mathcal{T}} \mathsf{Reg}_{\mathsf{MAB}}(\mathsf{T}; \mathcal{E}_{\mathcal{S}})$$

Fix any policy $\pi \in \Pi$. Condition on the set $\mathcal{S}$ being selected from $\mathcal{T}$. Let $R(\mathcal{S})$ be the number of times arms indexed in the set $\mathcal{S}$ are pulled in the $MT$ rounds. In that case we must have

$$\mathsf{Reg}_{\mathsf{MAB}}(\mathsf{T}; \mathcal{E}_{\mathcal{S}}) \geq \mathbb{P}_{\mathcal{S}}\Big(R(\mathcal{S}) \leq \frac{3MT}{4}\Big)\frac{MT\Delta}{4}.$$

Now, consider the estimation problem of which set $\mathcal{S}$ was selected from $\mathcal{T}$. Let $\widehat{X}$ be an estimator that takes as input the observations in the $MT$ rounds and returns a set $\widehat{\mathcal{S}}$ in the following way: it

finds $\widehat{\mathcal{S}}$ as the set of T arms that have been pulled the most number of times jointly and returns $\widehat{\mathcal{S}}$ if $R(\widehat{\mathcal{S}}) \geq 3MT/4$ and the null set $\emptyset$ otherwise. We consider the estimator $\widehat{X}$ to make an error if it returns a set $\widehat{\mathcal{S}}$ such that $\widehat{\mathcal{S}} \cap \mathcal{S} \leq T/4B$. If the estimator $\widehat{X}$ makes an error, note that $R(\widehat{\mathcal{S}}) \geq 3MT/4$ implies that $R(\widehat{\mathcal{S}} \setminus \mathcal{S}) \geq MT/2$ (since each arm can be pulled at most M times) - hence, it implies that $R(\mathcal{S}) \leq MT/2$. Therefore, if we denote Error as the error event, then we must have

$$\mathsf{Reg}_{\mathsf{MAB}}(\mathsf{T}; \mathcal{E}_{\mathcal{S}}) \geq \mathbb{P}_{\mathcal{S}}(R(\mathcal{S}) \leq \frac{3\mathsf{MT}}{4})\frac{\mathsf{MT}\Delta}{4} \geq \mathbb{P}_{\mathcal{S}}(\mathsf{Error})\frac{\mathsf{MT}\Delta}{4}$$

$$\implies \inf_{\pi \in \Pi} \sup_{\nu \in \mathcal{E}} \mathsf{Reg}_{\mathsf{MAB}}(\mathsf{MT}; \mathcal{E}) \geq \mathbb{P}(\mathsf{Error})\frac{\mathsf{MT}\Delta}{4} = \mathbb{E}_{\mathcal{S} \sim \mathcal{T}}\mathbb{P}_{\mathcal{S}}(\mathsf{Error})\frac{\mathsf{MT}\Delta}{4}$$

Therefore, our goal is to bound the quantity $\mathbb{E}_{\mathcal{S} \sim \mathcal{T}}\mathbb{P}_{\mathcal{S}}(\mathsf{Error})$. At this point we have reduced our problem to a multiple hypothesis testing problem. Therefore, in order to lower bound the probability of the event Error, we use Fano's inequality for approximate recovery:

**Lemma** (Fano's inequality with approximate recovery [28]). *For any random variables $V, \widehat{V}$ on alphabets $\mathcal{V}, \widehat{\mathcal{V}}$, consider an error when $d(V, \widehat{V}) \geq t$ for some $t > 0$ and distance function $d : \mathcal{V} \times \widehat{\mathcal{V}} \to \mathbb{R}$. In that case, if we denote the error event by* Error, *we must have*

$$\mathbb{P}(\mathsf{Error}) \geq 1 - \frac{I(V; \widehat{V}) + \log 2}{\log \frac{|\mathcal{V}|}{\mathsf{G}_{\max}}}$$

*where $\mathsf{G}_{\max} = \max_{\widehat{v} \in \widehat{\mathcal{V}}} \sum_{v \in \mathcal{V}} 1[d(v, \hat{v}) \leq t]$ and $I(V; \widehat{V})$ is the mutual information between the random variables $V$ and $\widehat{V}$.*

In the special case when the random variable $V$ is uniform, then we can upper bound the mutual information by $I(V; \widehat{V}) \leq \max_{v, \widehat{v} \in \mathcal{V}} \mathsf{KL}(P_{\widehat{V}|V=v}||P_{\widehat{V}|V=v'}) \leq \max_{v, \widehat{v} \in \mathcal{V}} \mathsf{KL}(P_v||P_{v'})$ where the second inequality follows from Data-processing inequality ($P_v = P(\cdot \mid v)$ corresponds to the probability of the observations given $V = v$).

Next, we apply it to our setting to prove an estimation error lower bound for our designed estimator $\widehat{X}$. In our setting, $\mathsf{G}_{\max}$ corresponds to the maximum possible number of sets in $\mathcal{T}$ that have intersection of size more than $T/4B$ with some fixed set $\mathcal{S} \in \mathcal{T}$. Clearly we have

$$\mathsf{G}_{\max} \leq \sum_{t=\mathsf{T}/4\mathsf{B}+1}^{\mathsf{T}} \binom{\mathsf{N}-\mathsf{T}}{\mathsf{T}-t}\binom{\mathsf{T}}{t} \leq \mathsf{T}\binom{\mathsf{N}-\mathsf{T}}{\mathsf{T}-\mathsf{T}/4\mathsf{B}}\binom{\mathsf{T}}{\mathsf{T}/4} \leq \mathsf{T}\left(\frac{2\mathsf{N}e}{\mathsf{T}}\right)^{\mathsf{T}-\mathsf{T}/4\mathsf{B}}\left(4e\right)^{\mathsf{T}/4\mathsf{B}}.$$

Furthermore, in our setting, we also have that

$$\max_{\mathcal{S},\mathcal{S}' \in \mathcal{T}} \mathsf{KL}(P_{\mathcal{S}}||P_{\mathcal{S}'})$$

$$\leq \sum_{i \in \mathcal{S} \cup \mathcal{S}'} \mathbb{E}_{\mathcal{S}}R(\{i\}) \max(\mathsf{KL}(\mathcal{N}(0, 1)||\mathcal{N}(\Delta, 1)), \mathsf{KL}(\mathcal{N}(\Delta, 1)||\mathcal{N}(0, 1))) \leq 2\mathsf{MBT}\Delta^2$$

- this follows from the fact that the distributions $P_{\mathcal{S}}, P_{\mathcal{S}'}$ are most separated in KL-Divergence if $\mathcal{S} \cap \mathcal{S}' = \emptyset$ and by using the fact that each arm can be pulled at most MB times. Therefore, we must have (provided $\mathsf{N} = c\mathsf{T}$ for some large enough constant $c > 0$, and T is large enough) for some constant $c' > 0$

$$\mathbb{P}(\mathsf{Error}) \geq 1 - \frac{I(\mathcal{S}; \widehat{\mathcal{S}}) + \log 2}{\log \frac{|\mathcal{T}|}{\mathsf{G}_{\max}}} \geq 1 - \frac{2\mathsf{MBT}\Delta^2 + \log 2}{\log\left(\frac{(\mathsf{N}/\mathsf{T})^{\mathsf{T}}}{\mathsf{T}\left(\frac{2\mathsf{N}e}{\mathsf{T}}\right)^{\mathsf{T}-\mathsf{T}/4\mathsf{B}}\left(4e\right)^{\mathsf{T}/4\mathsf{B}}}\right)}$$

$$\geq 1 - \frac{2\mathsf{MB}^2\mathsf{T}\Delta^2 + \mathsf{B}\log 2}{c'\mathsf{T}\log(\mathsf{N}/\mathsf{T})} \geq 0.9 - \frac{2\mathsf{MB}\Delta^2}{c'\log(\mathsf{N}/\mathsf{T})}.$$

Therefore, substituting $\Delta = \frac{c'\sqrt{\log(\mathsf{N}/\mathsf{T})}}{\mathsf{B}\sqrt{\mathsf{M}}}$, we have that for some constant $c'' \geq 0$

$$\inf_{\pi \in \Pi} \sup_{\nu \in \mathcal{E}} \mathsf{Reg}_{\mathsf{MAB}}(\mathsf{MT}; \mathcal{E}) \geq c''\mathsf{TB}^{-1}\sqrt{\mathsf{M}\log(\mathsf{N}/\mathsf{T})}$$

and therefore

$$\inf_{\pi \in \Pi} \sup_{\nu \in \mathcal{E}} \mathsf{Reg}(\mathsf{T}; \mathcal{E}) \geq \frac{1}{\mathsf{M}} \inf_{\pi \in \Pi} \sup_{\nu \in \mathcal{E}} \mathsf{Reg}_{\mathsf{MAB}}(\mathsf{MT}; \mathcal{E}) = \Omega\left(\frac{\mathsf{T}\sqrt{\log(\mathsf{N}/\mathsf{T})}}{\mathsf{B}\sqrt{\mathsf{M}}}\right).$$

**Algorithm 7** BBUIC (Blocked Latent Bandits with User and Item Clusters)

---

**Require:** Phase index $\ell$, List of disjoint nice subsets of users $\mathcal{M}^{(\ell)}$, list of corresponding subsets of active items $\mathcal{N}^{(\ell)}$, clusters $\mathsf{C}$, rounds $\mathsf{T}$, noise $\sigma^2 > 0$, round index $t_0$, exploit rounds $t_{\text{exploit}}$, estimate $\widetilde{\mathbf{P}}$ of $\mathbf{P}$, incoherence $\mu$, entry-wise error guarantee $\epsilon_\ell$ of $\widetilde{\mathbf{P}}$ restricted to all users in $\mathcal{M}^{(\ell)}$ and all items in $\mathcal{N}^{(\ell)}$, count matrix $\mathbf{K} \in \mathbb{N}^{\mathsf{M} \times \mathsf{N}}$.

1: **for** $i^{\text{th}}$ nice subset of users $\mathcal{M}^{(\ell,i)} \in \mathcal{M}^{(\ell)}$ with active items $\mathcal{N}^{(\ell,i)}$ ($i^{\text{th}}$ set in list $\mathcal{N}^{(\ell)}$) **do**
2:     Set $t = t_0$. Set $\epsilon_{\ell+1} = \epsilon_\ell/2$, $\Delta_\ell = \epsilon_\ell/88\mathsf{C}$ and $\Delta_{\ell+1} = \epsilon_{\ell+1}/88\mathsf{C}$.
3:     Run *exploit* component for users in $\mathcal{M}^{(\ell,i)}$ with active items $\mathcal{N}^{(\ell,i)}$. Obtain updated active set of items, round index and exploit rounds $\mathcal{N}^{(\ell,i)}, t, t_{\text{exploit}} \leftarrow$ Exploit_Item_Cluster$(\mathcal{M}^{(\ell,i)}, \mathcal{N}^{(\ell,i)}, t, t_{\text{exploit}}, \widetilde{\mathbf{P}}_{\mathcal{M}^{(\ell,i)}, \mathcal{N}^{(\ell,i)}}, \Delta_\ell)$.
4:     Set $d_1 = \max(|\mathcal{M}^{(\ell,i)}|, |\mathcal{N}^{(\ell,i)}|)$, $d_2 = \min(|\mathcal{M}^{(\ell,i)}|, |\mathcal{N}^{(\ell,i)}|)$ and $p = c\left(\frac{\sigma^2 \mu^3 \log d_1}{\Delta_{\ell+1}^2 d_2}\right)$ for some appropriate fixed constant $c > 0$.
5:     **if** $|\mathcal{N}^{(\ell,i)}| \geq \mathsf{T}^{1/3}$ and $p < 1$ **then**
6:         Run *explore* component for users in $\mathcal{M}^{(\ell,i)}$ with active items $\mathcal{N}^{(\ell,i)}$. Obtain updated estimate and round index $\widetilde{\mathbf{P}}, t \leftarrow$ Explore_Item_Cluster$(\mathcal{M}^{(\ell,i)}, \mathcal{N}^{(\ell,i)}, t, p)$ such that $\left\|\widetilde{\mathbf{P}}_{\mathcal{M}^{(\ell,i)}, \mathcal{N}^{(\ell,i)}} - \mathbf{P}_{\mathcal{M}^{(\ell,i)}, \mathcal{N}^{(\ell,i)}}\right\|_\infty \leq \Delta_{\ell+1}$ w.h.p.
7:         For every user $u \in \mathcal{M}^{(\ell,i)}$, compute $\mathcal{T}_u^{(\ell)} \equiv \{j \in \mathcal{N}^{(\ell,i)} \mid \widetilde{\mathbf{P}}_{u\pi_u(\mathsf{T}-t_{\text{exploit}})} - \widetilde{\mathbf{P}}_{uj} \leq 2\Delta_{\ell+1}\}$.
8:         Construct graph $\mathcal{G}^{(\ell,i)}$ whose nodes are users in $\mathcal{M}^{(\ell,i)}$ and an edge exists between two users $u, v \in \mathcal{M}^{(\ell,i)}$ if $\left|\widetilde{\mathbf{P}}_{ux}^{(\ell)} - \widetilde{\mathbf{P}}_{vx}^{(\ell)}\right| \leq 2\Delta_{\ell+1}$ for all arms $x \in \mathcal{N}^{(\ell,i)}$.
9:         Intitialize lists $\mathcal{M}_i^{(\ell+1)} = []$ and $\mathcal{N}_i^{(\ell+1)} = []$.
10:        For each connected component $\mathcal{M}^{(\ell,i,j)}$ $(\cup_j \mathcal{M}^{(\ell,i,j)} \equiv \mathcal{M}^{(\ell,i)})$, compute $\mathcal{N}^{(\ell,i,j)} \equiv \cup_{u \in \mathcal{M}^{(\ell,i,j)}} \mathcal{T}_u^{(\ell)}$.
11:        For each connected component $\mathcal{M}^{(\ell,i,j)}$, construct graph $\mathcal{G}_{\text{item}}^{(\ell,i,j)}$ whose nodes are items in $\mathcal{N}^{(\ell,i)}$ and an edge exists between two items $u, v \in \mathcal{N}^{(\ell,i)}$ if $\left|\widetilde{\mathbf{P}}_{xu}^{(\ell)} - \widetilde{\mathbf{P}}_{xv}^{(\ell)}\right| \leq 16\mathsf{C}\Delta_{\ell+1}$ for all users $x \in \mathcal{M}^{(\ell,i,j)}$. Update $\mathcal{N}^{(\ell,i,j)}$ to be the set of items $\mathcal{N}^{(\ell,i,j)} \equiv \{x \in \mathcal{N}^{(\ell,i)} \mid x$ is connected with some node in $\mathcal{N}^{(\ell,i,j)}\}$.
12:        Invoke B-LATTICE$(\ell + 1, \mathcal{M}_i^{(\ell+1)}, \mathcal{N}_i^{(\ell+1)}, \mathsf{C}, \mathsf{T}.\sigma^2, t, t_{\text{exploit}}, \widetilde{\mathbf{P}}, \epsilon_{\ell+1}, \mathbf{K}, \mathcal{G}_{\text{item}}^{(\ell,i,j)})$.
13:     **else**
14:         For each user $u \in \mathcal{M}^{(\ell,i)}$, recommend $\mathsf{T} - t$ unblocked items in $\mathcal{N}^{(\ell,i)}$ until end of rounds.
15:     **end if**
16: **end for**

---

## F Blocked Bandits having User and Item Clusters with blocking constraint $\mathsf{B} = 1$

Recall that in this setting, the $\mathsf{N}$ items can be grouped into $\mathsf{C}'$ disjoint clusters $\mathsf{D}^{(1)}, \mathsf{D}^{(2)}, \ldots, \mathsf{D}^{(\mathsf{C}')}$ that are unknown. The expected reward for user $u$ (belonging to cluster $a \in [\mathsf{C}]$) on being recommended item $j$ (belonging to cluster $b \in [\mathsf{C}']$) is $\mathbf{P}_{ij} = \mathbf{Q}_{ab}$ where $\mathbf{Q} \in \mathbb{R}^{\mathsf{C} \times \mathsf{C}'}$ is the small core reward matrix (unknown). In this setting, we provide our theoretical guarantees with $\mathsf{B} = 1$ i.e. any item can be recommended to a user only once under the blocking constraint.

Recall that the main reason our theoretical analysis required $\mathsf{B} = \Theta(\log \mathsf{T})$ for Thm. 1 setting is that if the observations that were used to compute estimates of some reward sub-matrix in a certain phase with certain guarantees that hold with some probability, then conditioning on such estimates with the said guarantees make the aforementioned observations dependent in analysis of future phases. In the BBIC setting, we show that in each phase, the possibility of the nice subsets of users and their corresponding active items is actually bounded and small. On the other hand, the possibilities were exponentially large in the number of items in the GBB setting. Therefore, we can use a *for all* argument here implying that for a set of already used observations, we can show that for all possible nice subsets of users and their active items, they can be used again to compute acceptable estimates. Such an analysis allows us to provide theoretical guarantees even with $\mathsf{B} = 1$ for the BBIC setting.

---

**Algorithm 8** `Exploit_Item_Cluster`(Exploit Component of a phase)

---

**Require:** Phase index $\ell$, nice subset of users $\mathcal{M}$, active items $\mathcal{N}$, round index $t_0$, estimate $\widetilde{\mathbf{P}}$ of $\mathbf{P}$ and error guarantee $\Delta_\ell$ such that $\left\|\widetilde{\mathbf{P}}_{\mathcal{M},\mathcal{N}} - \mathbf{P}_{\mathcal{M},\mathcal{N}}\right\|_\infty \le 88\mathsf{C}\Delta_\ell$ with high probability, similarity graph $\mathcal{G}_{\mathsf{item}}$ over the set of items $\mathcal{N}$.

1: **while** there exists $u \in \mathcal{M}^{(\ell,i)}$ such that $\widetilde{\mathbf{P}}_{u\widetilde{\pi}_u(1)|\mathcal{N}} - \widetilde{\mathbf{P}}_{u\widetilde{\pi}_u(\mathsf{T}-t_{\mathsf{exploit}})|\mathcal{N}} \ge 64\mathsf{C}\Delta_\ell\}$ **do**
2:   Compute $\mathcal{R}_u = \{j \in \mathcal{N} \mid \widetilde{\mathbf{P}}_{uj} \ge \widetilde{\mathbf{P}}_{u\widetilde{\pi}_u(1)|\mathcal{N}} - 2\Delta_{\ell+1}\}$ for every user $u \in \mathcal{M}$. Compute $\mathcal{S} = \cup_{u \in \mathcal{M}}\mathcal{R}_u$.
3:   Update $\mathcal{S} \leftarrow \{v \in \mathcal{G}_{\mathsf{item}} \mid v \text{ is connected with some node in } \mathcal{S}\}$.
4:   **for** rounds $t = t_0 + 1, t_0 + 2, \ldots, t_0 + |\mathcal{S}|\mathsf{B}$ **do**
5:     **for** each user $u \in \mathcal{M}$ **do**
6:       Denote by $x$ the $(t - t_0)^{\mathsf{th}}$ item in $\mathcal{S}$. If $\mathbf{K}_{ux} == 0$ ($x$ is unblocked), then recommend $x$ to user $u$ and update $\mathbf{K}_{ux} \leftarrow 1$. If $\mathbf{K}_{ux} == 1$ ($x$ is blocked), recommend any unblocked item $y$ in $\mathcal{N}$ (i.e $\mathbf{K}_{uy} = 0$) for the user $u$ and update $\mathbf{K}_{uy} \leftarrow 1$
7:     **end for**
8:   **end for**
9:   Update $\mathcal{N} \leftarrow \mathcal{N} \setminus \mathcal{S}$. Update $t_0 \leftarrow t_0 + |\mathcal{S}|$.
10: **end while**
11: Return $\mathcal{N}, t_0$.

---

**Algorithm 9** `Explore_Item_Cluster`(Explore Component of a phase)

---

**Require:** Phase index $\ell$, nice subset of users $\mathcal{M}$, active items $\mathcal{N}$, round index $t_0$, sampling probability $p$.

1: For each tuple of indices $(i,j) \in \mathcal{M} \times \mathcal{N}$, independently set $\delta_{ij} = 1$ with probability $p$ and $\delta_{ij} = 0$ with probability $1 - p$.
2: Denote $\Omega = \{(i,j) \in \mathcal{M} \times \mathcal{N} \mid \delta_{ij} = 1\}$ and $m = \max_{i \in \mathcal{M}} \mid |j \in \mathcal{N} \mid (i,j) \in \Omega|$ to be the maximum number of index tuples in a particular row. Initialize observations corresponding to indices in $\Omega$ to be $\mathcal{A} = \phi$.
3: **for** rounds $t = t_0 + 1, t_0 + 2, \ldots, t_0 + m$ **do**
4:   **for** each user $u \in \mathcal{M}$ **do**
5:     Find an item $z$ in $\{j \in \mathcal{N} \mid (u,j) \in \Omega, \delta_{uj} = 1\}$. If $\mathbf{K}_{uz} == 0$ ($z$ is unblocked), set $\rho_u(t) = z$ and recommend $z$ to user $u$. Observe $\mathbf{R}_{u\rho_u(t)}^{(t)}$ and update $\mathcal{A} = \mathcal{A} \cup \{\mathbf{R}_{u\rho_u(t)}^{(t)}\}$, $\mathbf{K}_{uz} \leftarrow 1$.
6:     If $\mathbf{K}_{uz} == 1$ ($z$ is blocked), recommend any unblocked item $\rho_u(t)$ in $\mathcal{N}$ s.t. $(u, \rho_u(t)) \notin \Omega$. Update $\mathbf{K}_{u\rho_u(t)} \leftarrow 1$. Set $\mathcal{A} = \mathcal{A} \cup \{\mathbf{R}_{u\rho_u(t')}^{(t')}\}$ where $t' < t$ is the round when $\rho_u(t') = z$ was recommended to user $u$.
7:   **end for**
8: **end for**
9: Compute the estimate $\widetilde{\mathbf{P}} = \mathtt{Estimate}(\mathcal{M}, \mathcal{N}, \sigma^2, \mathsf{C}, \Omega, \mathcal{A})$ and return $\widetilde{\mathbf{P}}, t_0 + m$.

---

Therefore in Algorithm 7 for the BBIC setting, we only have a single counter matrix $\mathbf{K} \in \{0,1\}^{\mathsf{M} \times \mathsf{N}}$ which is binary. The matrix $\mathbf{K}$ is initialized to be a zero matrix. Whenever an item $j$ is recommended to user $i$, we set $\mathbf{K}_{ij} = 1$. If, in a future phase, we need to recommend item $j$ to user $i$ again, due to the blocking constraint, we simply reuse the observation (see Step 6 in Alg. 9).

### F.1 Algorithm and Discussion

We start with a definition for *nice subsets of items* analogous to the *nice subset of users* (see Definition 1).

**Definition 4.** *A subset of items $\mathcal{S} \subseteq [\mathsf{N}]$ will be called "nice" if $\mathcal{S} \equiv \bigcup_{j \in \mathcal{A}} \mathcal{D}^{(j)}$ for some $\mathcal{A} \subseteq [\mathsf{C}']$. In other words, $\mathcal{S}$ can be represented as the union of some subset of clusters of items.*

As in the analysis of GBB setting, we will have the desirable properties A-C that should be satisfied with high probability at the beginning of the *explore* component of phase $\ell$. Recall that we defined the event $\mathcal{E}_2^{(\ell)}$ to be true if properties (A-C) are satisfied at the beginning of the *explore* component

of phase $\ell$ by the phased elimination algorithm. Here, we stipulate a further property D that the corresponding surviving set of items for each nice subset of users $\mathcal{M}^{(\ell,i)} \in \mathcal{M}^{(\ell)}$ is also a *nice subset of items*.

Furthermore, we defined the event $\mathcal{E}_3^{(\ell)}$ when eq. 10 is true for all nice subsets $\mathcal{M}^{(\ell,i)} \in \mathcal{M}'^{(\ell)}$ in the explore component of phase $\ell$. Algorithm 7 is a similar recursive algorithm as Algorithm 1 and the only modification is the addition of Step 11. As in Alg. 1, we instantiate Alg. 7 with phase index 1, list of *nice subsets* of users having a single element comprising all users i.e. $\mathcal{M}^{(1)} = [[\mathsf{M}]]$ and corresponding list of active items $\mathcal{N}^{(1)} = [[\mathsf{N}]]$, clusters C, rounds T, blocking constraint B, noise $\sigma^2$, round index 1, exploit rounds 0, estimate $\widetilde{\mathbf{P}}$ to be $\mathbf{0}^{\mathsf{M} \times \mathsf{N}}$, incoherence $\mu$ and $\epsilon_1 = O\left(\|\mathbf{P}\|_\infty, \frac{\sigma\sqrt{\mu}}{\log \mathsf{M}}\right)$. Here, we will have a single count matrix $\mathbf{K}$ which is binary and is initialized to be an all zero matrix. We are going to show recursively that conditioned on $\mathcal{E}_2^{(\ell)}, \mathcal{E}_3^{(\ell)}$, properties A-D will be satisfied at the beginning of phase $\ell + 1$.

**Base Case:** For $\ell = 1$ (the first phase), the number of rounds in the *exploit* component is zero and we start with the *explore* component. We initialize $\mathcal{M}^{(1,1)} = [\mathsf{N}], \mathcal{N}^{(1,1)} = [\mathsf{M}]$ and therefore, we have

$$\left| \max_{j \in \mathcal{N}^{(\ell,1)}} \mathbf{P}_{uj} - \min_{j \in \mathcal{N}^{(\ell,1)}} \mathbf{P}_{uj} \right| \le \|\mathbf{P}\|_\infty \text{ for all } u \in [\mathsf{M}].$$

Clearly, $[\mathsf{M}]$ is a *nice subset* of users, $[\mathsf{N}]$ is a *nice subset* of items and finally for every user $u \in [\mathsf{M}]$, the best $\mathsf{T}/\mathsf{B}$ items (*golden items*) $\{\pi_u(t)\}_{t=1}^{\mathsf{T}/\mathsf{B}}$ belong to the entire set of items. Thus for $\ell = 1$, conditions A-D are satisfied at the beginning of the *explore* component and therefore the event $\mathcal{E}_2^{(1)}$ is true. Furthermore, from Lemma 4, eq. 10 is true for the first phase with probability $1 - o(\mathsf{T}^{-12})$ implying that the event $\mathcal{E}_3^{(1)}$ is true with high probability.

**Inductive Argument:** Suppose, at the beginning of the phase $\ell$, we condition on the events $\bigcap_{j=1}^\ell \mathcal{E}_2^{(j)} \bigcap_{j=1}^\ell \mathcal{E}_2^{(j)}$ that Algorithm is $(\epsilon_j, j)-good$ for all $j \le \ell$. This means that conditions (A-D) are satisfied at the beginning of the *explore* component of all phases up to and including that of $\ell$ for each reward sub-matrix (indexed by $i \in [a_\ell]$) corresponding to the users in $\mathcal{M}^{(\ell,i)}$ and items in $\mathcal{N}^{(\ell,i)}$. Furthermore, we also condition on the event eq. 10 is true for all nice subsets of users $\mathcal{M}^{(\ell,i)} \in \mathcal{M}^{(\ell)}$ and their corresponding set of items $\mathcal{N}^{(\ell,i)}$ - implying that the event $\mathcal{E}_3^{(\ell)}$ is true. Recall that for a user $v$, the set $\mathcal{T}_v^{(\ell)}$ was constructed as

$$\mathcal{T}_v^{(\ell)} \equiv \{j \in \mathcal{N}^{(\ell,i)} \mid \widetilde{\mathbf{P}}_{vj} \ge \widetilde{\mathbf{P}}_{v\widetilde{\pi}_v(s')|\mathcal{N}^{(\ell,i)}} - 2\Delta_{\ell+1}\} \text{ where } s' = \mathsf{TB}^{-1} - \left|\mathcal{O}_{\mathcal{M}^{(\ell,i)}}^{(\ell)}\right| \quad (33)$$

which implies that every item in $\mathcal{T}_v^{(\ell)}$ is close to one of the *golden items* in $\mathcal{N}^{(\ell,i)}$. At the end of Step 10 in Alg. 7, we can still show that Lemma 9 holds for each set of users $\mathcal{M}^{(\ell,i,j)}$ as they form a connected component. Notice that in Step 11 in Lemma 9, for each set of users $\mathcal{M}^{(\ell,i,j)}$, we construct a graph $\mathsf{G}_{\text{item}}^{(\ell,i,j)}$ whose nodes correspond to the items in $\mathcal{N}^{(\ell,i)}$ and an edge exists between two items $u, v \in \mathcal{N}^{(\ell,i)}$ if $\left|\widetilde{\mathbf{P}}_{xu}^{(\ell)} - \widetilde{\mathbf{P}}_{xv}^{(\ell)}\right| \le 16\mathsf{C}\Delta_{\ell+1}$ for all users $x \in \mathcal{M}^{(\ell,i,j)}$. Analogous to the proof of Lemma 7, we can conclude here as well that items in the same *item cluster* form a clique. Therefore, any connected component in the graph $\mathsf{G}_{\text{item}}^{(\ell,i,j)}$ must correspond to a *nice* subset of items since condition D is true and $\mathcal{N}^{(\ell,i)}$ is already a nice subset of items. Hence the set of modified items $\mathcal{N}^{(\ell,i,j)}$ constructed at the end of Step 11 in Alg. 7 is a *nice* subset of items. Furthermore, we can also prove the corresponding version of Lemma 9:

**Lemma 19.** *Condition on the events $\mathcal{E}_2^{(\ell)}, \mathcal{E}_3^{(\ell)}$ being true. Consider a nice subset of users $\mathcal{M}^{(\ell,i,j)}$ and their corresponding set of active items $\mathcal{N}^{(\ell,i,j)}$ for which guarantees in eq. 11 holds. Fix any set $\mathcal{Y} = \mathcal{N}^{(\ell,i,j)} \setminus \mathcal{J}$ for some $\mathcal{J}$ such that for every user $u \in \mathcal{G}$, we have $\widetilde{\pi}_u(s') \mid \mathcal{N}^{(\ell,i,j)} \equiv \widetilde{\pi}_u(s) \mid \mathcal{Y}$ for $s' = \mathsf{TB}^{-1} - \left|\mathcal{O}_{\mathcal{M}^{(\ell,i)}}^{(\ell)}\right|$ and some common index $s$. In that case, we must have*

$$\max_{v \in \mathcal{G}} \left( \max_{x,y \in \mathcal{Y}} \widetilde{\mathbf{P}}_{vx} - \widetilde{\mathbf{P}}_{vy} \right) \le \max_{u \in \mathcal{G}} \left( \widetilde{\mathbf{P}}_{u\widetilde{\pi}_u(1)|\mathcal{Y}} - \widetilde{\mathbf{P}}_{u\widetilde{\pi}_u(s')|\mathcal{N}^{(\ell,i)}} \right) + 24(\mathsf{C} + \mathsf{C}')\Delta_{\ell+1}$$

*Proof.* Note that with the analysis of Lemma 9, the conclusion was true for the constructed $\mathcal{N}^{(\ell,i,j)}$ at the end of Step 10 in Alg. 7. However, in the modified set of items $\mathcal{N}^{(\ell,i,j)}$, we are only adding

items in $\mathcal{N}^{(\ell,i)}$ that either belong to the same cluster (in which case there is no added gap) or if we are adding items belonging to a different cluster, then the added gap on the RHS can be at most $\mathsf{C}'\Delta_{\ell+1}$. The above statement follows from a similar argument as in Lemma 13 where we showed that gap in expected reward between two users in $\mathcal{M}^{(\ell,i,j)}$ (or rather two users connected by a path in $\mathcal{G}^{(\ell,i)}$) for the same item is at most $O(\mathsf{C}\Delta_{\ell+1})$. $\qquad\square$

Hence, we have shown that each subset of users $\mathcal{M}^{(\ell,i,j)}$ is a *nice subset of users* and furthermore, the corresponding subset of items constructed at the end of Step 11 is *nice subset of items*. Next, we move on to the *exploit* component of phase $\ell+1$ where we use a similar trick (See Step 3 in Alg. 8) to ensure that at the end, we are left with a *nice subset of items*. As before, we can show that the constructed set of items $\mathcal{S}$ is itself a *nice subset of items*. We can again prove the following modified version of Corollary 2

**Corollary 3.** *Condition on the events* $\mathcal{E}_2^{(\ell)}, \mathcal{E}_3^{(\ell)}$ *being true. Consider a nice subset of users* $\mathcal{M}^{(\ell,i)} \in \mathcal{M}'^{(\ell)}$ *and their corresponding set of active items* $\mathcal{N}^{(\ell,i)}$ *for which guarantees in eq. 11 holds. Fix any subset* $\mathcal{Y} \subseteq \mathcal{N}^{(\ell,i)}$. *Consider two users* $u, v \in \mathcal{M}^{(\ell,i)}$ *having a path in the graph* $\mathcal{G}^{(\ell,i)}$. *Conditioned on the events* $\mathcal{E}_2^{(\ell)}, \mathcal{E}_3^{(\ell)}$, *we must have*

$$\max_{x,y \in \mathcal{S}} |\mathbf{P}_{ux} - \mathbf{P}_{uy}| \leq 16(\mathsf{C} + \mathsf{C}')\Delta_{\ell+1} \text{ and } \max_{x,y \in \mathcal{S}} |\mathbf{P}_{vx} - \mathbf{P}_{vy}| \leq 16(\mathsf{C} + \mathsf{C}')\Delta_{\ell+1}$$

*where* $\mathcal{S} \equiv \{z \in \mathcal{N}^{(\ell,i,j)} \mid z \text{ is connected with } \mathcal{R}_u^{(\ell)} \cup \mathcal{R}_v^{(\ell)}\}$.

*Proof.* The proof again follows from the fact that adding items belonging to a different cluster but connected via a path to the original items in $\mathcal{R}_u^{(\ell)} \cup \mathcal{R}_V^{(\ell)}$ can only add a term of at most $16\mathsf{C}'\Delta_{\ell+1}$ in the RHS. $\qquad\square$

Therefore, at the beginning of the *explore* component of phase $\ell+1$ for a particular *nice* subset of users $\mathcal{M}^{(\ell+1,z)}$, the corresponding set of items $\mathcal{N}^{(\ell+1,z)}$ must be a *nice* subset of items as well. Most importantly, what this implies is that for the low rank matrix completion step in the *explore* component of phase $\ell+1$, we can re-use observations from previous phases and provide theoretical guarantees as well. This is because, in eq. 10 in Lemma 4, we can take a union bound over all possible *nice subsets of users* and *all possible nice subsets of items*. Since the number of clusters $\mathsf{C}, \mathsf{C}' = O(1)$, the total possibilities is $O(1)$ as well (although exponential in the number of clusters). Hence, the complex dependencies of the previously made observations used to compute prior estimates resulted by the conditioning on surviving items and users is no longer a problem - we have simply made a *for all* argument. The rest of the analysis follows as in the GBB setting and we can arrive at a similar result as in Theorem 1 but with $\mathsf{B} = 1$ when the set of items can be clustered into $\mathsf{C}' = O(1)$ disjoint clusters.

