}\Big(\frac{2\mathsf{N}e}{\mathsf{T}}\Big)^{\mathsf{T} - \mathsf{T}/4\mathsf{B}}\Big(4e\Big)^{\mathsf{T}/4\mathsf{B}}.$$

Furthermore, in our setting, we also have that

$$\max_{\mathcal{S}, \mathcal{S}' \in \mathcal{T}} \mathsf{KL}(P_{\mathcal{S}} || P_{\mathcal{S}'})$$
$$\leq \sum_{i \in \mathcal{S} \cup \mathcal{S}'} \mathbb{E}_{\mathcal{S}}\mathsf{R}(\{i\}) \max(\mathsf{KL}(\mathcal{N}(0, 1) || \mathcal{N}(\Delta, 1)), \mathsf{KL}(\mathcal{N}(\Delta, 1) || \mathcal{N}(0, 1))) \leq 2\mathsf{MBT}\Delta^2$$

- this follows from the fact that the distributions $P_{\mathcal{S}}, P_{\mathcal{S}'}$ are most separated in KL-Divergence if $\mathcal{S} \cap \mathcal{S}' = \emptyset$ and by using the fact that each arm can be pulled at most $\mathsf{MB}$ times. Therefore, we must have (provided $\mathsf{N} = c\mathsf{T}$ for some large enough constant $c > 0$, and $\mathsf{T}$ is large enough) for some constant $c' > 0$

$$\mathbb{P}(\mathsf{Error}) \geq 1 - \frac{I(\mathcal{S}; \widehat{S}) + \log 2}{\log \frac{|\mathcal{T}|}{\mathsf{G}_{\max}}} \geq 1 - \frac{2\mathsf{MBT}\Delta^2 + \log 2}{\log\Big(\frac{(\mathsf{N}/\mathsf{T})^{\mathsf{T}}}{\mathsf{T}\Big(\frac{2\mathsf{N}e}{\mathsf{T}}\Big)^{\mathsf{T} - \mathsf{T}/4\mathsf{B}}\Big(4e\Big)^{\mathsf{T}/4\mathsf{B}}}\Big)}$$