# OpenReview forum: "Blocked Collaborative Bandits: Online Collaborative Filtering with Per-Item Budget Constraints"
_NeurIPS.cc/2023/Conference — NeurIPS 2023 poster_

### Official Review · Reviewer_HmVS · 2023-07-05

**Soundness:** 2 fair
**Presentation:** 3 good
**Contribution:** 3 good
**Rating:** 6
**Confidence:** 3

**Summary:**

This paper considers the online recommendation for a hard-clustered user.

Problem setup summary:
This problem involves $M$ users and $N$ items. the reward matrix $P \in R^{M \times N}$ is unknown. There are $T$ rounds in total, and each user receives a recommendation $\rho_u(t)$ at each round. Each user belongs to one of (unknown) clusters indexed by $\{1,2,\dots,C\}$. Each user can consume at most $B = O(\log T)$ same item.

The oracle-optimal strategy is to recommend top-($T/B$) items for $B$ (max possible) times, and the regret is the loss of recommending items except for these top arms.

This is multitask learning in the sense that the results are better than solving $N$-armed bandit for each of $M$ users (in that case, regret would be linear to $M$).

Section 2.1 introduces a bound on the accuracy of low-rank matrix completion (Lemma 1, [16]). Based on that it describes the performance of explore-then-commit with $B=1$. Section 3 introduced the B-Lattice algorithm, its regret bound (Thm 1), and regret lower bound (Thm 2)

The paper proposes B-LATTICE algorithm for online recommendation. The algorithms is phased and at each phase $l$ it obtains accuracy of $2^{-l}$. This paper is well-written. Although the algorithm described here is not truly practical in the sense it can be applied to real recommendation (see weaknesses). The contributions of the paper are enough good. My main concerns are about the solidness of the results.

#####
I think the paper is above the acceptance threshold even though the exposition can have some space for improvement, so I keep current rating (WA).

**Strengths:**

* A new scheme on the factorized recommendation with bandit settings (Tbl 1)
* parametric rate of regret
* Non-trivial (non-tight) regret lower bound using Fano's inequality.
* Simulation results in appendix

**Weaknesses:**

* Somewhat artificial budget constraint (top T/B items, where B is consistent over all items)
* Many parts of the algorithms are in the appendix (mainly, Alg 4,5)
* Alg depends on B, C that are hard-to-know before (in particular B).
* Alg is very involved.
* (minor) exposition on the order of elements can be improved.
  * For example, Lemma 1 informally describes incoherency before the main results, the main results part (Thm 1) do not include B-LATTICE detail even though it is bound on this algorithm.

**Questions:**

* Is Lemma 1 derived from Lemma 2 of [16] (or from Lemma 5 of [16])?

* Question on the applicability of Lemma1: Lemma 1 requires the sampling of $\Omega$ to be iid. I assume B-Lattice adopts the same bound, but can it be usable for such a complex sampling scheme?

* Questions on the bounds:
  * Can the authors clarify the incoherency condition (L176: $||\bar{U}||_{2,\infty}$ - this norm undefined?), and subset strong smoothness?
  * What is the dependence of Thm1 in $C$ (# of clusters)?
  * What $B = \Theta(\log T)$ exactly states? In particular, can i be $C \log T$ where $C>0$ is independent of $T,N,M$, and # of clusters?

* (minor) Are the technology used in this paper (bound on low-rank matrix completion and graphs) entirely new in this paper? How the basic technology can be compared with related work (e.g., Table 1)?

* (minor) Is there any way to simplify the algorithm? For example, can explore (Alg 2) be randomly recommending an available item, and exploit be randomly recommending an item weighted on the likelihood to be good?

**Limitations:**

Not applied.

---

> ### Author Rebuttal · Authors · 2023-08-09
>
> We thank the reviewer for making some excellent points. Below, we discuss them in details:
>
> ***1) Somewhat artificial budget constraints 2) Many parts of the algorithms are in the appendix (mainly, Alg 4,5) 3) Alg depends on B, C that are hard-to-know before (in particular B).****
>
> 1) We apologize but the first point is a bit unclear to us - we motivated the necessity of budget B from a marginal utility point of view in the Introduction. Will it be possible for the reviewer to clarify a bit more?
>
> Perhaps the reviewer is saying that B being fixed across all items is artificial - note that even if this is true, the regret problem in a fixed B setting across all items was unsolved theoretically. However, we agree that (user,item)-dependent B is an important direction of future work.
>
> 2) Due to space restrictions, it was necessary for us to keep several components in the Appendix. In the final version, with more available space, we will move back Algorithms 4,5 to the main paper.
>
> 3) Note that $B,C$ are positive integers and assumed to be very small. So,
>  even if they are unknown, we can easily estimate them very well from a small amount of exploratory data. We can definitely add a remark clarifying this in the paper.
>
> ***Is Lemma 1 derived from Lemma 2 of [16] (or from Lemma 5 of [16])?***
>
> Lemma 1 in our paper is derived from Lemma 2 of [16] (by setting $s=1$). Note that Lemma 5 of [16] is actually sub-optimal since its error guarantee has an additional undesirable factor of aspect ratio  - this is removed in Lemma 2 of [16].
>
> ***Question on the applicability of Lemma1: Lemma 1 requires the sampling of $\Omega$
>  to be i.i.d. I assume B-Lattice adopts the same bound, but can it be usable for such a complex sampling scheme?***
>
>  The reviewer is right to point out that Lemma 1 requires the sampling of $\Omega$ to be i.i.d. We do ensure this in our algorithm. For a specific sub-matrix given as input in Algorithm 2, we do sample $\Omega$ independently for matrix completion in Line 1 of Algorithm 2 - of course the sub-matrix that we choose to complete is determined based on the feedback history. More precisely, suppose Algorithm 2 is invoked with the set of users $\mathcal{M}$ , set of items $\mathcal{N}$ and sampling probability $p$. Then $\Omega$ is chosen such that every entry $(i,j)\in \mathcal{M}\times \mathcal{N}$ is present in $\Omega$ with probability $p$. Therefore, we can use the guarantees of Lemma 1 for the estimate of the sub-matrix $\mathbf{P}_{\mathcal{M},\mathcal{N}}$ (reward matrix $\mathbf{P}$ restricted to users in $\mathcal{M}$ and items in $\mathcal{N}$) .
>
>  ***1) Incoherence condition: $\left|\left|\mathbf{U}\right|\right|_{2,\infty}$ 2) subset strong smoothness (SSS)***
>
> $||\mathbf{U}||_{2,\infty}$ corresponds to the maximum euclidean norm of a row of the matrix $\mathbf{U}$.
>
> More precisely, for a matrix $\mathbf{U}\in \mathbb{R}^{M\times r}$, the norm $||\mathbf{U}||_{2,\infty}$
>
> is equal to $\max_{i \in [M]} ||\mathbf{U}_i||_2$.
>
> Thus, if $||\mathbf{U}||_{2,\infty}$  is small as in Lemma 1, then all rows of $\mathbf{U}$ have a small $\ell_2$ norm.
>
>
>  The SSS assumption states that all sub-matrices of a tall matrix restricted to at least a certain number of rows must have the minimum eigenvalue to be bounded from below.
>
> We will clarify these further in the next version of the manuscript.
>
>  ***What is the dependence of Thm1 in C?***
>
>  The regret guarantee in Theorem 1 eq. (4) scales as $O(C^3)$ . However, since $C$ is assumed to be a small constant, we have suppressed this dependence within the Big-O notation. Our main goal was to highlight the dependencies on $M,N,T$ in the regret guarantee.
>
>  ***What $B=\Theta(\log T)$ exactly states? Can it be $\dots$***
>
> Yes, the reviewer is correct. It suffices for us if $B=c\log T$ for any constant $c>c'$ where $c'$ is a constant independent of all other model parameters. This is because, we simply want $B$ to be at least the number of phases. Since our phase lengths increase exponentially, the number of phases is $O(\log T)$. We will add this clarification to the paper.
>
> ***Minor Points***
>
> 1) Yes, to the best of our knowledge, our matrix-completion based approach is completely novel. Previous works have often used a cluster first and then exploit approach which is greedy and often sub-optimal.
>
> 2) Yes, both suggestions of the reviewer - randomly recommending items in explore and recommending items weighted by some score in exploit are good suggestions for simplification and can be implemented in practice.
> We agree that in practice, our algorithm can be simplified significantly. We have in fact provided such a simplified version PB-LATTICE in Algorithm 6 (Section C in Appendix) that we have implemented in our simulations. We have also provided a greedy Explore Then Commit (ETC) Algorithm that is much simpler to state and analyze (Appendix B) but obtains sub-optimal guarantees (Remark 1)

---

> > ### Comment · Reviewer_HmVS · 2023-08-19
> > **Thank you for your response**
> >
> > * Somewhat artificial budget constraints
> >
> > I mean, for example, why the budget is $O(\log T) = o(T^{0.01})$? (as well as the homogeneity of $B$ as replied) Is there any theoretical justification for this?
> >
> > The largest limitation I acknowledge is the complexity of the algorithm and the limited experimental results described in the appendix (and in the author here in response to other reviewers).

---

> > > ### Author Response · Authors · 2023-08-20
> > > **Response from Authors**
> > >
> > > We thank the reviewer for the response. We have provided our comments below:
> > >
> > > **Artificial Budget Constraint**
> > >
> > > Note that our analysis and techniques can extend to $B=\Omega(\log T)$ with some minor technical changes. However, our goal is to provide guarantees in the practically important regime of small $B$. Our theoretical guarantees for B-LATTICE holds for $B=\Theta(\log T)$- we have left the problem of extending our theoretical guarantees to  $B=O(1)$ as a future work (see Remark 3 and Section 5 in paper). The $\log T$ term appears in the budget because of the number of phases (logarithmic in rounds) - we pull each arm at most once in each phase. Extending our guarantees to $B=o(\log T)$ requires handing dependent data in low rank matrix completion - this needs new techniques. Finally, we should point out that the greedy algorithm we propose (Algorithm 5 in Appendix B) works for any $B\ge 1$ (see Remark 1 in paper).
> > >
> > > **Complexity of Algorithm**
> > >
> > > Regarding complexity of our algorithm B-LATTICE,  at its core, B-LATTICE has 3 components namely Exploration, User Clustering and Exploitation. These $3$ components are conceptually simple - please see the descriptions in Section 4. The main complexity comes from implementing these components in a recursive manner because of the sequential nature of the data. In the final version, we will provide more explanation regarding the implementation with more available space.
> > >
> > > **Experimental Results**
> > >
> > > Please note that we have provided detailed experiments on synthetic datasets in Appendix C in the paper.  We have also provided experiments on the MovieLens dataset (real-world dataset) in the global response. We can also provide additional experiments on real datasets in the next version if the reviewer suggests so. However, we emphasize that the primary contributions of our paper are theoretical.

---

### Official Review · Reviewer_7P26 · 2023-07-06

**Soundness:** 3 good
**Presentation:** 3 good
**Contribution:** 3 good
**Rating:** 7
**Confidence:** 4

**Summary:**

In this paper, the authors tackle a collaborative-filtering type multi-armed bandit problem. There are many users, each facing a multi-armed bandit problem. However, there are only $C$ base multi-armed bandit instances, where each user comes from one of these instances. The authors provide a novel algorithm which only pulls each arm at most $B$ times per user, and provide information theoretic lower bounds for this new setting. The authors also provide an improved, practical version of this method.

**Strengths:**

1. The proposed algorithm is reasonably easy to follow, and makes intuitive sense. The authors explain the process clearly.
2. The authors provide a new lower bound, using a different Fano's inequality based method. The approach and proof of the upper bound make sense, but I have not checked the details thoroughly.
3. It would be good to emphasize that $N,M\gg T$ is actually a very relevant setting; $T$ is number of interactions per user, each user will only interact with the system a moderate number of times, but the overall number of users can be very large.

**Weaknesses:**

1. The assumptions regarding $C$ are very strong. a) that all users come from exactly 1 of $C$ instances (no within group variability), b) that $C$ is exactly known, and c) that $\tau$ is constant (it is unclear how the bound depends on $\tau$). Additional text regarding why these assumptions are reasonable, the brittleness of the algorithm to these differences, or on how these assumptions can be loosened, would strengthen the paper.
2. Empirical improvement: it does not appear that the algorithm has much better performance than the greedy baselines, unless I am misreading Figure 1? It could improve the paper to add simulations for larger $T$ to show how the asymptotic behavior compares (over Greedy). Additionally, comparing against [24] in the case where $B=T$ would be interesting. Finally, to highlight the improvement of collaboration, it could help to include a baseline of running e.g. UCB independently for each user, showing that this would yields dramatically worse performance.
3. Clarity: the work is overall quite clear and well-written, but there are some points (discussed below) that could help improve presentation.

Minor:
1. Lower bound as stated is not technically correct. Classical bandit lower bounds hold when the noise is truly Gaussian (or Bernoulli), and cannot be invoked for general sub-Gaussian random variables, as the variance proxy is an upper bound (e.g. a Gaussian with variance .01 is 1-sub-Gaussian).
2. Eqs after Line 1252 and 1259 have typing errors; maybe missing a $\geq$?

Typos:
Line 28: unknown cluster and *the* expected reward
53: suboptimal dependence on *the* number
207: "columns comprise of at"
249: "small for e.g."
Fig 1 caption: period missing before last sentence, space after PB-Lattice.
Redundancy in setting $\Delta_{\ell+1}$ in line 2 of Alg 1
Line 274,278: clarify that goal is to get an entrywise $O(2^{-\ell})$ estimate.
B-Lattice doesn't take in B as input, and this parameter doesn't show up anywhere in Alg 1. Alg 2 also doesn't take $B$ as input.
Using $\mathbf{C}$ as a counts matrix and $C$ as the number of clusters is confusing notation.


Algorithmic clarification:
1. It appears as though $\mathcal{M}^{(\ell)}$ is an *estimation* of disjoint nice subsets of users, whereas in Algorithm 1's input it is stated that they are truly nice subsets of users.
2. Line 287: each connected component is a nice subset *with high probability*? Several results need to be clarified that they're conditional on the good event in Lemma 1 (that estimation worked).
3. The algorithms defined do not appear to be "stand-alone", i.e. they have side effects involving the matrices C and D (which appear to be global variables)? A clearer description of what each algorithms requires as input, and what variables it modifies, would be helpful.

**Questions:**

See weaknesses. My primary question is regarding the assumption of knowing $C$, and that each user has exactly 1 of $C$ different mean vectors.

**Limitations:**

Discussed, modulo $C$ above.

---

> ### Author Rebuttal · Authors · 2023-08-09
>
> ***Knowledge of C***
>
> Please note that C is a small positive constant integer that does not scale with M,N,T (users,items,rounds). In fact, it suffices if we know a loose upper bound for C. Also, if C is unknown, then at the  beginning, we can use a small number of rounds to estimate C. For instance, in our model, we can use a constant number of rounds to gather data for all users corresponding to the same set of items and then fit k-means combined with the ELBOW method to estimate the value of C.
>
> ***each user has exactly 1 of C different mean vectors.***
>
> Excellent question! Although we have made this assumption for simplicity as in other papers in the literature [2,3,5,24], the assumption of each user having exactly 1 of C different mean vectors can be relaxed significantly. In fact, in [24], a similar cluster relaxation was done in the following way - there are C clusters such that 1) users in the same cluster have mean vectors  that differ entry-wise by at most $\nu$  2) for any 2 users in different clusters have mean vectors that differ entry-wise by at least $20\nu$.  Our work can be easily extended to this relaxed definition of clusters too! We will add a detailed remark regarding this in the main paper.
>
> ***$\tau$ is a constant***
>
> Again, this is for simplicity of exposition. To obtain the explicit dependence on $\tau$, we need to use Theorem 2 in Chen et. al. 2019 for matrix completion. We did the corresponding calculation and found the regret in Theorem 1 eq. (4) to be scaling as $O(\tau^3)$.
>
> ***Empirical improvement:***
>
> We thank the reviewer for very helpful and excellent questions/suggestions! We have discussed them in detail below:
>
> 1) *not having better performance than greedy baselines* - We would like to note that greedy algorithm, when run with the ``right'' exploration parameter, achieves the optimal $\sqrt{T}$ regret in MABs. For instance, see Eq (6.7) of [1] where the authors derive $\sqrt{T}$ regret of greedy with best exploration parameter. Also, see Figure 7.1 [1] where UCB barely outperforms greedy. The caveat though is that this right exploration parameter is dependent on the sub-optimality gaps which are unknown in advance; consequently, it is not implementable in practice. We believe a similar phenomenon happens in our setting (i.e., greedy with best exploration parameter is a very strong baseline with good regret guarantees). In our experiments, we compare B-LATTICE with this strong baseline and show that our algorithm outperforms (albeit slightly) the baseline. We believe this is a strong empirical evidence supporting the efficacy of our algorithm.
>
> That being said, as the reviewer suggests, we will add simulations with larger T in the revised manuscript.
>
> [1] https://tor-lattimore.com/downloads/book/book.pdf
>
> 2) \textit{ comparing against [24] in the case where $B=T$} Note that the definition of regret in the two settings is very different. For our setting, for each user, the best thing to do in hindsight is to recommend the top $T/B$ items $B$ times. On the other hand, for $B=T$, the best thing to do is to recommend the top item $T$ item. Because of this, the two algorithms are incomparable.
>
> 3) \textit{Baseline of UCB } Note that our simulations are done in the setting when $B=1$. This means that each item can be recommended only once to every user. In this setting, UCB boils down to just randomly recommending items - it becomes completely trivial. Hence there is very little point in comparing with individual UCB - it is as good as randomly recommending items.
>
> ***Lower bound as stated is not technically correct. Classical bandit lower bounds hold when the noise is truly Gaussian (or Bernoulli***
>
> Thanks for pointing it out. Our lower bounds are also proved specifically for when the noise is zero mean gaussian with unit variance. We will take care of this minor issue.
>
> We will address all the remaining minor issues/typos/clarification requests as pointed out by the reviewer in the subsequent version.
>
> ***Algorithmic Clarifications***
>
> 1) ***It appears as though $\mathcal{M}^{(\ell)}$ is an estimation of disjoint nice subsets of users, whereas in Algorithm 1's input it is stated that they are truly nice subsets of users.***
>
> The reviewer is correct. We do show that with high probability that $\mathcal{M}^{(\ell)}$ is a nice subset of users in every phase $\ell$ (Lemma 7 and Lines 890-904 in Appendix). As a matter of fact, B-LATTICE can run even if the sets of users in  $\mathcal{M}^{(\ell)}$ are not nice - but they do need to be disjoint. We will clarify these further.
>
> 2) ***Line 287: each connected component is a nice subset with high probability? Several results need to be clarified that they're conditional on the good event in Lemma 1 (that estimation worked).***
>
> Yes, each connected component is a nice subset with high probability. Thanks for pointing this out. We will clarify this clearly.
>
> 3) ***The algorithms defined do not appear to be "stand-alone", i.e. they have side effects involving the matrices C and D (which appear to be global variables)? A clearer description of what each algorithms requires as input, and what variables it modifies, would be helpful.***
>
> Yes, indeed the matrices C and D are global variables. As the reviewer suggests, we will definitely improve clarity of the algorithm further with more available space in the next version.

---

> > ### Comment · Reviewer_7P26 · 2023-08-10
> > **Response to rebuttal (increase score to 7)**
> >
> > Thanks for the clear and well-written rebuttal. These responses address my main points, and I have increased my score to a 7.
> >
> > Below I respond to specific points:
> > C: Interesting, does this "loose clustering" work off the shelf?
> > Empirical improvement: You claim that "We believe a similar phenomenon happens in our setting (i.e., greedy with best exploration parameter is a very strong baseline with good regret guarantees)". Do you have any theoretical justification for this? If so it would be very interesting + informative to include.
> >
> > Additionally, regarding the new simulations: you state that "we use standard imputation techniques to fill those missing values". This seems potentially prone to "overfitting", as this means that the matrix is being filled in via an optimization procedure, and that later B-LATTICE observes a subset of entries and performs another (potentially related) optimization procedure to guess the missing values. Additional clarification regarding how the imputation was performed, and how to verify that the gains of B-LATTICE aren't due to a matching between the two procedures, would be helpful.

---

> > > ### Author Response · Authors · 2023-08-11
> > > **Thanks and Answers to Follow-up questions**
> > >
> > > Thanks so much for the positive assessment of our work. Below, we provide answers to follow-up questions:
> > >
> > > 1) **C:** Yes, the "loose" clustering is off the shelf. We can compute the estimate of C and use some multiple of that in our algorithm as input - as we mentioned before, a loose upper bound is sufficient for our algorithm.
> > >
> > > 2) **Do you have any theoretical justification for this?** - This is a great question! Note that if the sub-optimality gaps are known, we can definitely show theoretically that greedy can do much better! As the reviewer suggests, we will include this in the final version.
> > >
> > > 3) **Clarification regarding imputation of missing values** - We follow the standard procedure as outlined in Section 4 in https://arxiv.org/abs/1606.00119. Namely, we use Python package *fancyimpute*  with the default settings. All the algorithms
> > > are completely agnostic to the process through which these matrices have been completed. We understand the reviewer's point regarding overfitting, But since the percentage of missing entries is very low compared to the available ground truth entries, there is not much chance of overfitting - this is the reason why for validation in real world experiments, we seek datasets with a very low percentage of missing values.

---

### Official Review · Reviewer_2VtF · 2023-07-06

**Soundness:** 3 good
**Presentation:** 3 good
**Contribution:** 3 good
**Rating:** 5
**Confidence:** 2

**Summary:**

The paper considers the problem of blocked collaborative bandits with multiple users that each belong to a cluster with associated bandit instance. The goal is to minimize the regret of each user while satisfying the constraint that each arm can be played at most B times for each user. The paper proposes an algorithm with sublinear regret upper bound.

**Strengths:**

- The paper proposes a novel algorithm that uses matrix completion and clustering techniques.
- Although not very surprising, a regret bound of O(sqrt{TN/M}) is proved when B=O(log T).
- The paper also proves a lower bound on the regret.

**Weaknesses:**

- The regret upper bound holds only for B=O(log T). As the papers main contribution is providing regret bounds with the existence of the budget constraint B, I believe the authors should provide the dependency of the regret bound on B.
- Assumption 1 is strong. It requires the information to not be concentrated in some clusters which in my opinion is an interesting case to deal with. Moreover, the choice of assumptions and constants is unnatural; for example why is |S|=T^{1/3} in Assumption 1.3?
- The regret lower bound does not match the achieved upper bound.

**Questions:**

- It seems that the regret lower bound in Theorem 2 is linear in T while the upper bound is sublinear?

**Limitations:**

Yes.

---

> ### Author Rebuttal · Authors · 2023-08-09
>
> We thank the reviewer for making some excellent points. Below, we discuss them in details:
>
> ***As the papers main contribution is providing regret bounds with the existence of the budget constraint B, I believe the authors should provide the dependency of the regret bound on B.***
>
> The regret scales as $\sqrt{B}$  - Since $B=O(\log T)$, this term is hidden inside the $\widetilde{O}()$ notation in the regret for simplicity - our goal was to emphasize the dependence on $M,N,T$ which are the large quantities. However, as the reviewer suggests, we will add this in the final version.
>
> ***Assumption 1 is strong. It requires the information to not be concentrated in some clusters which in my opinion is an interesting case to deal with. Moreover, the choice of assumptions and constants is unnatural; for example why is |S|=T^{1/3} in Assumption 1.3?***
>
> We completely agree with the reviewer that the information being concentrated in some clusters is an interesting case - indeed, relaxing Assumption 1 is an important research direction.
> Importantly, note that unlike previous works (for instance [11,20]), we do not assume anything about separation between cluster centers. In fact, note that in Assumption 1, we can scale down the singular values of $X$ by as much as we want to reduce the separation between cluster centers. However, such an operation does not change the regret guarantee.
>
> Also note that, we do need to start somewhere and we made a similar assumption as in Pal et. al. (2023) - [24] in the paper. Since $T$ is much smaller than $M,N$, the assumption is satisfied by a wide variety of matrices [24].
>
> We also agree that the constant $1/3$  in the expression pointed out by the reviewer might look unnatural - however, it has been made only for simplicity of exposition. To generalize, any constant $\beta<1/2$ would have been sufficient for us. This constant comes up in Step 5 of Algorithm 1 -  all we want to say is that if the number of surviving items is smaller than $T^{\beta}$  for a particular nice subset of users, then we have to recommend arbitrarily from the surviving items to the corresponding users. This leads to a order-wise smaller regret of $o(T^{1/2})$ since we show that the number of remaining rounds must always be smaller than the number of surviving items.
>
> We will clarify this point in the final manuscript with more available space.
>
> ***The regret lower bound does not match the achieved upper bound.***
>
> Yes, surprisingly, this seems to be a technically difficult problem and bridging the gap is left as future work. The difficulty, in particular, has been explained in L238-241. However, saying that, we do believe that our upper bound is tight and the lower bound can be improved.
> As mentioned in L250-255, our lower bound in Theorem 2 itself is very non-trivial and requires novel ideas involving reduction to multiple hypothesis testing problem and exploiting Fano's inequality. Note that our current lower bound matches the upper bound when $T$ is very small or very large (L243-L255).
>
>
>
> ***It seems that the regret lower bound in Theorem 2 is linear in T while the upper bound is sub-linear?***
>
> Excellent question! The second term in the lower bound the reviewer refers is of the form $\widetilde{\Omega}(T/\sqrt{M})$ assuming $B=O(\log T)$. Notice carefully that the denominator has a $\sqrt{M}$ factor but the numerator does not have a fractional power of $N$. On the other hand, the regret upper bound is of the form $\widetilde{O}(\sqrt{NT/M})$ . Since $N \gg T$, the lower bound is not a contradiction and can in fact be much smaller than the upper bound. The second term matches the upper bound only when $T\approx N$  i.e. $T$ is very large as mentioned in L251. We hope this clarifies the question.

---

### Official Review · Reviewer_tct3 · 2023-07-06

**Soundness:** 3 good
**Presentation:** 2 fair
**Contribution:** 2 fair
**Rating:** 4
**Confidence:** 3

**Summary:**

This paper studies the blocked collaborative bandits where each user represent a single bandit problem. The latent groups are assummed to exist among users in which all the users in a group share the same expected reward. But the constrant is placed on the number of serving time for each user to solve the challenge of lacking user-item interactions. Then, the authors introduce a complicated algorithms including the exploitation, exploration and clustering components. The theoretical analysis is provided and achieved a tigher regret bound for this setting.

**Strengths:**

This paper provide two regret bound. The first one recover the results and follows the regret analysis is online clustering of bandits. The second authors provide a lower bound, which consider the case when T is large.

**Weaknesses:**

The introduced algorithm is very complicated and I am worried about the efficiency of this algorithm. Authors didn't report the time cost of this algorithm and conduct ablation study for the exploration component. Because this paper tries to solve a very practical problem, the online recommendation, it only has very limited empirical evaluations, which run on the systhetic datasets. The main manuscript doesn't inlcude the empirical evaluation.

**Questions:**

In this problem setting, it seems to not have the assumption regarding the reward gap between two clusters. The reward gap assumption is commen in the problem of online clustering of bandits. How does it influence the analysis?

**Limitations:**

More empirical evaluations are needed.

---

> ### Author Rebuttal · Authors · 2023-08-09
>
> We thank the reviewer for asking some great questions. Below, we discuss them in details:
>
> ***time cost of this algorithm***
>
> Computationally, the main bottleneck of our algorithm is the matrix completion function *Estimate* invoked in Line 13 of the Explore Component (Algorithm 2). All the remaining steps have lower order run-times. Note that the  *Estimate* function is invoked at most $O(C\log T)$ times since there are can be at most $C$ disjoint nice subsets of users at a time and the number of phases is $\log T$. Moreover, note that the  *Estimate* function (Algorithm 4) solves a convex objective in Line 3 - this has a time-complexity of order $O(M^2N+N^2M)$. Moreover, a number of highly efficient techniques have been proposed for optimizing this objective even when M, N are in the order of millions (for example, see https://www.cs.utexas.edu/~cjhsieh/nuclear_icml_cameraready.pdf).
>
> ***No assumption regarding the reward gap between two clusters***
>
> Yes, indeed, we do not have any such assumption and we consider it as a strength of our work.
> - especially since previous works such as [11,20] have used this assumption crucially for a greedy solution where the users are clustered first and then collaboration is exploited. In our setting, intuitively speaking, consider two clusters whose reward gap is unknown - there are two possible cases 1) the reward gap is small -  in this case, good items for users of one cluster will also be good items for users of another cluster. We can gather more data corresponding to a common set of good items for users of both clusters jointly and estimate/complete the corresponding reward sub-matrix by exploiting the fact that the sub-matrix has rank at most $2$. 2) However, if the reward gap is large, then it will be revealed at some stage after the matrix completion step - now, we design a graph based similarity approach to cluster the users correctly.
>
> To summarize, the clustering is not done at the beginning (unlike previous works) but is done gradually as more information is obtained - thus clustering, exploration and exploitation are implemented jointly in our algorithm.
>
> ***conduct ablation study for the exploration component.***
>
> We are sorry but this statement is unclear to us. If the exploration component in the algorithm is removed, then the exploit component will not work - therefore the algorithm itself will not work anymore. Are we missing something? Can the reviewer please clarify what they mean by this statement?
>
> ***limited empirical evaluations, which run on the systhetic datasets***
>
> Please see the global post and the attached pdf. We have provided detailed experiments/plots on the well known MovieLens dataset to further validate the efficacy of our algorithm.
>
> [1] https://tor-lattimore.com/downloads/book/book.pdf
>
> Note that our main contributions are theoretical in nature and experiments are mostly for validation of theoretical guarantees.

---

> > ### Comment · Reviewer_tct3 · 2023-08-15
> >
> > Thanks for authors' response and added experiments.
> >
> > (1) Experiments. Since the authors only compared the proposed algorithm with the greedy baseline, is there any other work as same as your problem setting? From my perspective, these works for online of clustering can be modified to the problem setting by adding the constraint. But, I understand, one week is too short to add more experiments.
> >
> > (2) Gap assumption. Authors discussed two cases with two different solutions. How can your algorithm differentiate these two cases if there is no gap assumption?
> >
> > (3) Theoretical analysis. For theorem 1, How does $C$ affect your regret bound? Because these is not $C$ in your regret bound, are you assuming $C =1$? For theorem 2, I don't find a special meaning of it when considering $C=1$. Because if $C=1$, we can regard all users as one "big" user because they all have identical expected rewards. Then, this analysis can be done by using the standard MAB. We are more interested in the case when $C > O(1)$ but $C << M$.

---

> > > ### Author Response · Authors · 2023-08-16
> > > **Further Responses**
> > >
> > > We thank the reviewer for following up with some nice questions. Below, we provide detailed answer to them:
> > >
> > > **Experiments.**
> > >
> > > To answer the reviewer's question, we compare with  a strong baseline, an Alternating Minimization (AM) based algorithm introduced in [1] and modify the AM algorithm by adding the blocking constraint. Note that even without the blocking constraint, AM algorithm outperforms several baselines as reported in [1]. The AM algorithm is very general  since it handles a low rank reward matrix (a special case of which is the cluster setting).
> > >
> > > In the same MovieLens setting described in the global response (with $M=200$ users, $N=200$ items and $T=60$ rounds), the modified AM algorithm, ran with the same parameters as reported in [1], obtains a Cumulative Regret of 6901.93. On the other hand,  as shown in the plot,
> > > our algorithm obtains a Cumulative regret of 6543.91 while the greedy algorithm with 30 and 10 exploration rounds obtains a cumulative regret of 7302.19 and 7794.96 respectively. Thus our algorithm, outperforms the AM algorithm with the blocking constraint. However, we point out again that the AM algorithm does not have any theoretical guarantees at all (not even without the blocking constraint) while our algorithm has strong theoretical guarantees.
> > >
> > >
> > > [1] Alternating Linear Bandits for Online Matrix-Factorization Recommendation (2018)
> > >
> > > **Gap assumption. Authors discussed two cases with two different solutions. How can your algorithm differentiate these two cases if there is no gap assumption?**
> > >
> > > To get the intuition for how the algorithm can differentiate between the two cases, we first consider the phased elimination algorithm  for standard Multi-armed Bandits (MAB) (see Exercise [6.8] in [2]). This algorithm doesn't require the knowledge of any gap parameters, but still achieves the optimal instance dependent regret. It does this by getting more accurate estimates of the arm rewards over phases and eliminating arms based on a pre-specified threshold that decreases exponentially with each phase. If there is huge gap in the rewards between the best arm and the rest, then the sub-optimal arms will be identified and eliminated in earlier phases. On the other hand, if the gap is small, then the sub-optimal arms will be eliminated in later phases. In both these cases, it can be shown that the algorithm achieves the optimal regret while being oblivious to the gap.
> > >
> > > Our algorithm is designed along similar lines-  it runs in phases and in every phase, we estimate the user clusters and simultaneously eliminate arms within each cluster.  In the beginning of the algorithm, these clusters are coarse. As the algorithm proceeds, we refine our clusters and eliminate arms based on a pre-specified threshold. If the gap between clusters is large, our algorithm can separate these clusters in earlier phases. On the other hand, if the gap between clusters is small, our algorithm can separate them in later phases where we have more data. In both these cases,  we can show that our regret is minimax optimal, without actually knowing the distance between clusters.
> > >
> > > To be more precise, we have a graph based approach for clustering users. The graph based approach takes a pre-specificied parameter $\Delta_{\ell}$ (in phase $\ell$) that decreases exponentially with the phase index.   Here the graph corresponds to a user-similarity graph constructed based on the estimates from our low rank matrix completion module.  and subsequently, the clustering proceeds by simply considering each connected component of the user similarity graph as a cluster.
> > >
> > > If the clustering succeeds that is, the sets of users belonging to separate true clusters are identified, then we are good. But note that even if the clustering does not succeed, we show that it is because the entry-wise gap between the cluster centers must be small - i.e. the gap must be $O(\Delta_{\ell})$.  Because of this, we can treat the users jointly as a single cluster of users. Hence, we will suffer bounded regret in the exploration component in the next phase. Note that we have not used any explicit knowledge of gap anywhere. Since $\Delta_{\ell}$ decreases exponentially, the regret decreases exponentially with the phase index $\ell$  (even if we are not able to distinguish between the true clusters of users).
> > >
> > > [2] https://tor-lattimore.com/downloads/book/book.pdf
> > >
> > > **Theoretical analysis. For theorem 1, How does $C$ affect your regret bound?**
> > >
> > > Of course, the setting of C=1 is trivial as the reviewer mentions.
> > >
> > > We have assumed $C$ to be a constant (i.e. it does not scale with M,N,T) - see Assumption 2. Our regret guarantee scales polynomially in $C$ (more precisely, the dependence is $O(C^3)$) - but for simplicity of results/exposition, we have hidden the dependence of $C$ within the $\widetilde{O}(\cdot)$ notation. However, if the reviewer suggests, we can make the dependence of $C$ explicit in our result.

---

> > > > ### Comment · Reviewer_tct3 · 2023-08-17
> > > >
> > > > Thanks for the response, but my concerns are not fully addressed. For Q2, there is no lemma that can show that the proposed algorithm can separate two closed clusters or the complexity of the number of rounds needed to separate them. For Q3, the related work such as (https://arxiv.org/pdf/1401.8257.pdf),   depends on $\sqrt{C}$. $O(C^3)$ is too high.   So, I keep my original assessment.

---

> > > > > ### Author Response · Authors · 2023-08-17
> > > > > **Response from Authors**
> > > > >
> > > > > **Q2 (Gap Assumption):**   We believe there has been a misunderstanding. As we have mentioned before, our algorithm has the same intuitive idea as phased elimination algorithms for standard Multi-Armed bandits ( for the exact proof, we suggest the reviewer to first see Exercise 6.8 in [1]) that obtains the optimal regret without having the knowledge of any gap parameters. We emphasize again that our phased algorithm does not try to separate the true clusters - if the gap is large, the clusters are automatically separated in earlier phases. if the gap is small, users belonging to them continue to be treated as a single cluster until later phases and the regret will decrease with phases too. When phase index is ~ $\lceil log(1/gap)\rceil$ ($O(1/gap^2)$ rounds), the separation will also happen. Until this point, regret would only still be O(1/gap). This two case explanation is ONLY for the analysis and it does not figure in explicitly in the algorithm whose parameters are agnostic to the true gaps. Again, we emphasize that the high level idea is same as in phased elimination for MAB.
> > > > >
> > > > > For the exact proof, please see Corollary 2 in the Appendix where we explicitly demonstrate the entry-wise difference in rewards between two users in different clusters who have not been separated in a certain phase. Also see the detailed regret analysis in Appendix D.2
> > > > >
> > > > > [1] https://tor-lattimore.com/downloads/book/book.pdf
> > > > >
> > > > > **Q3(Dependence on $C$)** - We emphasize that $C$ is usually a very small constant in practice. Similar poly(C) dependence have also been provided in other related works [2,3]. **Our goal was to first obtain the right dependencies on $M,N,T$ (users, items, rounds) which are the large quantities in our setting - this itself was an important open problem in our setting with blocking constraint.**  Only, recently in [3], the optimal rates in $M,N,T$ were provided in our setting without the blocking constraint. We provide the first algorithm matching the rates in [3] in the blocked setting. However, theoretically speaking, we agree that improving the dependence on $C$ is important as future work.
> > > > >
> > > > > **Since the reviewer points to [4] for the dependence on $C$, we request them to please look at the detailed discussion of [4] given in Section 1.1 in [3].**
> > > > > We must highlight the fact that [4] solves the linear bandit problem with i.i.d contexts (without the blocking constraint). Yes, the results of [4] are applicable to our setting without the blocking constraint - however, their result gets a significantly sub-optimal dependence on $M,N$ itself (which are much larger than $C$). **In fact as discussed in [3], without the blocking constraint, [4] obtains a highly sub-optimal regret of $O(\sqrt{M^2CT}+M^3N)$**  - the dependence on $M,N$ is highly sub-optimal. Again, we emphasize that getting the right dependence on $M,N,T$ is the first priority since they are much larger than $C$ .
> > > > >
> > > > > Finally, the dependence on $C$ in our results can be improved significantly using cliques instead of connected components in the algorithm (at the cost of a more complex analysis). To simplify the analysis and presentation in the paper, we did not try to optimize the dependence on $C$. We can add a remark regarding this in the paper.
> > > > >
> > > > > [2] https://arxiv.org/abs/1606.00119
> > > > > [3] https://arxiv.org/abs/2301.07040
> > > > > [4] https://arxiv.org/pdf/1401.8257.pdf

---

### Official Review · Reviewer_tR2a · 2023-07-12

**Soundness:** 3 good
**Presentation:** 2 fair
**Contribution:** 3 good
**Rating:** 7
**Confidence:** 3

**Summary:**

This paper studies blocked collaborative bandits. Multiple users, clustered in a fixed number of clusters according to the rewards obtained from them are provided recommendations in each discrete time instance till T timesteps. The paper proposes algorithms that minimize the cumulative regret with the constraint that no user is recommended the same arm (item ) more than B times. The authors propose an algorithm called B-LATTICE  that collaborates across users while satisfying budget constraints, and minimising regret. The authors show theoretical guarantees of the proposed algorithm and also validate their results empirically with synthetic data.

**Strengths:**

The paper introduces an important setting with practical application. The main strength of the paper is its sublinear regret guarantee. The authors give the first sublinear regret algorithm in the proposed setting. Furthermore, the analysis and proposed algorithms are involved and nontrivial; the authors do an excellent job of providing an intuitive understanding of their proposed algorithms.

**Weaknesses:**

The paper does a poor job of explaining the setting and its connection to Bandits. Following are my questions and (some) suggestions to the authors.

1. A crisp summary of how the problem is MAB problem is missing from the introduction/setting section. For instance, the following points are not clear in the first reading of the first 2 sections; what constitutes an arm (items?), who pulls arms (a recommendation engine or users), it is not clear that rewards are not only arm-specific but also user dependent (which in my opinion most important difference from classic MAB setting).

2. Can authors shed some light on what is known apriori to the algorithm/recommendation engine? For instance, the authors use the fact that the number of true clusters  $C$ is known in Algorithm B-LATTICE, however, I don't understand why T is needed as an input. In particular. can the regret guarantee of the proposed algorithm be converted into an anytime guarantee (for any value of large enough T)? In contrast, in the Greedy algorithm (Remark 1) authors explicitly use T.

3. I don't see why the authors normalize the cumulative regret by $M$ (Equation 2). Is it crucial to consider per-user regret guarantee? Howe do the results fare without this normalization?





**Questions:**

see weaknesses.

---

> ### Author Rebuttal · Authors · 2023-08-09
>
> We thank the reviewer for making some excellent points. Below, we discuss them in details:
>
> ***A crisp summary of how the problem is MAB problem is missing from the introduction/setting section. For instance, the following points are not clear in the first reading of the first 2 sections; what constitutes an arm (items?), who pulls arms (a recommendation engine or users), it is not clear that rewards are not only arm-specific but also user dependent***
>
> We apologize for the ambiguity. We will clarify this point in the final manuscript with more available space.
>
> To answer the reviewer's question, as discussed in Section 2, our setting has $M$ users, $N$ items and $T$ rounds. The expected (user, item) affinity is both item-specific and user-dependent - captured by the $M\times N$ reward matrix $\mathbf{P}$. In each round, for each user $u$, the recommendation engine recommends an unblocked item $\rho_u(t)$ (possibly different for each user) based on the feedback history until the previous round.  Subsequently the recommendation engine observes noisy feedback  for the recommendations of each of the $M$ users at round $t$.
>
> This corresponds to the following analogue of multi-armed bandits (MAB) - we have $M$ users each involved in a separate MAB problem with the same set of $N$ arms (corresponding to the $N$ items) and T rounds. The mean reward of each arm is different for each user captured by the $M\times N$ reward matrix $\mathbf{P}$. In each round $t$, every agent $u$ simultaneously pulls an unblocked arm $\rho_u(t)$ of their choice based on the feedback history  of all users (including $u$) from previous rounds. Subsequently the noisy feedback for all users and their corresponding arm pulls at round $t$ is revealed to everyone.
>
> ***Can authors shed some light on what is known apriori to the algorithm/recommendation engine? For instance, the authors use the fact that the number of true clusters $C$ is known in Algorithm B-LATTICE, however, I don't understand why T is needed as an input. In particular. can the regret guarantee of the proposed algorithm be converted into an anytime guarantee (for any value of large enough T)? In contrast, in the Greedy algorithm (Remark 1) authors explicitly use T.***
>
> Great question! Algorithm 1/Recommendation engine has apriori access to $M,N,C,T,B,\sigma^2$ corresponding to users, items, clusters, rounds, budget and noise variance. Note that the number of rounds $T$ is only needed in Step 5 of Algorithm 1 for a technical/theoretical reason - here we say that if the number of surviving items is smaller than $T^{1/3}$ for a  particular nice subset of users, then we directly go to Step 13 in Algorithm 1 and recommend unblocked items in the surviving set arbitrarily to the corresponding users in the nice subset until the end of rounds. Note that we can get around this issue by using doubling trick commonly used in bandit literature for anytime algorithms (Besson and Kaufmann 2018) - however we have resisted its use for simplicity. Moreover, in practice, we do not need this step at all - see Algorithm 6 (a simplified version of B-LATTICE) in Appendix C for instance that we have implemented for our experiments.
>
> The reviewer is correct to point out that in the greedy algorithm, the value of $T$ is required crucially in contrast.
>
> ***I don't see why the authors normalize the cumulative regret by $M$ (Equation 2). Is it crucial to consider per-user regret guarantee? Howe do the results fare without this normalization?***
>
> The normalization factor of $M$ does not hold much significance as the reviewer points out. As a starting point, our goal was to study the average regret over users and hence the normalization over $M$. However, regret is an additive quantity and therefore removing the normalization factor will simply lead to a re-scaled regret of $\widetilde{O}(\sqrt{(M+N)T})$.

---

> > ### Comment · Reviewer_tR2a · 2023-08-10
> >
> > Thank you for your detailed response to my questions. I maintain my original view on the acceptance of the paper.

---

> > > ### Author Response · Authors · 2023-08-13
> > > **Thanks!**
> > >
> > > We thank the reviewer for their positive assessment of our paper. Please let us know if there is anything else that we can clarify.

---

### Author Rebuttal · Authors · 2023-08-09

**Experiments on Movielens-10M dataset**

Reviewer tct3 requested for additional experiments on real-world datasets - in response, we show a set of experiments on the popular MovieLens dataset. Relevant figures can be found in the attached pdf.

We take the Movielens 10M dataset (https://grouplens.org/datasets/movielens/10m/), take 200 users who have rated most movies, 200 movies that have been rated the most and consider the $200\times 200$ reward matrix associated with those users and movies. The number of missing entries in that matrix is $<20$% and we use standard imputation techniques to fill those missing values. We consider this reward matrix to be the ground truth and remains unknown to the algorithm. Subsequently, we experiment with our set-up  - $M=200$ users, $N=200$ items, number of rounds $T=60$ and budget $B=1$. In every round, one chosen unblocked item is consumed by the each user who then provides a noisy feedback - the expectation of this feedback is the corresponding entry of the ground truth reward matrix and the additive noise is zero-mean gaussian with variance $0.5$. The goal is to sequentially recommend items to every user based on prior history and minimize the regret across all users (see equation 2 in the paper). We compare our algorithm B-LATTICE (a simplified version in Algorithm 6) with ETC algorithm (Algorithm 5) and tuned exploration parameter. All relevant plots have been provided in the attached pdf.

Note that the greedy/ETC algorithm, when run with the ``right'' exploration parameter, achieves the optimal regret in MABs. For instance, see Eq (6.7) of [1] where the authors derive optimal regret of greedy with best exploration parameter. Also, see Figure 7.1 [1] where UCB barely outperforms greedy. The caveat though is that this right exploration parameter is dependent on the sub-optimality gaps which are unknown in advance; consequently, it is not implementable in practice.  In our experiments, we compare B-LATTICE with this strong baseline  and show that our algorithm outperforms the baseline.  Furthermore, we also show that our algorithm has a small cold-start period (time before it starts meaningful recommendations)  and also has high round-wise reward in every round - in other words, it makes good recommendations in every round. On the other hand, the greedy/ETC algorithm (with $30$ exploration rounds) has a significantly large cold-start period (30 rounds).

[1] https://tor-lattimore.com/downloads/book/book.pdf

---

### Decision · Program_Chairs · 2023-09-21

**Decision:**

Accept (poster)

**Comment:**

Based on the discussion in the review period, the reviewers see this paper as having the following main strengths:
- The paper proposes a novel algorithm that uses matrix completion and clustering techniques.
- Solid regret upper and lower bounds
- different from many previous works, this work makes no gap assumption between the clusters
- Reasonable experiments for a theory paper

The main weaknesses recognized were:
- The Algorithm is very involved (although the presentation was reasonable) and needs to have knowledge of usually-unknown parameters
- The dependence on the number of clusters in the regret bound is high ($O(C^3)$)
- Many clarity improvements necessary, based on the author-reviewer discussions

Overall, the reviewers think that the strengths outweigh the concerns for this paper.